# CRITICAL BATCH SIZE MINIMIZES STOCHASTIC FIRST-ORDER ORACLE COMPLEXITY OF DEEP LEARNING OPTIMIZER USING HYPERPARAMETERS CLOSE TO ONE

## ABSTRACT

Practical results have shown that deep learning optimizers using small constant learning rates, hyperparameters close to one, and large batch sizes can find the model parameters of deep neural networks that minimize the loss functions. We first show theoretical evidence that the momentum method (Momentum) and adaptive moment estimation (Adam) perform well in the sense that the upper bound of the theoretical performance measure is small with a small constant learning rate, hyperparameters close to one, and a large batch size. Next, we show that there exists a batch size called the critical batch size minimizing the stochastic first-order oracle (SFO) complexity, which is the stochastic gradient computation cost, and that SFO complexity increases once the batch size exceeds the critical batch size. Finally, we provide numerical results that support our theoretical results. That is, the numerical results indicate that Adam using a small constant learning rate, hyperparameters close to one, and the critical batch size minimizing SFO complexity has faster convergence than Momentum and stochastic gradient descent (SGD).

## 1 INTRODUCTION

### 1.1 BACKGROUND

Useful deep learning optimizers have been proposed to find the model parameters of the deep neural networks that minimize loss functions called the expected risk and empirical risk, such as stochastic gradient descent (SGD) (Robbins & Monro, 1951; Zinkevich, 2003; Nemirovski et al., 2009; Ghadimi & Lan, 2012; 2013), momentum methods (Polyak, 1964; Nesterov, 1983), and adaptive methods. The various adaptive methods include Adaptive Gradient (AdaGrad) (Duchi et al., 2011), Root Mean Square Propagation (RMSProp) (Tieleman & Hinton, 2012), Adaptive Moment Estimation (Adam) (Kingma & Ba, 2015), Adaptive Mean Square Gradient (AMSGrad) (Reddi et al., 2018), Yogi (Zaheer et al., 2018), Adam with decoupled weight decay (AdamW) (Loshchilov & Hutter, 2019), and AdaBelief (named for adapting stepsizes by the belief in observed gradients) (Zhuang et al., 2020).

Theoretical analyses of adaptive methods for nonconvex optimization were presented in (Zaheer et al., 2018; Zou et al., 2019; Chen et al., 2019; Zhou et al., 2020; Zhuang et al., 2020; Chen et al., 2021) (see (Jain et al., 2018; Fehrman et al., 2020; Chen et al., 2020; Scaman & Malherbe, 2020; Loizou et al., 2021) for convergence analyses of SGD). A particularly interesting feature of adaptive methods is the use of hyperparameters, denoted by $\beta_1$ and $\beta_2$, that can be set to influence the method performance $\mathrm{P}(K) := \frac{1}{K} \sum_{k=1}^{K} \mathbb{E}[\|\nabla f(\boldsymbol{\theta}_k)\|^2]$, where $\nabla f$ is the gradient of a loss function $f \colon \mathbb{R}^d \to \mathbb{R}$, $(\boldsymbol{\theta}_k)_{k=1}^{K}$ is the sequence generated by an optimizer, and $K$ is the number of steps. The previous results are summarized in Table 1 indicating that using $\beta_1$ and/or $\beta_2$ close to 0 makes the upper bound of $\mathrm{P}(K)$ small (see also Appendix A.1).

Meanwhile, practical results for adaptive methods were presented in (Kingma & Ba, 2015; Reddi et al., 2018; Zaheer et al., 2018; Zou et al., 2019; Chen et al., 2019; Zhuang et al., 2020;

Chen et al., 2021). These studies have shown that using, for example, $\beta_1 \in \{0.9, 0.99\}$ and $\beta_2 \in \{0.99, 0.999\}$ provides superior performance for training deep neural networks. The practically useful $\beta_1$ and $\beta_2$ are each close to 1, whereas in contrast, the theoretical results (Table 1) show that using $\beta_1$ and/or $\beta_2$ close to 0 makes the upper bounds of the performance measures small.

Table 1: Upper bounds of performance measure of optimizers with learning rate $\alpha_k$ and hyperparameters $\beta_1$ and $\beta_2$ ($G > 0$, $s \in (0, 1/2)$, $L$ denotes the Lipschitz constant of the Lipschitz continuous gradient of the loss function $f$, $K$ denotes the number of steps, $b$ is the batch size, $\alpha_{b,\max}$ depends on $b$ and the largest eigenvalue of the Hessian of $f$, $h$ is a monotone decreasing function with respect to $\beta_1$, and $C_3$ is defined as in Table 2. $\beta \approx a$ implies that, if $\beta$ is close to $a$, then the upper bounds are small.)

| Optimizer | Learning Rate $\alpha_k$ | Parameters $\beta_1, \beta_2$ | Upper Bound |
|---|---|---|---|
| Tail-averaged SGD (Jain et al., 2018) | $\mathcal{O}(\alpha_{b,\max})$ | $\beta_1 = 0$ $\beta_2 = 0$ | $\mathcal{O}\left(\dfrac{1}{K^2} + \dfrac{1}{Kb}\right)$ |
| Adam (Zaheer et al., 2018) | $\mathcal{O}\left(\dfrac{1}{L}\right)$ | $\beta_1 = 0$ $\beta_2 \geq 1 - \mathcal{O}\left(\dfrac{1}{G^2}\right)$ | $\mathcal{O}\left(\dfrac{1}{K} + \dfrac{1}{b}\right)$ |
| Generic Adam (Zou et al., 2019) | $\mathcal{O}\left(\dfrac{1}{\sqrt{k}}\right)$ | $\beta_1 \approx 0,$ $\beta_2 = 1 - \dfrac{1}{k} \approx 1$ | $\mathcal{O}\left(\dfrac{\log K}{\sqrt{K}}\right)$ |
| AdaFom (Chen et al., 2019) | $\dfrac{1}{\sqrt{k}}$ | $\beta_1 \approx 0$ | $\mathcal{O}\left(\dfrac{\log K}{\sqrt{K}}\right)$ |
| AMSGrad (Zhou et al., 2020) | $\alpha$ | $0 \approx \beta_1 < \sqrt{\beta_2}$ | $\mathcal{O}\left(\dfrac{1}{K^{\frac{1}{2}-s}}\right)$ |
| AdaBelief (Zhuang et al., 2020) | $\mathcal{O}\left(\dfrac{1}{\sqrt{k}}\right)$ | $\beta_1 \approx 0, \beta_2 \approx 0$ | $\mathcal{O}\left(\dfrac{\log K}{\sqrt{K}}\right)$ |
| Padam (Chen et al., 2021) | $\alpha$ | $\beta_1 \approx 0, \beta_2 \approx 0$ | $\mathcal{O}\left(\dfrac{1}{K^{\frac{1}{2}-s}}\right)$ |
| Adam, AMSGrad (this paper) | $\alpha$ varying $\alpha_k$ | $\beta_1 \approx 1, \beta_2 \approx 1$ $\beta_1 \approx 1, \beta_2 \approx 1$ | $\mathcal{O}\left(\dfrac{1}{K} + \dfrac{1}{b}\right) + C_3$ $\mathcal{O}\left(\dfrac{1}{K} + \dfrac{1}{Kb}\right) + h(\beta_1)$ |

The practical performance of a deep learning optimizer strongly depends on the batch size. In (Smith et al., 2018), it was numerically shown that using an enormous batch size leads to a reduction in the number of parameter updates and model training time. The theoretical results in (Zaheer et al., 2018) showed that using large batch sizes makes the upper bound of $\mathrm{P}(K)$ of an adaptive method small (Table 1). Convergence analyses of SGD in (Cotter et al., 2011; Chen et al., 2020; Arjevani et al., 2022) indicated that running SGD with a decaying learning rate and large batch size for sufficiently many steps leads to convergence to a local minimizer of a loss function. Accordingly, the practical results for large batch sizes match the theoretical ones.

In (Shallue et al., 2019; Zhang et al., 2019), it was studied how increasing the batch size affects the performances of deep learning optimizers. In both studies, it was numerically shown that increasing batch size tends to decrease the number of steps $K$ needed for training deep neural networks, but with diminishing returns. Moreover, it was shown that momentum methods can exploit larger

batches than SGD (Shallue et al., 2019), and that K-FAC and Adam can exploit larger batches than momentum methods (Zhang et al., 2019).

## 1.2 MOTIVATION

### 1.2.1 HYPERPARAMETERS CLOSE TO ONE AND CONSTANT LEARNING RATE

As described in Section 1.1, the practically useful $\beta_1$ and $\beta_2$ are each close to 1, whereas in contrast, the theoretical results show that using $\beta_1$ and/or $\beta_2$ close to 0 makes the upper bounds of the performance measures small. Hence, there is a gap between theory ($\beta_1, \beta_2 \approx 0$) and practice ($\beta_1, \beta_2 \approx 1$) for adaptive methods. As a consequence, the first motivation of this paper is to bridge this gap.

Since using small constant learning rates is robust for training deep neural networks (Kingma & Ba, 2015; Reddi et al., 2018; Zaheer et al., 2018; Zou et al., 2019; Chen et al., 2019; Zhuang et al., 2020; Chen et al., 2021), we focus on using a small constant learning rate $\alpha$. Here, we note that using a learning rate depending on the Lipschitz constant $L$ of the gradient $\nabla f$ would be unrealistic. This is because computing the Lipschitz constant $L$ is NP-hard (Virmaux & Scaman, 2018, Theorem 2). The results ("(this paper)" row in Table 1) using varying learning rates are given in Appendix A.4.

### 1.2.2 CRITICAL BATCH SIZE

The second motivation of this paper is to clarify theoretically the relationship between the diminishing returns reported in (Shallue et al., 2019; Zhang et al., 2019) and batch size. Numerical evaluations in (Shallue et al., 2019; Zhang et al., 2019) have definitively shown that, for deep learning optimizers, the number of steps $K$ needed to train a deep neural network halves for each doubling of the batch size $b$ and that there is a region of diminishing returns beyond the *critical batch size $b^\star$*. This implies that there is a positive number $C$ such that

$$Kb \approx 2^C \text{ for } b \leq b^\star \text{ and } Kb \geq 2^C \text{ for } b \geq b^\star, \tag{1}$$

where $K$ and $b$ are defined for $i, j \in \mathbb{N}$ by $K = 2^i$ and $b = 2^j$ (For example, Figure 1 in Section 4 shows $C \approx 20$ and $b^\star \approx 2^{11}$ for Adam used to train ResNet-20 on the CIFAR-10 dataset). We define the *stochastic first-order oracle (SFO) complexity* of a deep learning optimizer as $N := Kb$ on the basis of the number of steps $K$ needed for training the deep neural network and the batch size $b$ used in the optimizer. Let $b^\star$ be a critical batch size such that there are diminishing returns for all batch sizes beyond $b^\star$, as asserted in (Shallue et al., 2019; Zhang et al., 2019). This fact, expressed in (1), implies that, while SFO complexity $N$ initially almost does not change (i.e., $K$ halves for each doubling of $b$), $N$ is minimized at critical batch size $b^\star$, and there are diminishing returns once the batch size exceeds $b^\star$.

## 1.3 CONTRIBUTION

Our results are summarized in Table 2 (see also the "(this paper)" row in Table 1). Our goal is to find a local minimizer of a loss function $f$ over $\mathbb{R}^d$, i.e., a stationary point $\boldsymbol{\theta}^\star \in \mathbb{R}^d$ satisfying $\nabla f(\boldsymbol{\theta}^\star) = \mathbf{0}$, which is equivalent to the variational inequality (VI) defined for all $\boldsymbol{\theta} \in \mathbb{R}^d$ by $\nabla f(\boldsymbol{\theta}^\star)^\top(\boldsymbol{\theta}^\star - \boldsymbol{\theta}) \leq 0$. Here, we show the relationship between (i) $\mathbb{E}[\nabla f(\boldsymbol{\theta}_k)^\top(\boldsymbol{\theta}_k - \boldsymbol{\theta})] \leq \epsilon$ ($\boldsymbol{\theta} \in \mathbb{R}^d, k \in \mathbb{N}$) and (ii) $\mathbb{E}[\|\nabla f(\boldsymbol{\theta}_k)\|^2] \leq \epsilon$ ($k \in \mathbb{N}$), where $\epsilon > 0$ is the precision. Let us assume that $(\boldsymbol{\theta}_k)_{k \in \mathbb{N}}$ is bounded. Suppose that (i) holds. Then, there exists a subsequence $(\boldsymbol{\theta}_{k_i})_{i \in \mathbb{N}}$ of $(\boldsymbol{\theta}_k)_{k \in \mathbb{N}}$ such that $(\boldsymbol{\theta}_{k_i})_{i \in \mathbb{N}}$ converges to $\boldsymbol{\theta}^*$. The continuity of $\nabla f$ thus implies that, for all $\boldsymbol{\theta} \in \mathbb{R}^d$, $\mathbb{E}[\nabla f(\boldsymbol{\theta}^*)^\top(\boldsymbol{\theta}^* - \boldsymbol{\theta})] \leq \epsilon$. Putting $\boldsymbol{\theta} := \boldsymbol{\theta}^* - \nabla f(\boldsymbol{\theta}^*)$ ensures that $\mathbb{E}[\|\nabla f(\boldsymbol{\theta}^*)\|^2] \leq \epsilon$. Suppose that (ii) holds. Then, the definition of the inner product and Jensen's inequality imply that, for all $\boldsymbol{\theta} \in \mathbb{R}^d$, $\mathbb{E}[\nabla f(\boldsymbol{\theta}_k)^\top(\boldsymbol{\theta}_k - \boldsymbol{\theta})] \leq \text{Dist}(\boldsymbol{\theta})\sqrt{\epsilon}$, where $\text{Dist}(\boldsymbol{\theta}) := \sup\{\|\boldsymbol{\theta}_k - \boldsymbol{\theta}\| : k \in \mathbb{N}\} < +\infty$. Therefore, it is adequate to use (i) and $\epsilon$-approximation VI$(K, \boldsymbol{\theta}) := \frac{1}{K}\sum_{k=1}^K \mathbb{E}[\nabla f(\boldsymbol{\theta}_k)^\top(\boldsymbol{\theta}_k - \boldsymbol{\theta})] \leq \epsilon$ (Table 2) as the performance measure of an optimizer.

### 1.3.1 ADVANTAGE OF SETTING A SMALL CONSTANT LEARNING RATE AND HYPERPARAMETERS CLOSE TO ONE

We can show that the upper bound $\frac{C_1}{K} + \frac{C_2}{b} + C_3$ of VI$(K, \boldsymbol{\theta})$ becomes small when $\alpha$ is small, $\beta_1$ and $\beta_2$ are close to 1, and $K$ is large. This implies that Momentum and Adam perform well when $\alpha$ is small and $\beta_1$ and $\beta_2$ are each set close to 1. Section 3.1 shows this result in detail.

Table 2: Relationship between batch size $b$ and the number of steps $K$ to achieve an $\epsilon$-approximation of an optimizer using a constant learning rate $\alpha$ and hyperparameters $\beta_1, \beta_2$. The critical batch size $b^\star$ minimizes SFO complexity $N$ ($G$, $\sigma^2$, $M$, and $v_*$ are positive constants, $D(\boldsymbol{\theta})$ is a positive real number depending on $\boldsymbol{\theta} \in \mathbb{R}^d$, and $h$ is monotone decreasing with respect to $\beta_1$)

| Optimizer | SGD | Momentum | Adam |
|---|---|---|---|
| $C_1$ | $\dfrac{\mathbb{E}[\|\boldsymbol{\theta}_1 - \boldsymbol{\theta}\|^2]}{2\alpha}$ | $\dfrac{\mathbb{E}[\|\boldsymbol{\theta}_1 - \boldsymbol{\theta}\|^2]}{2\alpha\beta_1}$ | $\dfrac{dD(\boldsymbol{\theta})\sqrt{M}}{2\alpha\beta_1\sqrt{1-\beta_2}}$ |
| $C_2$ | $\dfrac{\sigma^2\alpha}{2}$ | $\dfrac{\sigma^2\alpha}{2\beta_1}$ | $\dfrac{\sigma^2\alpha}{2\sqrt{v_*}\beta_1(1-\beta_1)}$ |
| $C_3$ | $\dfrac{G^2\alpha}{2}$ | $\dfrac{G^2\alpha}{2\beta_1} + h(\beta_1)$ | $\dfrac{G^2\alpha}{2\sqrt{v_*}\beta_1(1-\beta_1)} + h(\beta_1)$ |
| Upper Bound of VI | $\mathrm{VI}(K,\boldsymbol{\theta}) := \dfrac{1}{K}\sum_{k=1}^{K}\mathbb{E}\left[\nabla f(\boldsymbol{\theta}_k)^\top(\boldsymbol{\theta}_k - \boldsymbol{\theta})\right] \leq \dfrac{C_1}{K} + \dfrac{C_2}{b} + C_3 = \epsilon$ | | |
| Steps $K$ and SFO $N$ | $K = \dfrac{C_1 b}{(\epsilon - C_3)b - C_2}$ $\quad N = \dfrac{C_1 b^2}{(\epsilon - C_3)b - C_2}$ | | |
| Critical Batch $b^\star$ | $b^\star = \dfrac{2C_2}{\epsilon - C_3}$ | | |

### 1.3.2 CRITICAL BATCH SIZE

As described in Section 1.2.2, the practical performance of a deep learning optimizer strongly depends on the batch size (Shallue et al., 2019; Zhang et al., 2019). The advantage of this paper is to clarify *theoretically* the relationship between batch size and the performance of deep learning optimizers and develop a theory demonstrating the existence of critical batch sizes, which was shown numerically by (Shallue et al., 2019; Zhang et al., 2019). Motivated by the results in (Shallue et al., 2019; Zhang et al., 2019) and Section 1.2.2, we use SFO complexity $N := Kb$ as the performance measure of a deep learning optimizer. We first show that the number of steps $K$ to satisfy $\mathrm{VI}(K,\boldsymbol{\theta}) \leq \epsilon$ can be defined as in Table 2. As a function, $K$ is convex and monotone decreasing with respect to batch size $b$. Next, we show that SFO complexity $N$ defined as in Table 2 is convex with respect to batch size $b$. This result agrees with the fact of (1). Moreover, SFO complexity $N$ is minimized at $b^\star$ defined as in Table 2. This result guarantees the existence of the critical batch size $b^\star$. Section 3.2 shows the above results in detail. However, the accurate setting of the critical batch size $b^\star$ defined as in Table 2 would be difficult since $b^\star$ involves unknown parameters, such as $G$ and $D(\boldsymbol{\theta})$ (see Section 2.2.3). The advantage of our analysis is that we can estimate appropriate batch sizes using the formula for $b^\star$ before implementing deep learning optimizers. Section 4 will discuss estimation of appropriate batch sizes in detail.

## 2 NONCONVEX OPTIMIZATION AND DEEP LEARNING OPTIMIZERS

This section gives a nonconvex optimization problem in deep neural networks and optimizers for solving the problem under standard assumptions.

### 2.1 NONCONVEX OPTIMIZATION IN DEEP LEARNING

Let $\mathbb{R}^d$ be a $d$-dimensional Euclidean space with inner product $\langle \boldsymbol{x}, \boldsymbol{y} \rangle := \boldsymbol{x}^\top \boldsymbol{y}$ inducing the norm $\|\boldsymbol{x}\|$ and $\mathbb{N}$ be the set of nonnegative integers. Define $[n] := \{1, 2, \ldots, n\}$ for $n \geq 1$. Given a parameter $\boldsymbol{\theta} \in \mathbb{R}^d$ and a data point $z$ in a data domain $Z$, a machine learning model provides a prediction whose quality is measured by a differentiable nonconvex loss function $\ell(\boldsymbol{\theta}; z)$. We aim to minimize the expected loss defined for all $\boldsymbol{\theta} \in \mathbb{R}^d$ by $f(\boldsymbol{\theta}) = \mathbb{E}_{z \sim \mathcal{D}}[\ell(\boldsymbol{\theta}; z)] = \mathbb{E}[\ell_\xi(\boldsymbol{\theta})]$, where $\mathcal{D}$ is a probability distribution over $Z$, $\xi$ denotes a random variable with distribution function $P$, and $\mathbb{E}[\cdot]$ denotes the expectation taken with respect to $\xi$. A particularly interesting example of $f(\boldsymbol{\theta})$ is the empirical average loss defined for all $\boldsymbol{\theta} \in \mathbb{R}^d$ by $f(\boldsymbol{\theta}; S) = \frac{1}{n}\sum_{i \in [n]} \ell(\boldsymbol{\theta}; z_i) = \frac{1}{n}\sum_{i \in [n]} \ell_i(\boldsymbol{\theta})$, where $S = (z_1, z_2, \ldots, z_n)$ denotes the training set and $\ell_i(\cdot) := \ell(\cdot; z_i)$ denotes the loss function corresponding to the $i$-th training data $z_i$.

## 2.2 DEEP LEARNING OPTIMIZERS

### 2.2.1 CONDITIONS

We assume that a stochastic first-order oracle (SFO) exists such that, for a given $\boldsymbol{\theta} \in \mathbb{R}^d$, it returns a stochastic gradient $\mathsf{G}_\xi(\boldsymbol{\theta})$ of the function $f$, where a random variable $\xi$ is supported on $\Xi$ independently of $\boldsymbol{\theta}$. The following are standard conditions when considering a deep learning optimizer.

(C1) $f \colon \mathbb{R}^d \to \mathbb{R}$ is continuously differentiable.

(C2) Let $(\boldsymbol{\theta}_k)_{k \in \mathbb{N}} \subset \mathbb{R}^d$ be the sequence generated by a deep learning optimizer. For each iteration $k$, $\mathbb{E}_{\xi_k}[\mathsf{G}_{\xi_k}(\boldsymbol{\theta}_k)] = \nabla f(\boldsymbol{\theta}_k)$, where $\xi_0, \xi_1, \ldots$ are independent samples and the random variable $\xi_k$ is independent of $(\boldsymbol{\theta}_l)_{l=0}^k$. There exists a nonnegative constant $\sigma^2$ such that $\mathbb{E}_{\xi_k}[\|\mathsf{G}_{\xi_k}(\boldsymbol{\theta}_k) - \nabla f(\boldsymbol{\theta}_k)\|^2] \le \sigma^2$.

(C3) For each iteration $k$, the optimizer samples a batch $B_k$ of size $b$ independently of $k$ and estimates the full gradient $\nabla f$ as $\nabla f_{B_k}(\boldsymbol{\theta}_k) := \frac{1}{b} \sum_{i \in [b]} \mathsf{G}_{\xi_{k,i}}(\boldsymbol{\theta}_k)$, where $\xi_{k,i}$ is a random variable generated by the $i$-th sampling in the $k$-th iteration.

### 2.2.2 ADAM

Algorithm 1 is the Adam optimizer (Kingma & Ba, 2015) under (C1)–(C3). The symbol $\odot$ in step 6 is defined for all $\boldsymbol{x} = (x_i)_{i=1}^d \in \mathbb{R}^d$, $\boldsymbol{x} \odot \boldsymbol{x} := (x_i^2)_{i=1}^d \in \mathbb{R}^d$, and $\mathrm{diag}(x_i)$ in step 8 is a diagonal matrix with diagonal components $x_1, x_2, \ldots, x_d$.

---

**Algorithm 1** Adam (Kingma & Ba, 2015)

---

**Require:** $\alpha \in (0, +\infty)$, $b \in (0, +\infty)$, $\beta_1 \in (0, 1)$, $\beta_2 \in [0, 1)$
1: $k \leftarrow 0$, $\boldsymbol{\theta}_0 \in \mathbb{R}^d$, $\boldsymbol{m}_{-1} := \boldsymbol{0}$, $\boldsymbol{v}_{-1} := \boldsymbol{0}$
2: **loop**
3:      $\nabla f_{B_k}(\boldsymbol{\theta}_k) := \frac{1}{b} \sum_{i \in [b]} \mathsf{G}_{\xi_{k,i}}(\boldsymbol{\theta}_k)$
4:      $\boldsymbol{m}_k := \beta_1 \boldsymbol{m}_{k-1} + (1 - \beta_1) \nabla f_{B_k}(\boldsymbol{\theta}_k)$
5:      $\hat{\boldsymbol{m}}_k := (1 - \beta_1^{k+1})^{-1} \boldsymbol{m}_k$
6:      $\boldsymbol{v}_k := \beta_2 \boldsymbol{v}_{k-1} + (1 - \beta_2) \nabla f_{B_k}(\boldsymbol{\theta}_k) \odot \nabla f_{B_k}(\boldsymbol{\theta}_k)$
7:      $\hat{\boldsymbol{v}}_k := (1 - \beta_2^{k+1})^{-1} \boldsymbol{v}_k$
8:      $\mathsf{H}_k := \mathrm{diag}(\sqrt{\hat{v}_{k,i}})$
9:      $\boldsymbol{\theta}_{k+1} := \boldsymbol{\theta}_k - \alpha \mathsf{H}_k^{-1} \hat{\boldsymbol{m}}_k$
10:     $k \leftarrow k + 1$
11: **end loop**

---

The SGD optimizer under (C1)–(C3) is Algorithm 1 when $\beta_1 = 0$ and $\mathsf{H}_k$ is the identity matrix. The Momentum optimizer under (C1)–(C3) is defined for all $k \in \mathbb{N}$ by $\boldsymbol{\theta}_{k+1} := \boldsymbol{\theta}_k - \alpha \boldsymbol{m}_k$.

### 2.2.3 ASSUMPTIONS

We assume the following conditions that were used in (Kingma & Ba, 2015, Theorem 4.1):

(A1) There exist positive numbers $G$ and $B$ such that, for all $k \in \mathbb{N}$, $\|\nabla f(\boldsymbol{\theta}_k)\| \le G$ and $\|\nabla f_{B_k}(\boldsymbol{\theta}_k)\| \le B$ (see also Appendix A.5).

(A2) For all $\boldsymbol{\theta} \in \mathbb{R}^d$, there exists a positive number $\mathrm{Dist}(\boldsymbol{\theta})$ such that, for all $k \in \mathbb{N}$, $\|\boldsymbol{\theta}_k - \boldsymbol{\theta}\| \le \mathrm{Dist}(\boldsymbol{\theta})$.

Let $(g_{k,i}^2)_{i=1}^d := \nabla f_{B_k}(\boldsymbol{\theta}_k) \odot \nabla f_{B_k}(\boldsymbol{\theta}_k)$ $(k \in \mathbb{N})$. Assumption (A1) implies that $M := \sup\{\max_{i \in [d]} g_{k,i}^2 \colon k \in \mathbb{N}\} < +\infty$. Assumption (A2) implies that $D(\boldsymbol{\theta}) := \sup\{\max_{i \in [d]} (\theta_{k,i} - \theta_i)^2 \colon k \in \mathbb{N}\} < +\infty$. We define $v_* := \inf\{\min_{i \in [d]} v_{k,i} \colon k \in \mathbb{N}\}$. Theorem 3 in (Reddi et al., 2018) shows that there exists a stochastic convex optimization problem such that Adam using $\beta_1 < \sqrt{\beta_2}$ (e.g., $\beta_1 = 0.9$ and $\beta_2 = 0.999$) does not converge to the optimal solution. If for all $k \in \mathbb{N}$ and all $i \in [d]$, $\hat{v}_{k,i}$ in Adam satisfies

$$\hat{v}_{k+1,i} \ge \hat{v}_{k,i}, \tag{2}$$

then Adam with a decaying learning rate $\alpha_k = \mathcal{O}(\frac{1}{\sqrt{k}})$ and $\beta_1$ and $\beta_2$ satisfying $\beta_1 < \sqrt{\beta_2}$ can solve the stochastic convex optimization problem (Reddi et al., 2018, (2), Theorem 4). We thus assume condition (2) for Adam to guarantee the convergence of Adam.

## 3 OUR RESULTS

This section states our theoretical results (Theorem 3.1) in Table 2 and our contribution (Sections 3.1 and 3.2) in detail. The proof of Theorem 3.1 is given in Appendix A.3.

**Theorem 3.1.** *The sequence $(\boldsymbol{\theta}_k)_{k\in\mathbb{N}}$ generated by each of SGD, Momentum, and Adam with (2) under (C1)–(C3) and (A1) and (A2) satisfies the following:*

(i) [Upper bound of $\mathrm{VI}(K,\boldsymbol{\theta})$] *For all $K \geq 1$ and all $\boldsymbol{\theta} \in \mathbb{R}^d$,*

$$\mathrm{VI}(K,\boldsymbol{\theta}) := \frac{1}{K}\sum_{k=1}^{K}\mathbb{E}\left[\nabla f(\boldsymbol{\theta}_k)^{\top}(\boldsymbol{\theta}_k - \boldsymbol{\theta})\right] \leq \frac{C_1}{K} + \frac{C_2}{b} + C_3,$$

*where $C_i$ $(i = 1, 2, 3)$ for SGD are*

$$C_1 := \frac{\mathbb{E}[\|\boldsymbol{\theta}_1 - \boldsymbol{\theta}\|^2]}{2\alpha},\ C_2 := \frac{\sigma^2\alpha}{2},\ C_3 := \frac{G^2\alpha}{2},$$

*$C_i$ $(i = 1, 2, 3)$ for Momentum are*

$$C_1 := \frac{\mathbb{E}[\|\boldsymbol{\theta}_1 - \boldsymbol{\theta}\|^2]}{2\alpha\beta_1},\ C_2 := \frac{\sigma^2\alpha}{2\beta_1},$$

$$C_3 := \frac{G^2\alpha}{2\beta_1} + \mathrm{Dist}(\boldsymbol{\theta})\left\{\frac{G(1-\beta_1)}{\beta_1} + 2\sqrt{\sigma^2 + G^2}\left(\frac{1}{\beta_1} + 2(1-\beta_1)\right)\right\},$$

*$C_i$ $(i = 1, 2, 3)$ for Adam with (2) are*

$$C_1 := \frac{dD(\boldsymbol{\theta})\sqrt{M}}{2\alpha\beta_1\sqrt{1-\beta_2}},\ C_2 := \frac{\sigma^2\alpha}{2\sqrt{v_*}\beta_1(1-\beta_1)},$$

$$C_3 := \frac{G^2\alpha}{2\sqrt{v_*}\beta_1(1-\beta_1)} + \mathrm{Dist}(\boldsymbol{\theta})\left\{\frac{G(1-\beta_1)}{\beta_1} + 2\sqrt{\sigma^2 + G^2}\left(\frac{1}{\beta_1} + 2(1-\beta_1)\right)\right\},$$

*and the parameters are defined as in Section 2.2.*

(ii) [Steps to satisfy $\mathrm{VI}(K,\boldsymbol{\theta}) \leq \epsilon$] *The number of steps $K$ defined by*

$$K(b) = \frac{C_1 b}{(\epsilon - C_3)b - C_2} \tag{3}$$

*satisfies $\mathrm{VI}(K,\boldsymbol{\theta}) \leq \epsilon$ and the function $K(b)$ defined by (3) is convex and monotone decreasing with respect to batch size $b$ $(> \frac{C_2}{\epsilon-C_3} > 0)$ (see also Appendix A.6 for the condition $\epsilon - C_3 > 0$).*

(iii) [Minimization of SFO complexity] *The SFO complexity defined by*

$$N = K(b)b = \frac{C_1 b^2}{(\epsilon - C_3)b - C_2} \tag{4}$$

*is convex with respect to batch size $b$ $(> \frac{C_2}{\epsilon-C_3} > 0)$. The batch size*

$$b^{\star} := \frac{2C_2}{\epsilon - C_3} \tag{5}$$

*attains the minimum value $N^{\star} = K(b^{\star})b^{\star} = \frac{4C_1 C_2}{(\epsilon-C_3)^2}$ of $N$.*

The proof of Theorem 3.1(i) ensures that $C_i$ $(i = 1, 2, 3)$ for AMSGrad (Algorithm 1 with $\hat{\boldsymbol{m}}_k = \boldsymbol{m}_k$, $\hat{\boldsymbol{v}}_k = \boldsymbol{v}_k$, $\tilde{v}_{k,i} := \max\{\tilde{v}_{k-1,i}, v_{k,i}\}$ (i.e., $\tilde{v}_{k+1,i} \geq \tilde{v}_{k,i}$; see (2)), and $\mathsf{H}_k := \mathrm{diag}(\sqrt{\tilde{v}_{k,i}})$) are

$$C_1 := \frac{dD(\boldsymbol{\theta})\sqrt{M}}{2\alpha\beta_1},\ C_2 := \frac{\sigma^2\alpha}{2\sqrt{\tilde{v}_*}\beta_1},$$

$$C_3 := \frac{G^2\alpha}{2\sqrt{\tilde{v}_*}\beta_1} + \mathrm{Dist}(\boldsymbol{\theta})\left\{\frac{G(1-\beta_1)}{\beta_1} + 2\sqrt{\sigma^2 + G^2}\left(\frac{1}{\beta_1} + 2(1-\beta_1)\right)\right\}, \tag{6}$$

where $\tilde{\boldsymbol{v}}_{-1} := \mathbf{0}$ and $\tilde{\boldsymbol{v}}_* := \inf\{\min_{i\in[d]} \tilde{v}_{k,i} \colon k \in \mathbb{N}\}$ (see Appendix A.3.4).

We give a brief outline of the proof strategy of Theorem 3.1, with an emphasis on the main difficulty that has to be overcome in order not to assume Lipschitz smoothness of $f$ (i.e., $\nabla f$ is Lipschitz continuous with the Lipschitz constant $L$). First, we show that (C2) and (A1) imply that $(\mathbb{E}[\|\boldsymbol{m}_k\|])_{k\in\mathbb{N}}$ and $(\mathbb{E}[\|\mathbf{d}_k\|])_{k\in\mathbb{N}}$ are bounded, where $\mathbf{d}_k := -\mathsf{H}_k^{-1}\hat{\boldsymbol{m}}_k$. Since we do not assume Lipschitz smoothness of $f$, we cannot use the descent lemma, i.e., $f(\boldsymbol{y}) \leq f(\boldsymbol{x}) + \nabla f(\boldsymbol{x})^\top (\boldsymbol{y} - \boldsymbol{x}) + \frac{L}{2}\|\boldsymbol{y} - \boldsymbol{x}\|^2$ $(\boldsymbol{x}, \boldsymbol{y} \in \mathbb{R}^d)$. This is the main difficulty to prove Theorem 3.1. Almost all of the previous analyses of adaptive methods are based on the descent lemma, and hence, they can use the expectation of the squared norm of the full gradient $\mathbb{E}[\|\nabla f(\boldsymbol{\theta}_k)\|^2]$ as the performance measure. Accordingly, we must use other performance measures that are different from $\mathbb{E}[\|\nabla f(\boldsymbol{\theta}_k)\|^2]$. This paper uses the performance measure $\mathrm{VI}(K, \boldsymbol{\theta})$. We next show that $\sum_{k=1}^K \mathbb{E}[\boldsymbol{m}_{k-1}^\top(\boldsymbol{\theta}_k - \boldsymbol{\theta})] \leq a_K + b_K + c_K \leq C_1 + \frac{C_2 K}{b} + \tilde{C}_3 K$, where $C_1$ and $C_2$ are defined as in Theorem 3.1 and $\tilde{C}_3 > 0$. In particular, (2), (A1), and (A2) imply $a_K \leq C_1$, the boundedness condition of $(\mathbb{E}[\|\mathbf{d}_k\|])_{k\in\mathbb{N}}$ implies $b_K \leq \frac{C_2 K}{b}$, and (A2) and the Cauchy–Schwarz inequality imply $c_K \leq \tilde{C}_3 K$. The definition of $\boldsymbol{m}_k$, the Cauchy–Schwarz inequality, the triangle inequality, and Jensen's inequality imply Theorem 3.1(i). Theorem 3.1(i) and $\frac{C_1}{K} + \frac{C_2}{b} + C_3 = \epsilon$ lead to Theorem 3.1(ii) and (iii).

### 3.1 ADVANTAGE OF SETTING A SMALL CONSTANT LEARNING RATE AND HYPERPARAMETERS CLOSE TO ONE

We first show theoretical evidence that Adam using a small constant learning rate $\alpha$, $\beta_1$ and $\beta_2$ close to 1, and a large number of steps $K$ performs well. Theorem 3.1(i) indicates that the upper bound of $\mathrm{VI}(K, \boldsymbol{\theta})$ for Adam is

$$\mathrm{VI}(K, \boldsymbol{\theta}) \leq \frac{dD(\boldsymbol{\theta})\sqrt{M}}{2\alpha\beta_1\sqrt{1-\beta_2}K} + \frac{\alpha(\sigma^2 b^{-1} + G^2)}{2\sqrt{v_*}\beta_1(1-\beta_1)K}\sum_{k=1}^K \sqrt{1-\beta_2^{k+1}} + h(\beta_1), \qquad (7)$$

where $\beta_1 \in (0, 1)$, $\beta_2 \in [0, 1)$, and

$$h(\beta_1) := \mathrm{Dist}(\boldsymbol{\theta})\left\{\frac{G(1-\beta_1)}{\beta_1} + 2\sqrt{\sigma^2 + G^2}\left(\frac{1}{\beta_1} + 2(1-\beta_1)\right)\right\} \qquad (8)$$

(the strict evaluation (7) of Theorem 3.1(i) comes from (31) in Appendix A). Since the function $h(\beta_1)$ defined by (8) is monotone decreasing, using $\beta_1$ close to 1 makes $h(\beta_1)$ small. Since $\frac{1}{\beta_1(1-\beta_1)}$ is monotone increasing for $\beta_1 \geq 1/2$, using $\beta_1$ close to 1 makes $\frac{1}{\beta_1(1-\beta_1)}$ large. Hence, we need to set a small $\alpha$ to make $\frac{\alpha(\sigma^2 b^{-1} + G^2)}{2\sqrt{v_*}\beta_1(1-\beta_1)}$ small. The function $\sqrt{1-\beta_2^{k+1}}$ is monotone decreasing with respect to $\beta_2$, while using $\beta_2$ close to 1 makes $\frac{1}{\sqrt{1-\beta_2}}$ large. When $\beta_2$ close to 1 and a small learning rate $\alpha$ are used, we need to use a large number of steps $K$ to make $\frac{dD(\boldsymbol{\theta})\sqrt{M}}{2\alpha\beta_1\sqrt{1-\beta_2}K}$ small.

### 3.2 CRITICAL BATCH SIZE

Theorem 3.1(ii) indicates that the number of steps $K$ to satisfy $\mathrm{VI}(K, \boldsymbol{\theta}) \leq \epsilon$ can be expressed as (3). The function $K(b)$ defined by (3) is convex and monotone decreasing. Hence, the form of $K$ defined by (3) supports theoretically the relationship between $K$ and $b$ shown in (Shallue et al., 2019; Zhang et al., 2019) (see also Figures 1 and 3 in this paper). Theorem 3.1(iii) indicates that SFO complexity defined by (4) is convex with respect to batch size $b$. This result agrees with the fact of (1) (see also Figures 2 and 4 in this paper). Moreover, SFO complexity $N := Kb$ is minimized at $b^\star$ defined by (5); e.g., $b_{\mathrm{S}}^\star$ for SGD, $b_{\mathrm{M}}^\star$ for Momentum, and $b_{\mathrm{A}}^\star$ for Adam are respectively

$$b_{\mathrm{S}}^\star = \frac{\sigma_{\mathrm{S}}^2 \alpha}{\epsilon - C_{3,\mathrm{S}}}, \ b_{\mathrm{M}}^\star = \frac{\sigma_{\mathrm{M}}^2 \alpha}{\beta_1(\epsilon - C_{3,\mathrm{M}})}, \ b_{\mathrm{A}}^\star = \frac{\sigma_{\mathrm{A}}^2 \alpha}{\sqrt{v_*}\beta_1(1-\beta_1)(\epsilon - C_{3,\mathrm{A}})}, \qquad (9)$$

where $\sigma_{\mathrm{S}}$, $\sigma_{\mathrm{M}}$, and $\sigma_{\mathrm{A}}$ are positive constants depending on SGD, Momentum, and Adam and $C_3 = C_{3,\mathrm{S}}, C_{3,\mathrm{M}}, C_{3,\mathrm{A}}$ are positive constants defined as in Theorem 3.1. From (9), the lower bounds $b^*$ of $b_{\mathrm{S}}^\star, b_{\mathrm{M}}^\star$, and $b_{\mathrm{A}}^\star$ are respectively

$$b_{\mathrm{S}}^\star > b_{\mathrm{S}}^* := \frac{\sigma_{\mathrm{S}}^2 \alpha}{\epsilon}, \ b_{\mathrm{M}}^\star > b_{\mathrm{M}}^* := \frac{\sigma_{\mathrm{M}}^2 \alpha}{\beta_1 \epsilon}, \ b_{\mathrm{A}}^\star > b_{\mathrm{A}}^* := \frac{\sigma_{\mathrm{A}}^2 \alpha}{\sqrt{v_*}\beta_1(1-\beta_1)\epsilon}. \qquad (10)$$

## 4 NUMERICAL RESULTS

We evaluated the performances of SGD, Momentum, and Adam with different batch sizes. The metrics were the number of steps $K$ and the SFO complexity $N$ satisfying $f(\boldsymbol{\theta}_K) \leq 10^{-1}$, where $\boldsymbol{\theta}_K$ is generated for each of SGD, Momentum, and Adam using batch size $b$. The stopping condition was 200 epochs. The experimental environment consisted of two Intel(R) Xeon(R) Gold 6148 2.4-GHz CPUs with 20 cores each, a 16-GB NVIDIA Tesla V100 900-Gbps GPU, and the Red Hat Enterprise Linux 7.6 OS. The code was written in Python 3.8.2 using the NumPy 1.17.3 and PyTorch 1.3.0 packages. A constant learning rate $\alpha = 10^{-3}$ was commonly used. Momentum used $\beta_1 = 0.9$. Adam used $\beta_1 = 0.9$ and $\beta_2 = 0.999$ (Kingma & Ba, 2015).

Here, we use (10) and estimate appropriate batch sizes in the sense that SFO complexity is minimized. The definitions of $v_*$ and $v_{k,i}$ (see also (26) in Appendix A) imply that, for $k \in [K]$ and all $i \in [d]$, $v_* \leq v_{k,i} \leq \max_{k \in [K]} \max_{i \in [d]} g_{k,i}^2 =: g_{k^*,i^*}^2 \leq \sum_{i=1}^d g_{k^*,i}^2 = \|\nabla f_{B_{k^*}}(\boldsymbol{\theta}_{k^*})\|^2$. Condition (C2) implies that $\mathbb{E}[\|\nabla f_{B_k}(\boldsymbol{\theta}_k)\|^2] \leq \frac{\sigma^2}{b} + \mathbb{E}[\|\nabla f(\boldsymbol{\theta}_k)\|^2]$ (see also (14) in Appendix A). Conditions (C2) and (C3) imply that, if $b$ is large, then $\sigma$ is small. Hence, assuming that $\frac{\sigma^2}{b} \approx 0$ implies that $v_* \leq \|\nabla f(\boldsymbol{\theta}_{k^*})\|^2 = \|\frac{1}{n} \sum_{i \in [n]} \nabla \ell_i(\boldsymbol{\theta}_{k^*})\|^2 = \frac{1}{n^2} \|\sum_{i \in [n]} \nabla \ell_i(\boldsymbol{\theta}_{k^*})\|^2 =: \frac{1}{n^2}\|\boldsymbol{G}_{k^*}\|^2 = \frac{1}{n^2} \sum_{i \in [d]} G_{k^*,i}^2 \leq \frac{d}{n^2} \max_{i \in [d]} G_{k^*,i}^2$. Since deep learning optimizers can approximate stationary points of $f$, we assume that, for example, $G_{k^*,i} \approx \epsilon$. Then, (10) implies that

$$b_{\mathrm{S}}^* := \frac{\sigma_{\mathrm{S}}^2}{10^3 \epsilon}, \; b_{\mathrm{M}}^* := \frac{\sigma_{\mathrm{M}}^2}{9 \cdot 10^2 \epsilon}, \; b_{\mathrm{A}}^* := \frac{\sigma_{\mathrm{A}}^2}{9 \cdot 10\sqrt{v_* \epsilon}} > \frac{\sigma_{\mathrm{A}}^2 n}{9 \cdot 10\sqrt{d}\epsilon^2} =: b_{\mathrm{A}}^{**}. \tag{11}$$

Conditions (C2) and (C3) imply that, if $b$ is large, then $\sigma$ is small. If SGD and Momentum can exploit large batch sizes, then $\sigma_{\mathrm{S}}$ and $\sigma_{\mathrm{M}}$ are small. However, (11) implies that $b_{\mathrm{S}}^*$ and $b_{\mathrm{M}}^*$ must be small when $\sigma_{\mathrm{S}}$ and $\sigma_{\mathrm{M}}$ are small. Accordingly, SGD and Momentum would not be able to use large batch sizes. Meanwhile, from (11), we expect that Adam can exploit a large batch size $b_{\mathrm{A}}^\star > b_{\mathrm{A}}^* > b_{\mathrm{A}}^{**}$, for example, when $b_{\mathrm{A}}^{**} < 2^{11}$ (CIFAR-10; $n = 50000$, ResNet-20; $d \approx 1.1 \times 10^7$, $b_{\mathrm{A}}^{**} \approx 1600$) and $2^{10} < b_{\mathrm{A}}^{**} < 2^{11}$ (MNIST; $n = 60000$, ResNet-18; $d \approx 1.0 \times 10^7$, $b_{\mathrm{A}}^{**} \approx 2000$) (He et al., 2016; Leong et al., 2020), where $\sigma_{\mathrm{A}}^2 \approx 10^{-2}$ and $\epsilon \approx 10^{-3}$ are used.

### 4.1 RESNET-20 ON THE CIFAR-10 DATASET

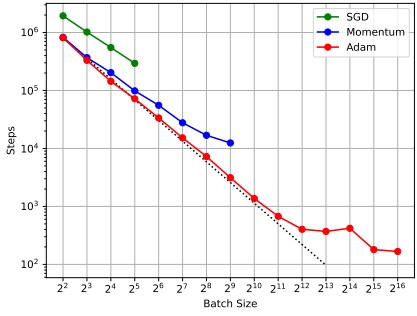
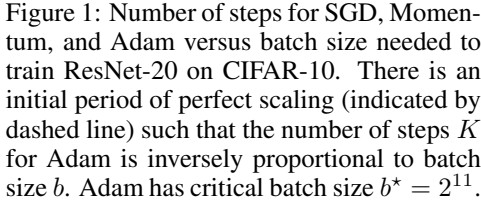

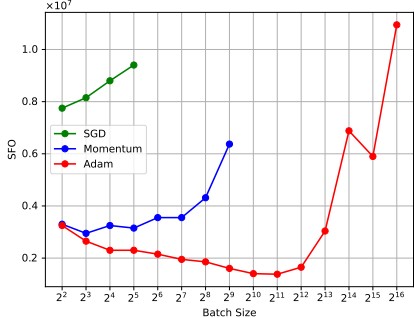

Figure 1: Number of steps for SGD, Momentum, and Adam versus batch size needed to train ResNet-20 on CIFAR-10. There is an initial period of perfect scaling (indicated by dashed line) such that the number of steps $K$ for Adam is inversely proportional to batch size $b$. Adam has critical batch size $b^\star = 2^{11}$.

Figure 2: SFO complexities for SGD, Momentum, and Adam versus batch size needed to train ResNet-20 on CIFAR-10. SFO complexity of Adam (resp. Momentum) is minimized at critical batch size $b^\star = 2^{11}$ (resp. $b^\star = 2^3$), whereas SFO complexity for SGD tends to increase with batch size.

Let us consider training ResNet-20 on the CIFAR-10 dataset with $n = 50000$. Figure 1 shows the number of steps for SGD, Momentum, and Adam versus batch size. For SGD and Momentum, the number of steps $K$ needed for $f(\boldsymbol{\theta}_K) \leq 10^{-1}$ initially decreased. However, SGD with $b \geq 2^6$ and

Momentum with $b \geq 2^{10}$ did not satisfy $f(\boldsymbol{\theta}_K) \leq 10^{-1}$ before the stopping condition was reached. Adam had an initial period of perfect scaling (indicated by dashed line) such that the number of steps $K$ needed for $f(\boldsymbol{\theta}_K) \leq 10^{-1}$ was inversely proportional to batch size $b$, and critical batch size $b^\star = 2^{11}$ such that $K$ was not inversely proportional to the batch size beyond $b^\star$; i.e., there were diminishing returns.

Figure 2 plots the SFO complexities for SGD, Momentum, and Adam versus batch size. For SGD, SFO complexity was minimum at $b^\star = 2^2$; for Momentum, it was minimum at $b^\star = 2^3$. This implies that SGD and Momentum could not use large batch sizes, as shown in (11). For Adam, SFO complexity was minimum at the critical batch size $b^\star = 2^{11}$ that was close to estimation of critical batch $b_A^\star > b_A^{**}$ with $1600 \approx b_A^{**} < b_A^\star = 2^{11} = 2048$, as shown in (11). We also checked that the elapsed time for Adam monotonically decreased for $b \leq 2^{11}$ and that the elapsed time for critical batch size $b^\star = 2^{11}$ was the shortest. The elapsed time for $b \geq 2^{12}$ increased with the SFO complexity, as shown in Figure 2 (see Tables 3, 4, 5, and 6 in Appendix A).

## 4.2 RESNET-18 ON THE MNIST DATASET

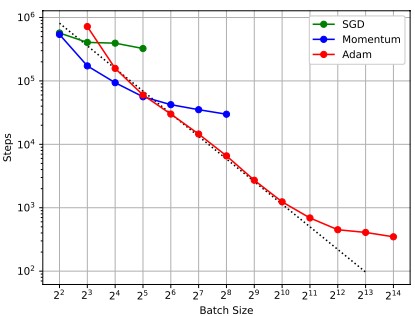

Figure 3: Number of steps for SGD, Momentum, and Adam versus batch size needed to train ResNet-18 on MNIST. There is an initial period of perfect scaling (indicated by dashed line) such that the number of steps $K$ for Adam is inversely proportional to batch size $b$. Adam has critical batch size $b^\star = 2^{10}$.

Figure 4: SFO complexities for SGD, Momentum, and Adam versus batch size needed to train ResNet-18 on MNIST. SFO complexity of Adam (resp. Momentum) is minimized at critical batch size $b^\star = 2^{10}$ (resp. $b^\star = 2^3$), whereas SFO complexity for SGD tends to increase with batch size.

Let us consider training ResNet-18 on the MNIST dataset with $n = 60000$. Figures 3 and 4 indicate that Adam could exploit larger batch sizes than SGD and Momentum. Moreover, these figures indicate that Momentum minimized SFO complexity at the critical batch size $b^\star = 2^3$ and Adam minimized SFO complexity at the critical batch size $b^\star = 2^{10}$ that was close to estimation of critical batch $b_A^\star > b_A^{**}$ with $2^{10} < b_A^{**} \approx 2000 < 2^{11}$, as shown in (11). We can also check that the elapsed time for critical batch size $b^\star = 2^{10}$ was the shortest (see Tables 7, 8, 9, and 10 in Appendix A).

We also estimated appropriate batch sizes of Adam for (i) CNN on MNIST ($n = 60000$, $d \approx 7.7 \times 10^6$) and (ii) ResNet-32 on CIFAR-10 ($n = 50000$, $d \approx 2.0 \times 10^7$) from $b_A^{**}$ in (11) and checked that actual critical batch sizes in (Zhang et al., 2019, Figure 5(a), (e)) are close to estimated batch sizes ((i) $b_A^\star = 2^{11} = 2048 > b_A^{**} \approx 2000$, (ii) $b_A^\star = 2^{10} = 1024 \approx b_A^{**} \approx 1200$).

## 5 CONCLUSION

This paper showed the relationship between batch size $b$ and the number of steps $K$ to achieve an $\epsilon$-approximation of deep learning optimizers using a small constant learning rate $\alpha$ and hyperparameters $\beta_1$ and $\beta_2$ close to 1. From the convexity of SFO complexity $N$, there exists a global minimizer $b^\star$ of $N$ that is the critical batch size. We also gave numerical results indicating that Adam using a small constant learning rate, hyperparameters close to one, and the critical batch size has faster convergence than Momentum and SGD. Moreover, we estimated appropriate batch sizes from our formula for $b^\star$ and showed that actual critical batch sizes are close to estimated batch sizes.

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

## A APPENDIX

Unless stated otherwise, all relationships between random variables are supposed to hold almost surely. Let $S$ be a positive definite matrix, which is denoted by $S \in \mathbb{S}_{++}^d$. The $S$-inner product of $\mathbb{R}^d$ is defined for all $\boldsymbol{x}, \boldsymbol{y} \in \mathbb{R}^d$ by $\langle \boldsymbol{x}, \boldsymbol{y} \rangle_S := \langle \boldsymbol{x}, S\boldsymbol{y} \rangle = \boldsymbol{x}^\top (S\boldsymbol{y})$, and the $S$-norm is defined by $\|\boldsymbol{x}\|_S := \sqrt{\langle \boldsymbol{x}, S\boldsymbol{x} \rangle}$.

## A.1 PREVIOUS RESULTS IN TABLE 1

This section provides the upper bounds of the performance measure of optimizers indicated in Table 1.

### A.1.1 TAIL-AVERAGED SGD

We consider the stochastic approximation problem of least square regression that is to minimize the expected square loss function $f(\boldsymbol{\theta}) := \frac{1}{2}\mathbb{E}_{(\boldsymbol{x},y)\sim\mathcal{D}}[y - \boldsymbol{x}^\top\boldsymbol{\theta}]$ (see Section 2.1 for the mathematical preliminaries). Tail-averaged SGD (Jain et al., 2018, Algorithm 1) with a constant learning rate $\alpha$ depending on the batch size $b$ and the largest eigenvalue of the Hessian of $f$ satisfies

$$\mathbb{E}[f(\bar{\boldsymbol{\theta}})] - f(\boldsymbol{\theta}^\star) \leq \frac{2(1 - \alpha\mu)^s}{\alpha^2\mu^2(\frac{n}{b} - s)^2}(f(\boldsymbol{\theta}_0) - f(\boldsymbol{\theta}^\star)) + \frac{4\sigma^2}{b(\frac{n}{b} - s)}, \tag{12}$$

where $s$ is the initial iterations, and $n$ is the total samples, $\mu > 0$ is the smallest eigenvalue of the Hessian of $f$, $(\boldsymbol{\theta}_i)_{i>s}$ is the sequence generated by Tail-averaged SGD, and $\bar{\boldsymbol{\theta}} := \frac{1}{\lfloor\frac{n}{b}\rfloor - s}\sum_{i>s}\boldsymbol{\theta}_i$ (Jain et al., 2018, Theorem 1). The upper bound of Tail-averaged SGD when $K := \frac{n}{b} - s$ is

$$\mathbb{E}[f(\bar{\boldsymbol{\theta}})] - f(\boldsymbol{\theta}^\star) = \mathcal{O}\left(\frac{1}{K^2} + \frac{1}{Kb}\right).$$

### A.1.2 ADAM

Theorem 1 in (Zaheer et al., 2018) and the proof of Theorem 1 in (Zaheer et al., 2018) show that, under the condition that $\nabla f$ is Lipschitz continuous with the Lipschitz constant $L$, Adam using $\alpha = \mathcal{O}(L^{-1})$, $\beta_1 = 0$, and $\beta_2 \geq 1 - \mathcal{O}(G^{-2})$ satisfies

$$\frac{1}{K}\sum_{k=1}^K\mathbb{E}\left[\|\nabla f(\boldsymbol{\theta}_k)\|^2\right] \leq 2\left(\sqrt{\beta_2}G + \epsilon\right)\left\{\frac{f(\boldsymbol{\theta}_1) - f(\boldsymbol{\theta}^\star)}{\alpha K} + \left(\frac{G\sqrt{1 - \beta_2}}{\epsilon^2} + \frac{L\alpha}{2\epsilon^2}\right)\frac{\sigma^2}{b}\right\},$$

that is, the upper bound of Adam with $\beta_1 = 0$ is

$$\frac{1}{K}\sum_{k=1}^K\mathbb{E}\left[\|\nabla f(\boldsymbol{\theta}_k)\|^2\right] = \mathcal{O}\left(\frac{1}{K} + \frac{1}{b}\right).$$

### A.1.3 GENERIC ADAM

Theorem 4 in (Zou et al., 2019) shows that, under the condition that $\nabla f$ is Lipschitz continuous with the Lipschitz constant $L$, Generic Adam using $\alpha_k = \mathcal{O}(\frac{1}{\sqrt{k}})$, $\beta_1 \in (0, 1)$, and $\beta_{2k} = 1 - \frac{\alpha}{k+1}$ satisfies

$$\mathbb{E}\left[\|\nabla f(\boldsymbol{\theta}_\tau)\|^{\frac{4}{3}}\right]^{\frac{3}{2}} \leq \frac{D + D'\sum_{k=1}^K\alpha_k\sqrt{1 - \beta_{2k}}}{\alpha_K K},$$

where $\tau$ is randomly chosen from $\{1, 2, \ldots, K\}$, $\alpha > 0$, $\gamma \in (0, 1)$, $\boldsymbol{v}_0 = (v_{0,i})_{i=1}^d$, $D_0, D_1 > 0$,

$$D' := \frac{2D_0^2D_3d\sqrt{B^4 + v_{0,1}d}}{(1 - \beta_1)\beta_{2,1}}, \quad D := \frac{2D_0\sqrt{B^4 + v_{0,1}d}}{1 - \beta_1}\left\{D_4 + D_3D_0d\alpha\log\left(1 + \frac{B^4}{v_{0,1}d}\right)\right\},$$

$$D_3 := \frac{D_0}{\sqrt{D_1}(1 - \sqrt{\gamma})}\left\{\frac{D_0^2\alpha L}{D_1(1 - \sqrt{\gamma})^2} + 2\left(\frac{\beta_1}{(1 - \beta_1)\sqrt{D_1(1 - \gamma)\beta_{2,1}}} + 1\right)^2B^2\right\},$$

$$D_4 := f(\boldsymbol{\theta}_1) - f(\boldsymbol{\theta}^\star).$$

This implies that the upper bound of Generic Adam is

$$\mathbb{E}\left[\|\nabla f(\boldsymbol{\theta}_\tau)\|^{\frac{4}{3}}\right]^{\frac{3}{2}} = \mathcal{O}\left(\frac{\log K}{\sqrt{K}}\right)$$

(see also Corollary 10 in (Zou et al., 2019) when $s = \frac{1}{2}$ and $r = 1$).

### A.1.4 ADAFOM

Corollary 3.2 in (Chen et al., 2019) shows that, under the condition that $\nabla f$ is Lipschitz continuous with the Lipschitz constant $L$, the AdaGrad with First Order Momentum (AdaFom) algorithm using $\alpha_k = \frac{1}{\sqrt{k}}$ and $\beta_1 \in (0, 1)$ satisfies

$$\frac{1}{K} \sum_{k=1}^{K} \mathbb{E}\left[\|\nabla f(\boldsymbol{\theta}_k)\|^2\right] \leq \frac{Q_1 + Q_2 \log K}{\sqrt{K}},$$

where $\bar{G} > 0$, $c > 0$,

$$Q_1 := G\left\{D_1 d(1 - \log(c^2) + 2\log G) + \frac{D_2 d}{c} + \frac{D_3 d}{c^2} + D_4\right\}, \ Q_2 := GD_1 d,$$

$$D_1 := \frac{3}{2}L + \frac{1}{2} + \frac{L^2 \beta_1}{1 - \beta_1}\left(\frac{1}{1 - \beta_1}\right)^2, \ D_2 := G^2\left(\frac{\beta_1}{1 - \beta_1} + 2\right),$$

$$D_3 := G^2\left\{1 + \left(\frac{L}{1 - \beta_1}\right)^2 \frac{\beta_1}{1 - \beta_1}\right\}\left(\frac{\beta_1}{1 - \beta_1}\right)^2,$$

$$D_4 := \left(\frac{\beta_1}{1 - \beta_1}\right)(G^2 + \bar{G}^2) + \left(\frac{\beta_1}{1 - \beta_1}\right)^2 \bar{G}^2 + 2\alpha_1 G^2 \mathbb{E}\left[\left\|\frac{1}{\sqrt{\hat{\boldsymbol{v}}_1}}\right\|_1\right] + \mathbb{E}[f(\boldsymbol{\theta}_1) - f(\boldsymbol{\theta}^\star)].$$

### A.1.5 AMSGRAD

Theorem 3 in (Zhou et al., 2020) shows that, under the condition that $\nabla f$ is Lipschitz continuous with the Lipschitz constant $L$, AMSGrad using a constant learning rate $\alpha$ and $\beta_1$ and $\beta_2$ with $\beta_1 < \sqrt{\beta_2}$ satisfies

$$\frac{1}{K} \sum_{k=1}^{K} \mathbb{E}\left[\|\nabla f(\boldsymbol{\theta}_k)\|^2\right] \leq \frac{D_1}{\alpha K} + \frac{D_2 d}{K} + \frac{\alpha D_3 d}{K^{\frac{1}{2} - s}},$$

where $s \in (0, \frac{1}{2})$, $G_\infty > 0$, $\gamma > 0$,

$$D_1 := 2G_\infty(f(\boldsymbol{\theta}_1) - f(\boldsymbol{\theta}^\star)), \ D_2 := \frac{2G_\infty^3}{\gamma(1 - \beta_1)}, \ D_3 := \frac{2LG_\infty^2}{\gamma\sqrt{1 - \beta_2}(1 - \frac{\beta_1}{\sqrt{\beta_2}})}\left(1 + \frac{2\beta_1^2}{1 - \beta_1}\right).$$

Hence, the upper bound of AMSGrad is

$$\frac{1}{K} \sum_{k=1}^{K} \mathbb{E}\left[\|\nabla f(\boldsymbol{\theta}_k)\|^2\right] = \mathcal{O}\left(\frac{1}{K^{\frac{1}{2} - s}}\right).$$

### A.1.6 ADABELIEF

Theorem 2.2 in (Zhuang et al., 2020) shows that, under the condition that $\nabla f$ is Lipschitz continuous with the Lipschitz constant $L$, AdaBelief using $\alpha_k = \frac{\alpha}{\sqrt{k}}$, where $\alpha > 0$, and $\beta_1, \beta_2 \in (0, 1)$ satisfies

$$\frac{1}{K} \sum_{k=1}^{K} \mathbb{E}\left[\|\nabla f(\boldsymbol{\theta}_k)\|^2\right] \leq \frac{Q_1 + Q_2 \log K}{\alpha\sqrt{K}},$$

where $c > 0$, $\bar{G} > 0$,

$$Q_1 := G\left\{\frac{D_1 \alpha^2(G^2 + \sigma^2)}{c} + \frac{D_2 d\alpha}{\sqrt{c}} + \frac{D_3 d\alpha^2}{c} + D_4\right\}, \ Q_2 := \frac{GD_1 \alpha^2(G^2 + \sigma^2)}{c},$$

$$D_1 := \frac{3}{2}L + \frac{1}{2} + \frac{L^2 \beta_1}{1 - \beta_1}\left(\frac{1}{1 - \beta_1}\right)^2, \ D_2 := G^2\left(\frac{\beta_1}{1 - \beta_1} + 2\right),$$

$$D_3 := G^2\left\{1 + \left(\frac{L}{1 - \beta_1}\right)^2 \frac{\beta_1}{1 - \beta_1}\right\}\left(\frac{\beta_1}{1 - \beta_1}\right)^2,$$

$$D_4 := \left(\frac{\beta_1}{1 - \beta_1}\right)(G^2 + \bar{G}^2) + \left(\frac{\beta_1}{1 - \beta_1}\right)^2 \bar{G}^2 + 2\alpha_1 G^2 \mathbb{E}\left[\left\|\frac{1}{\sqrt{\hat{\boldsymbol{v}}_1}}\right\|_1\right] + \mathbb{E}[f(\boldsymbol{\theta}_1) - f(\boldsymbol{\theta}^\star)].$$

### A.1.7 PADAM

Corollary 4.5 in (Chen et al., 2021) shows that, under the condition that $\nabla f$ is Lipschitz continuous with the Lipschitz constant $L$, the Partially adaptive momentum estimation (Padam) method using a constant learning rate $\alpha$, $p \in [0, \frac{1}{4}]$, and $\beta_1$ and $\beta_2$ with $\beta_1 < \beta_2^{2p}$ satisfies

$$\frac{1}{K} \sum_{k=1}^{K} \mathbb{E}\left[\|\nabla f(\boldsymbol{\theta}_k)\|^2\right] \leq \frac{D_1}{\alpha K} + \frac{D_2 d}{K} + \frac{\alpha D_3 d G_\infty}{K^{\frac{1}{2}-s}},$$

where $s \in (0, \frac{1}{2})$, $G_\infty > 0$,

$$D_1 := 2G_\infty^{2p}(f(\boldsymbol{\theta}_1) - f(\boldsymbol{\theta}^\star)), \ D_2 := \frac{4G_\infty^{2+2p}\mathbb{E}[\|\hat{\boldsymbol{v}}_1^{-p}\|_1]}{d(1-\beta_1)} + 4G_\infty^2,$$

$$D_3 := \frac{4LG_\infty^{1-2p}}{(1-\beta_2)^{2p}} + \frac{8LG_\infty^{1-2p}(1-\beta_1)}{(1-\beta_2)^{2p}(1-\frac{\beta_1}{\beta_2^{2p}})}\left(\frac{\beta_1}{1-\beta_1}\right)^2.$$

Hence, the upper bound of Padam is

$$\frac{1}{K} \sum_{k=1}^{K} \mathbb{E}\left[\|\nabla f(\boldsymbol{\theta}_k)\|^2\right] = \mathcal{O}\left(\frac{1}{K^{\frac{1}{2}-s}}\right).$$

## A.2 LEMMAS

**Lemma A.1.** *Suppose that (C1), (C2), and (C3) hold. Then, Adam satisfies the following: for all $k \in \mathbb{N}$ and all $\boldsymbol{\theta} \in \mathbb{R}^d$,*

$$\mathbb{E}\left[\|\boldsymbol{\theta}_{k+1} - \boldsymbol{\theta}\|_{\mathsf{H}_k}^2\right] = \mathbb{E}\left[\|\boldsymbol{\theta}_k - \boldsymbol{\theta}\|_{\mathsf{H}_k}^2\right] + \alpha^2 \mathbb{E}\left[\|\mathbf{d}_k\|_{\mathsf{H}_k}^2\right]$$

$$+ 2\alpha \left\{\frac{\beta_1}{\tilde{\beta}_{1k}}\mathbb{E}\left[(\boldsymbol{\theta} - \boldsymbol{\theta}_k)^\top \boldsymbol{m}_{k-1}\right] + \frac{\hat{\beta}_1}{\tilde{\beta}_{1k}}\mathbb{E}\left[(\boldsymbol{\theta} - \boldsymbol{\theta}_k)^\top \nabla f(\boldsymbol{\theta}_k)\right]\right\},$$

*where $\mathbf{d}_k := -\mathsf{H}_k^{-1}\hat{\boldsymbol{m}}_k$, $\hat{\beta}_1 := 1 - \beta_1$, and $\tilde{\beta}_{1k} := 1 - \beta_1^{k+1}$.*

*Proof.* Let $\boldsymbol{\theta} \in \mathbb{R}^d$ and $k \in \mathbb{N}$. The definition $\boldsymbol{\theta}_{k+1} := \boldsymbol{\theta}_k + \alpha \mathbf{d}_k$ implies that

$$\|\boldsymbol{\theta}_{k+1} - \boldsymbol{\theta}\|_{\mathsf{H}_k}^2 = \|\boldsymbol{\theta}_k - \boldsymbol{\theta}\|_{\mathsf{H}_k}^2 + 2\alpha \langle \boldsymbol{\theta}_k - \boldsymbol{\theta}, \mathbf{d}_k \rangle_{\mathsf{H}_k} + \alpha^2 \|\mathbf{d}_k\|_{\mathsf{H}_k}^2.$$

Moreover, the definitions of $\mathbf{d}_k$, $\boldsymbol{m}_k$, and $\hat{\boldsymbol{m}}_k$ ensure that

$$\langle \boldsymbol{\theta}_k - \boldsymbol{\theta}, \mathbf{d}_k \rangle_{\mathsf{H}_k} = \langle \boldsymbol{\theta}_k - \boldsymbol{\theta}, \mathsf{H}_k \mathbf{d}_k \rangle = \langle \boldsymbol{\theta} - \boldsymbol{\theta}_k, \hat{\boldsymbol{m}}_k \rangle = \frac{1}{\tilde{\beta}_{1k}}(\boldsymbol{\theta} - \boldsymbol{\theta}_k)^\top \boldsymbol{m}_k$$

$$= \frac{\beta_1}{\tilde{\beta}_{1k}}(\boldsymbol{\theta} - \boldsymbol{\theta}_k)^\top \boldsymbol{m}_{k-1} + \frac{\hat{\beta}_1}{\tilde{\beta}_{1k}}(\boldsymbol{\theta} - \boldsymbol{\theta}_k)^\top \nabla f_{B_k}(\boldsymbol{\theta}_k).$$

Hence,

$$\|\boldsymbol{\theta}_{k+1} - \boldsymbol{\theta}\|_{\mathsf{H}_k}^2 = \|\boldsymbol{\theta}_k - \boldsymbol{\theta}\|_{\mathsf{H}_k}^2 + \alpha^2 \|\mathbf{d}_k\|_{\mathsf{H}_k}^2$$

$$+ 2\alpha \left\{\frac{\beta_1}{\tilde{\beta}_{1k}}(\boldsymbol{\theta} - \boldsymbol{\theta}_k)^\top \boldsymbol{m}_{k-1} + \frac{\hat{\beta}_1}{\tilde{\beta}_{1k}}(\boldsymbol{\theta} - \boldsymbol{\theta}_k)^\top \nabla f_{B_k}(\boldsymbol{\theta}_k)\right\}. \tag{13}$$

Conditions (C2) and (C3) guarantee that

$$\mathbb{E}\left[\mathbb{E}\left[(\boldsymbol{\theta} - \boldsymbol{\theta}_k)^\top \nabla f_{B_k}(\boldsymbol{\theta}_k)\Big|\boldsymbol{\theta}_k\right]\right] = \mathbb{E}\left[(\boldsymbol{\theta} - \boldsymbol{\theta}_k)^\top \mathbb{E}\left[\nabla f_{B_k}(\boldsymbol{\theta}_k)\Big|\boldsymbol{\theta}_k\right]\right] = \mathbb{E}\left[(\boldsymbol{\theta} - \boldsymbol{\theta}_k)^\top \nabla f(\boldsymbol{\theta}_k)\right].$$

Therefore, the lemma follows from taking the expectation on both sides of (13). This completes the proof. $\square$

The discussion in the proof of Lemma A.1 also gives the following lemma.

**Lemma A.2.** *Suppose that (C1), (C2), and (C3) hold. Then, SGD satisfies the following: for all $k \in \mathbb{N}$ and all $\boldsymbol{\theta} \in \mathbb{R}^d$,*

$$\mathbb{E}\left[\|\boldsymbol{\theta}_{k+1} - \boldsymbol{\theta}\|^2\right] = \mathbb{E}\left[\|\boldsymbol{\theta}_k - \boldsymbol{\theta}\|^2\right] + \alpha^2 \mathbb{E}\left[\|\nabla f_{B_k}(\boldsymbol{\theta}_k)\|^2\right] + 2\alpha \mathbb{E}\left[(\boldsymbol{\theta} - \boldsymbol{\theta}_k)^\top \nabla f(\boldsymbol{\theta}_k)\right].$$

*Moreover, Momentum satisfies the following: for all $k \in \mathbb{N}$ and all $\boldsymbol{\theta} \in \mathbb{R}^d$,*

$$\mathbb{E}\left[\|\boldsymbol{\theta}_{k+1} - \boldsymbol{\theta}\|^2\right] = \mathbb{E}\left[\|\boldsymbol{\theta}_k - \boldsymbol{\theta}\|^2\right] + \alpha^2 \mathbb{E}\left[\|\boldsymbol{m}_k\|^2\right]$$
$$+ 2\alpha \left\{\beta_1 \mathbb{E}\left[(\boldsymbol{\theta} - \boldsymbol{\theta}_k)^\top \boldsymbol{m}_{k-1}\right] + \hat{\beta}_1 \mathbb{E}\left[(\boldsymbol{\theta} - \boldsymbol{\theta}_k)^\top \nabla f(\boldsymbol{\theta}_k)\right]\right\},$$

*where $\hat{\beta}_1 := 1 - \beta_1$.*

The following lemma indicates the bounds on $(\mathbb{E}[\|\boldsymbol{m}_k\|^2])_{k \in \mathbb{N}}$ and $(\mathbb{E}[\|\mathbf{d}_k\|_{\mathsf{H}_k}^2])_{k \in \mathbb{N}}$.

**Lemma A.3.** *Adam under (C2) and (A1), for all $k \in \mathbb{N}$ satisfies*

$$\mathbb{E}\left[\|\boldsymbol{m}_k\|^2\right] \leq \frac{\sigma^2}{b} + G^2, \quad \mathbb{E}\left[\|\mathbf{d}_k\|_{\mathsf{H}_k}^2\right] \leq \frac{\sqrt{\tilde{\beta}_{2k}}}{\tilde{\beta}_{1k}\sqrt{v_*}}\left(\frac{\sigma^2}{b} + G^2\right),$$

*where $v_* := \inf\{\min_{i \in [d]} v_{k,i}\colon k \in \mathbb{N}\}$, $\tilde{\beta}_{1k} := 1 - \beta_{1k}^{k+1}$, and $\tilde{\beta}_{2k} := 1 - \beta_{2k}^{k+1}$.*

*Proof.* Condition (C2) implies that

$$\begin{aligned}
\mathbb{E}\left[\|\nabla f_{B_k}(\boldsymbol{\theta}_k)\|^2 \Big| \boldsymbol{\theta}_k\right] &= \mathbb{E}\left[\|\nabla f_{B_k}(\boldsymbol{\theta}_k) - \nabla f(\boldsymbol{\theta}_k) + \nabla f(\boldsymbol{\theta}_k)\|^2 \Big| \boldsymbol{\theta}_k\right] \\
&= \mathbb{E}\left[\|\nabla f_{B_k}(\boldsymbol{\theta}_k) - \nabla f(\boldsymbol{\theta}_k)\|^2 \Big| \boldsymbol{\theta}_k\right] + \mathbb{E}\left[\|\nabla f(\boldsymbol{\theta}_k)\|^2 \Big| \boldsymbol{\theta}_k\right] \\
&\quad + 2\mathbb{E}\left[(\nabla f_{B_k}(\boldsymbol{\theta}_k) - \nabla f(\boldsymbol{\theta}_k))^\top \nabla f(\boldsymbol{\theta}_k) \Big| \boldsymbol{\theta}_k\right] \\
&= \mathbb{E}\left[\|\nabla f_{B_k}(\boldsymbol{\theta}_k) - \nabla f(\boldsymbol{\theta}_k)\|^2 \Big| \boldsymbol{\theta}_k\right] + \|\nabla f(\boldsymbol{\theta}_k)\|^2,
\end{aligned} \tag{14}$$

which, together with (C2) and (A1), in turn implies that

$$\mathbb{E}\left[\|\nabla f_{B_k}(\boldsymbol{\theta}_k)\|^2\right] \leq \frac{\sigma^2}{b} + G^2. \tag{15}$$

The convexity of $\|\cdot\|^2$, together with the definition of $\boldsymbol{m}_k$ and (15), guarantees that, for all $k \in \mathbb{N}$,

$$\begin{aligned}
\mathbb{E}\left[\|\boldsymbol{m}_k\|^2\right] &\leq \beta_1 \mathbb{E}\left[\|\boldsymbol{m}_{k-1}\|^2\right] + \hat{\beta}_1 \mathbb{E}\left[\|\nabla f_{B_k}(\boldsymbol{\theta}_k)\|^2\right] \\
&\leq \beta_1 \mathbb{E}\left[\|\boldsymbol{m}_{k-1}\|^2\right] + \hat{\beta}_1\left(\frac{\sigma^2}{b} + G^2\right).
\end{aligned}$$

Induction thus ensures that, for all $k \in \mathbb{N}$,

$$\mathbb{E}\left[\|\boldsymbol{m}_k\|^2\right] \leq \max\left\{\|\boldsymbol{m}_{-1}\|^2, \frac{\sigma^2}{b} + G^2\right\} = \frac{\sigma^2}{b} + G^2, \tag{16}$$

where $\boldsymbol{m}_{-1} = \boldsymbol{0}$. For $k \in \mathbb{N}$, $\mathsf{H}_k \in \mathbb{S}_{++}^d$ guarantees the existence of a unique matrix $\overline{\mathsf{H}}_k \in \mathbb{S}_{++}^d$ such that $\mathsf{H}_k = \overline{\mathsf{H}}_k^2$ (Horn & Johnson, 1985, Theorem 7.2.6). We have that, for all $\boldsymbol{x} \in \mathbb{R}^d$, $\|\boldsymbol{x}\|_{\mathsf{H}_k}^2 = \|\overline{\mathsf{H}}_k \boldsymbol{x}\|^2$. Accordingly, the definitions of $\mathbf{d}_k$ and $\hat{\boldsymbol{m}}_k$ imply that, for all $k \in \mathbb{N}$,

$$\mathbb{E}\left[\|\mathbf{d}_k\|_{\mathsf{H}_k}^2\right] = \mathbb{E}\left[\left\|\overline{\mathsf{H}}_k^{-1}\mathsf{H}_k \mathbf{d}_k\right\|^2\right] \leq \frac{1}{\tilde{\beta}_{1k}^2}\mathbb{E}\left[\left\|\overline{\mathsf{H}}_k^{-1}\right\|^2 \|\boldsymbol{m}_k\|^2\right],$$

where

$$\left\|\overline{\mathsf{H}}_k^{-1}\right\| = \left\|\mathrm{diag}\left(\hat{v}_{k,i}^{-\frac{1}{4}}\right)\right\| = \max_{i \in [d]} \hat{v}_{k,i}^{-\frac{1}{4}} = \max_{i \in [d]}\left(\frac{v_{k,i}}{\tilde{\beta}_{2k}}\right)^{-\frac{1}{4}} =: \left(\frac{v_{k,i^*}}{\tilde{\beta}_{2k}}\right)^{-\frac{1}{4}}.$$

Moreover, the definition of

$$v_* := \inf\left\{v_{k,i^*}\colon k \in \mathbb{N}\right\}$$

and (16) imply that, for all $k \in \mathbb{N}$,

$$\mathbb{E}\left[\|\mathbf{d}_k\|_{\mathsf{H}_k}^2\right] \leq \frac{\tilde{\beta}_{2k}^{\frac{1}{2}}}{\tilde{\beta}_{1k}^2 v_*^{\frac{1}{2}}}\left(\frac{\sigma^2}{b} + G^2\right),$$

completing the proof. $\qquad\square$

### A.3 PROOF OF THEOREM 3.1

#### A.3.1 SGD

We show Theorem 3.1 for SGD.

*Proof.* (i) Lemma A.2 and (15) imply that, for all $k \in \mathbb{N}$ and all $\boldsymbol{\theta} \in \mathbb{R}^d$,

$$2\alpha \mathbb{E}\left[(\boldsymbol{\theta}_k - \boldsymbol{\theta})^\top \nabla f(\boldsymbol{\theta}_k)\right] = \mathbb{E}\left[\|\boldsymbol{\theta}_k - \boldsymbol{\theta}\|^2\right] - \mathbb{E}\left[\|\boldsymbol{\theta}_{k+1} - \boldsymbol{\theta}\|^2\right] + \alpha^2 \mathbb{E}\left[\|\nabla f_{B_k}(\boldsymbol{\theta}_k)\|^2\right]$$

$$\leq \mathbb{E}\left[\|\boldsymbol{\theta}_k - \boldsymbol{\theta}\|^2\right] - \mathbb{E}\left[\|\boldsymbol{\theta}_{k+1} - \boldsymbol{\theta}\|^2\right] + \alpha^2 \left(\frac{\sigma^2}{b} + G^2\right).$$

Summing the above inequality from $k = 1$ to $k = K$ leads to the finding that, for all $K \geq 1$,

$$2\alpha \sum_{k=1}^{K} \mathbb{E}\left[(\boldsymbol{\theta}_k - \boldsymbol{\theta})^\top \nabla f(\boldsymbol{\theta}_k)\right] \leq \mathbb{E}\left[\|\boldsymbol{\theta}_1 - \boldsymbol{\theta}\|^2\right] - \mathbb{E}\left[\|\boldsymbol{\theta}_{K+1} - \boldsymbol{\theta}\|^2\right] + \alpha^2 \left(\frac{\sigma^2}{b} + G^2\right) K,$$

which implies that, for all $K \geq 1$ and all $\boldsymbol{\theta} \in \mathbb{R}^d$,

$$\mathrm{VI}(K, \boldsymbol{\theta}) := \frac{1}{K} \sum_{k=1}^{K} \mathbb{E}\left[(\boldsymbol{\theta}_k - \boldsymbol{\theta})^\top \nabla f(\boldsymbol{\theta}_k)\right] \leq \frac{\mathbb{E}\left[\|\boldsymbol{\theta}_1 - \boldsymbol{\theta}\|^2\right]}{2\alpha K} + \frac{\alpha}{2}\left(\frac{\sigma^2}{b} + G^2\right)$$

$$= \underbrace{\frac{\mathbb{E}\left[\|\boldsymbol{\theta}_1 - \boldsymbol{\theta}\|^2\right]}{2\alpha}}_{C_1} \frac{1}{K} + \underbrace{\frac{\sigma^2 \alpha}{2}}_{C_2} \frac{1}{b} + \underbrace{\frac{G^2 \alpha}{2}}_{C_3}. \tag{17}$$

(ii) Let $\boldsymbol{\theta} \in \mathbb{R}^d$ and $\epsilon > 0$. Condition $\frac{C_1}{K} + \frac{C_2}{b} + C_3 = \epsilon$ is equivalent to

$$K = K(b) = \frac{C_1 b}{(\epsilon - C_3) b - C_2}. \tag{18}$$

Since $\epsilon = \frac{C_1}{K} + \frac{C_2}{b} + C_3 > C_3$, we consider the case $b > \frac{C_2}{\epsilon - C_3} > 0$ to guarantee that $K > 0$. From (17), the function $K$ defined by (18) satisfies $\mathrm{VI}(K, \boldsymbol{\theta}) \leq \frac{C_1}{K} + \frac{C_2}{b} + C_3 = \epsilon$. Moreover, from (18),

$$\frac{\mathrm{d}K(b)}{\mathrm{d}b} = \frac{-C_1 C_2}{\{(\epsilon - C_3) b - C_2\}^2} \leq 0, \quad \frac{\mathrm{d}^2 K(b)}{\mathrm{d}b^2} = \frac{2C_1 C_2 (\epsilon - C_3)}{\{(\epsilon - C_3) b - C_2\}^3} \geq 0,$$

which implies that $K$ is convex and monotone decreasing with respect to $b$.

(iii) We have that

$$Kb = K(b)b = \frac{C_1 b^2}{(\epsilon - C_3) b - C_2}.$$

Accordingly,

$$\frac{\mathrm{d}K(b)b}{\mathrm{d}b} = \frac{C_1 b\{(\epsilon - C_3) b - 2C_2\}}{\{(\epsilon - C_3) b - C_2\}^2}, \quad \frac{\mathrm{d}^2 K(b)b}{\mathrm{d}b^2} = \frac{2C_1 C_2^2}{\{(\epsilon - C_3) b - C_2\}^3} \geq 0,$$

which implies that $K(b)b$ is convex with respect to $b$ and

$$\frac{\mathrm{d}K(b)b}{\mathrm{d}b} \begin{cases} < 0 & \text{if } b < b^\star, \\ = 0 & \text{if } b = b^\star = \frac{2C_2}{\epsilon - C_3}, \\ > 0 & \text{if } b > b^\star. \end{cases}$$

The point $b^\star$ attains the minimum value $K(b^\star)b^\star = \frac{4C_1 C_2}{(\epsilon - C_3)^2}$ of $K(b)b$. This completes the proof. $\square$

### A.3.2   MOMENTUM

We show Theorem 3.1 for Momentum.

*Proof.* (i) Lemma A.2 ensures that, for all $k \in \mathbb{N}$ and all $\boldsymbol{\theta} \in \mathbb{R}^d$,

$$
\mathbb{E}\left[(\boldsymbol{\theta}_k - \boldsymbol{\theta})^\top \boldsymbol{m}_{k-1}\right] = \frac{1}{2\alpha\beta_1}\left\{\mathbb{E}\left[\|\boldsymbol{\theta}_k - \boldsymbol{\theta}\|^2\right] - \mathbb{E}\left[\|\boldsymbol{\theta}_{k+1} - \boldsymbol{\theta}\|^2\right]\right\} + \frac{\alpha}{2\beta_1}\mathbb{E}\left[\|\boldsymbol{m}_k\|^2\right]
$$
$$
+ \frac{\hat{\beta}_1}{\beta_1}\mathbb{E}\left[(\boldsymbol{\theta} - \boldsymbol{\theta}_k)^\top \nabla f(\boldsymbol{\theta}_k)\right],
$$

which, together with Lemma A.3, the Cauchy–Schwarz inequality, and (A1) and (A2), implies that

$$
\mathbb{E}\left[(\boldsymbol{\theta}_k - \boldsymbol{\theta})^\top \boldsymbol{m}_{k-1}\right] \leq \frac{1}{2\alpha\beta_1}\left\{\mathbb{E}\left[\|\boldsymbol{\theta}_k - \boldsymbol{\theta}\|^2\right] - \mathbb{E}\left[\|\boldsymbol{\theta}_{k+1} - \boldsymbol{\theta}\|^2\right]\right\} + \frac{\alpha}{2\beta_1}\left(\frac{\sigma^2}{b} + G^2\right)
$$
$$
+ \frac{\hat{\beta}_1}{\beta_1}\mathrm{Dist}(\boldsymbol{\theta})G.
$$

Summing the above inequality from $k = 1$ to $k = K$ gives a relation that implies that

$$
\sum_{k=1}^{K}\mathbb{E}\left[(\boldsymbol{\theta}_k - \boldsymbol{\theta})^\top \boldsymbol{m}_{k-1}\right] \leq \frac{1}{2\alpha\beta_1}\left\{\mathbb{E}\left[\|\boldsymbol{\theta}_1 - \boldsymbol{\theta}\|^2\right] - \mathbb{E}\left[\|\boldsymbol{\theta}_{K+1} - \boldsymbol{\theta}\|^2\right]\right\} + \frac{\alpha}{2\beta_1}\left(\frac{\sigma^2}{b} + G^2\right)K
$$
$$
+ \frac{\hat{\beta}_1}{\beta_1}\mathrm{Dist}(\boldsymbol{\theta})GK,
$$

and hence,

$$
\frac{1}{K}\sum_{k=1}^{K}\mathbb{E}\left[(\boldsymbol{\theta}_k - \boldsymbol{\theta})^\top \boldsymbol{m}_{k-1}\right] \leq \frac{\mathbb{E}\left[\|\boldsymbol{\theta}_1 - \boldsymbol{\theta}\|^2\right]}{2\alpha\beta_1 K} + \frac{\alpha}{2\beta_1}\left(\frac{\sigma^2}{b} + G^2\right) + \frac{\hat{\beta}_1}{\beta_1}\mathrm{Dist}(\boldsymbol{\theta})G.
$$

Moreover, we have that, for all $k \in \mathbb{N}$ and all $\boldsymbol{\theta} \in \mathbb{R}^d$,

$$
(\boldsymbol{\theta}_k - \boldsymbol{\theta})^\top \boldsymbol{m}_k = (\boldsymbol{\theta}_k - \boldsymbol{\theta})^\top \boldsymbol{m}_{k-1} + \hat{\beta}_1(\boldsymbol{\theta}_k - \boldsymbol{\theta})^\top(\nabla f_{B_k}(\boldsymbol{\theta}_k) - \boldsymbol{m}_{k-1})
$$
$$
\leq (\boldsymbol{\theta}_k - \boldsymbol{\theta})^\top \boldsymbol{m}_{k-1} + \hat{\beta}_1\mathrm{Dist}(\boldsymbol{\theta})\left(\|\nabla f_{B_k}(\boldsymbol{\theta}_k)\| + \|\boldsymbol{m}_{k-1}\|\right),
$$

where the first equality comes from the definition of $\boldsymbol{m}_k$ and the first inequality comes from the Cauchy–Schwarz inequality, the triangle inequality, and (A2). Hence, from Lemma A.3, (15), Jensen's inequality, and $b \geq 1$,

$$
\mathbb{E}\left[(\boldsymbol{\theta}_k - \boldsymbol{\theta})^\top \boldsymbol{m}_k\right] \leq \mathbb{E}\left[(\boldsymbol{\theta}_k - \boldsymbol{\theta})^\top \boldsymbol{m}_{k-1}\right] + 2\hat{\beta}_1\mathrm{Dist}(\boldsymbol{\theta})\sqrt{\frac{\sigma^2}{b} + G^2}
$$
$$
\leq \mathbb{E}\left[(\boldsymbol{\theta}_k - \boldsymbol{\theta})^\top \boldsymbol{m}_{k-1}\right] + 2\hat{\beta}_1\mathrm{Dist}(\boldsymbol{\theta})\sqrt{\sigma^2 + G^2}. \tag{19}
$$

Therefore, for all $K \geq 1$ and all $\boldsymbol{\theta} \in \mathbb{R}^d$,

$$
\frac{1}{K}\sum_{k=1}^{K}\mathbb{E}\left[(\boldsymbol{\theta}_k - \boldsymbol{\theta})^\top \boldsymbol{m}_k\right]
$$
$$
\leq \frac{\mathbb{E}\left[\|\boldsymbol{\theta}_1 - \boldsymbol{\theta}\|^2\right]}{2\alpha\beta_1 K} + \frac{\sigma^2\alpha}{2\beta_1 b} + \frac{G^2\alpha}{2\beta_1} + \hat{\beta}_1\mathrm{Dist}(\boldsymbol{\theta})\left(\frac{G}{\beta_1} + 2\sqrt{\sigma^2 + G^2}\right). \tag{20}
$$

The definition of $\boldsymbol{m}_k$ ensures that

$$
(\boldsymbol{\theta}_k - \boldsymbol{\theta})^\top \nabla f_{B_k}(\boldsymbol{\theta}_k)
$$
$$
= (\boldsymbol{\theta}_k - \boldsymbol{\theta})^\top \boldsymbol{m}_k + (\boldsymbol{\theta}_k - \boldsymbol{\theta})^\top(\nabla f_{B_k}(\boldsymbol{\theta}_k) - \boldsymbol{m}_{k-1}) + (\boldsymbol{\theta}_k - \boldsymbol{\theta})^\top(\boldsymbol{m}_{k-1} - \boldsymbol{m}_k)
$$
$$
= (\boldsymbol{\theta}_k - \boldsymbol{\theta})^\top \boldsymbol{m}_k + \frac{1}{\beta_1}(\boldsymbol{\theta}_k - \boldsymbol{\theta})^\top(\nabla f_{B_k}(\boldsymbol{\theta}_k) - \boldsymbol{m}_k) + \hat{\beta}_1(\boldsymbol{\theta}_k - \boldsymbol{\theta})^\top(\boldsymbol{m}_{k-1} - \nabla f_{B_k}(\boldsymbol{\theta}_k)),
$$

which, together with the Cauchy–Schwarz inequality, the triangle inequality, and (A2), implies that

$$(\boldsymbol{\theta}_k - \boldsymbol{\theta})^\top \nabla f_{B_k}(\boldsymbol{\theta}_k)$$
$$\leq (\boldsymbol{\theta}_k - \boldsymbol{\theta})^\top \boldsymbol{m}_k + \frac{1}{\beta_1} \mathrm{Dist}(\boldsymbol{\theta})(\|\nabla f_{B_k}(\boldsymbol{\theta}_k)\| + \|\boldsymbol{m}_k\|) + \hat{\beta}_1 \mathrm{Dist}(\boldsymbol{\theta})(\|\nabla f_{B_k}(\boldsymbol{\theta}_k)\| + \|\boldsymbol{m}_{k-1}\|).$$

Lemma A.3, (15), Jensen's inequality, and $b \geq 1$ guarantee that

$$\mathbb{E}\left[(\boldsymbol{\theta}_k - \boldsymbol{\theta})^\top \nabla f(\boldsymbol{\theta}_k)\right] \leq \mathbb{E}\left[(\boldsymbol{\theta}_k - \boldsymbol{\theta})^\top \boldsymbol{m}_k\right] + 2\left(\frac{1}{\beta_1} + \hat{\beta}_1\right) \mathrm{Dist}(\boldsymbol{\theta}) \sqrt{\frac{\sigma^2}{b} + G^2}$$

$$\leq \mathbb{E}\left[(\boldsymbol{\theta}_k - \boldsymbol{\theta})^\top \boldsymbol{m}_k\right] + 2\left(\frac{1}{\beta_1} + \hat{\beta}_1\right) \mathrm{Dist}(\boldsymbol{\theta}) \sqrt{\sigma^2 + G^2}. \tag{21}$$

Therefore, (20) ensures that, for all $K \geq 1$ and all $\boldsymbol{\theta} \in \mathbb{R}^d$,

$$\frac{1}{K} \sum_{k=1}^{K} \mathbb{E}\left[(\boldsymbol{\theta}_k - \boldsymbol{\theta})^\top \nabla f(\boldsymbol{\theta}_k)\right]$$

$$\leq \underbrace{\frac{\mathbb{E}\left[\|\boldsymbol{\theta}_1 - \boldsymbol{\theta}\|^2\right]}{2\alpha\beta_1} \frac{1}{K}}_{C_1} + \underbrace{\frac{\sigma^2\alpha}{2\beta_1} \frac{1}{b}}_{C_2} + \underbrace{\frac{G^2\alpha}{2\beta_1} + \mathrm{Dist}(\boldsymbol{\theta}) \left\{\frac{G\hat{\beta}_1}{\beta_1} + 2\sqrt{\sigma^2 + G^2}\left(\frac{1}{\beta_1} + 2\hat{\beta}_1\right)\right\}}_{C_3}.$$

(ii) A discussion similar to the one showing (ii) in Theorem 3.1 for SGD would show (ii) in Theorem 3.1 for Momentum.

(iii) An argument similar to that which obtained (iii) in Theorem 3.1 for SGD would prove (iii) in Theorem 3.1 for Momentum. □

### A.3.3 ADAM

We show Theorem 3.1 for Adam.

*Proof.* (i) Let $\boldsymbol{\theta} \in \mathbb{R}^d$. Lemma A.1 guarantees that for all $k \in \mathbb{N}$,

$$\mathbb{E}\left[(\boldsymbol{\theta}_k - \boldsymbol{\theta})^\top \boldsymbol{m}_{k-1}\right] = \underbrace{\frac{\tilde{\beta}_{1k}}{2\alpha\beta_1}\left\{\mathbb{E}\left[\|\boldsymbol{\theta}_k - \boldsymbol{\theta}\|_{\mathsf{H}_k}^2\right] - \mathbb{E}\left[\|\boldsymbol{\theta}_{k+1} - \boldsymbol{\theta}\|_{\mathsf{H}_k}^2\right]\right\}}_{a_k} + \underbrace{\frac{\alpha\tilde{\beta}_{1k}}{2\beta_1}\mathbb{E}\left[\|\mathbf{d}_k\|_{\mathsf{H}_k}^2\right]}_{b_k}$$

$$+ \underbrace{\frac{\hat{\beta}_1}{\beta_1}\mathbb{E}\left[(\boldsymbol{\theta} - \boldsymbol{\theta}_k)^\top \nabla f(\boldsymbol{\theta}_k)\right]}_{c_k}. \tag{22}$$

We define $\gamma_k := \frac{\tilde{\beta}_{1k}}{2\beta_1\alpha}$ $(k \in \mathbb{N})$. Then, for all $K \geq 1$,

$$\sum_{k=1}^{K} a_k = \gamma_1 \mathbb{E}\left[\|\boldsymbol{\theta}_1 - \boldsymbol{\theta}\|_{\mathsf{H}_1}^2\right] + \underbrace{\sum_{k=2}^{K}\left\{\gamma_k \mathbb{E}\left[\|\boldsymbol{\theta}_k - \boldsymbol{\theta}\|_{\mathsf{H}_k}^2\right] - \gamma_{k-1}\mathbb{E}\left[\|\boldsymbol{\theta}_k - \boldsymbol{\theta}\|_{\mathsf{H}_{k-1}}^2\right]\right\}}_{\Gamma_K}$$

$$- \gamma_K \mathbb{E}\left[\|\boldsymbol{\theta}_{K+1} - \boldsymbol{\theta}\|_{\mathsf{H}_K}^2\right]. \tag{23}$$

Since $\overline{\mathsf{H}}_k \in \mathbb{S}_{++}^d$ exists such that $\mathsf{H}_k = \overline{\mathsf{H}}_k^2$, we have $\|\boldsymbol{x}\|_{\mathsf{H}_k}^2 = \|\overline{\mathsf{H}}_k \boldsymbol{x}\|^2$ for all $\boldsymbol{x} \in \mathbb{R}^d$. Accordingly, we also have

$$\Gamma_K = \mathbb{E}\left[\sum_{k=2}^{K}\left\{\gamma_k \left\|\overline{\mathsf{H}}_k(\boldsymbol{\theta}_k - \boldsymbol{\theta})\right\|^2 - \gamma_{k-1}\left\|\overline{\mathsf{H}}_{k-1}(\boldsymbol{\theta}_k - \boldsymbol{\theta})\right\|^2\right\}\right].$$

From $\overline{\mathsf{H}}_k = \mathrm{diag}(\hat{v}_{k,i}^{1/4})$, we have that, for all $\boldsymbol{x} = (x_i)_{i=1}^d \in \mathbb{R}^d$, $\|\overline{\mathsf{H}}_k \boldsymbol{x}\|^2 = \sum_{i=1}^d \sqrt{\hat{v}_{k,i}} x_i^2$. Hence, for all $K \geq 2$,

$$\Gamma_K = \mathbb{E}\left[\sum_{k=2}^K \sum_{i=1}^d \left(\gamma_k \sqrt{\hat{v}_{k,i}} - \gamma_{k-1}\sqrt{\hat{v}_{k-1,i}}\right)(\theta_{k,i} - \theta_i)^2\right]. \tag{24}$$

Condition (2) and $\gamma_k \geq \gamma_{k-1}$ $(k \geq 1)$ imply that, for all $k \geq 1$ and all $i \in [d]$,

$$\gamma_k \sqrt{\hat{v}_{k,i}} - \gamma_{k-1}\sqrt{\hat{v}_{k-1,i}} \geq 0. \tag{25}$$

Moreover, (A2) ensures that $D(\boldsymbol{\theta}) := \sup\{\max_{i \in [d]}(\theta_{k,i} - \theta_i)^2 \colon k \in \mathbb{N}\} < +\infty$. Accordingly, for all $K \geq 2$,

$$\Gamma_K \leq D(\boldsymbol{\theta})\mathbb{E}\left[\sum_{k=2}^K \sum_{i=1}^d \left(\gamma_k \sqrt{\hat{v}_{k,i}} - \gamma_{k-1}\sqrt{\hat{v}_{k-1,i}}\right)\right] = D(\boldsymbol{\theta})\mathbb{E}\left[\sum_{i=1}^d \left(\gamma_K \sqrt{\hat{v}_{K,i}} - \gamma_1 \sqrt{\hat{v}_{1,i}}\right)\right].$$

Let $\nabla f_{B_k}(\boldsymbol{\theta}_k) \odot \nabla f_{B_k}(\boldsymbol{\theta}_k) := (g_{k,i}^2) \in \mathbb{R}_+^d$. Assumption (A1) ensures that there exists $M \in \mathbb{R}$ such that, for all $k \in \mathbb{N}$, $\max_{i \in [d]} g_{k,i}^2 \leq M$. The definition of $\boldsymbol{v}_k$ guarantees that, for all $i \in [d]$ and all $k \in \mathbb{N}$,

$$v_{k,i} = \beta_2 v_{k-1,i} + \hat{\beta}_2 g_{k,i}^2.$$

Induction thus ensures that, for all $i \in [d]$ and all $k \in \mathbb{N}$,

$$v_{k,i} \leq \max\{v_{0,i}, M\} = M, \tag{26}$$

where $\boldsymbol{v}_0 = (v_{0,i}) = \boldsymbol{0}$. From the definition of $\hat{\boldsymbol{v}}_k$, we have that, for all $i \in [d]$ and all $k \in \mathbb{N}$,

$$\hat{v}_{k,i} = \frac{v_{k,i}}{\tilde{\beta}_{2k}} \leq \frac{M}{\tilde{\beta}_{2k}}. \tag{27}$$

Therefore, (23), $\mathbb{E}[\|\boldsymbol{\theta}_1 - \boldsymbol{\theta}\|_{\mathsf{H}_1}^2] \leq D(\boldsymbol{\theta})\mathbb{E}[\sum_{i=1}^d \sqrt{\hat{v}_{1,i}}]$, and (27) imply, for all $K \geq 1$,

$$\begin{aligned}
\sum_{k=1}^K a_k &\leq \gamma_1 D(\boldsymbol{\theta})\mathbb{E}\left[\sum_{i=1}^d \sqrt{\hat{v}_{1,i}}\right] + D(\boldsymbol{\theta})\mathbb{E}\left[\sum_{i=1}^d \left(\gamma_K \sqrt{\hat{v}_{K,i}} - \gamma_1 \sqrt{\hat{v}_{1,i}}\right)\right] \\
&= \gamma_K D(\boldsymbol{\theta})\mathbb{E}\left[\sum_{i=1}^d \sqrt{\hat{v}_{K,i}}\right] \\
&\leq \gamma_K D(\boldsymbol{\theta})\sum_{i=1}^d \sqrt{\frac{M}{\tilde{\beta}_{2K}}} \\
&\leq \frac{dD(\boldsymbol{\theta})\sqrt{M}\tilde{\beta}_{1K}}{2\beta_1\alpha\sqrt{\tilde{\beta}_{2K}}}.
\end{aligned} \tag{28}$$

Inequality (28) with $\tilde{\beta}_{1K} := 1 - \beta_1^{K+1} \leq 1$ and $\tilde{\beta}_{2K} := 1 - \beta_2^{K+1} \geq 1 - \beta_2 =: \hat{\beta}_2$ implies that

$$\sum_{k=1}^K a_k \leq \frac{dD(\boldsymbol{\theta})\sqrt{M}\tilde{\beta}_{1K}}{2\beta_1\alpha\sqrt{\tilde{\beta}_{2K}}} \leq \frac{dD(\boldsymbol{\theta})\sqrt{M}}{2\beta_1\alpha\sqrt{\hat{\beta}_2}}. \tag{29}$$

Lemma A.3 guarantees that, for all $k \in \mathbb{N}$,

$$b_k = \frac{\alpha\tilde{\beta}_{1k}}{2\beta_1}\mathbb{E}\left[\|\mathbf{d}_k\|_{\mathsf{H}_k}^2\right] \leq \frac{\alpha\tilde{\beta}_{1k}}{2\beta_1}\frac{\sqrt{\tilde{\beta}_{2k}}}{\tilde{\beta}_{1k}^2\sqrt{v_*}}\left(\frac{\sigma^2}{b} + G^2\right) = \frac{\alpha\sqrt{\tilde{\beta}_{2k}}}{2\sqrt{v_*}\beta_1\tilde{\beta}_{1k}}\left(\frac{\sigma^2}{b} + G^2\right). \tag{30}$$

Inequality (30) with $\tilde{\beta}_{1k} := 1 - \beta_1^{k+1} \geq 1 - \beta_1 =: \hat{\beta}_1$ and $\tilde{\beta}_{2k} := 1 - \beta_2^{k+1} \leq 1$ implies that

$$b_k \leq \frac{\alpha\sqrt{\tilde{\beta}_{2k}}}{2\sqrt{v_*}\beta_1\tilde{\beta}_{1k}}\left(\frac{\sigma^2}{b} + G^2\right) \leq \frac{\alpha}{2\sqrt{v_*}\beta_1\hat{\beta}_1}\left(\frac{\sigma^2}{b} + G^2\right). \tag{31}$$

The Cauchy–Schwarz inequality and (A2) imply that, for all $k \in \mathbb{N}$,

$$c_k = \frac{\hat{\beta}_1}{\beta_1} \mathbb{E}\left[(\boldsymbol{\theta} - \boldsymbol{\theta}_k)^\top \nabla f(\boldsymbol{\theta}_k)\right] \le \mathrm{Dist}(\boldsymbol{\theta}) G \frac{\hat{\beta}_1}{\beta_1}. \tag{32}$$

Hence, (22), (29), (31), and (32) ensure that, for all $K \ge 1$,

$$\frac{1}{K} \sum_{k=1}^{K} \mathbb{E}\left[(\boldsymbol{\theta}_k - \boldsymbol{\theta})^\top \boldsymbol{m}_{k-1}\right] \le \frac{dD(\boldsymbol{\theta})\sqrt{M}}{2\beta_1 \alpha \sqrt{\hat{\beta}_2 K}} + \frac{\alpha(\sigma^2 b^{-1} + G^2)}{2\sqrt{v_*}\beta_1 \hat{\beta}_1} + \mathrm{Dist}(\boldsymbol{\theta}) G \frac{\hat{\beta}_1}{\beta_1}.$$

Therefore, from (19), for all $K \ge 1$,

$$\begin{aligned}
&\frac{1}{K} \sum_{k=1}^{K} \mathbb{E}\left[(\boldsymbol{\theta}_k - \boldsymbol{\theta})^\top \boldsymbol{m}_k\right] \\
&\le \frac{dD(\boldsymbol{\theta})\sqrt{M}}{2\beta_1 \alpha \sqrt{\hat{\beta}_2 K}} + \frac{\sigma^2 \alpha}{2\sqrt{v_*}\beta_1 \hat{\beta}_1 b} + \frac{G^2 \alpha}{2\sqrt{v_*}\beta_1 \hat{\beta}_1} + \hat{\beta}_1 \mathrm{Dist}(\boldsymbol{\theta})\left(\frac{G}{\beta_1} + 2\sqrt{\sigma^2 + G^2}\right).
\end{aligned} \tag{33}$$

From (21) and (33), for all $K \ge 1$,

$$\begin{aligned}
&\frac{1}{K} \sum_{k=1}^{K} \mathbb{E}\left[(\boldsymbol{\theta}_k - \boldsymbol{\theta})^\top \nabla f(\boldsymbol{\theta}_k)\right] \\
&\le \underbrace{\frac{dD(\boldsymbol{\theta})\sqrt{M}}{2\alpha\beta_1 \sqrt{\hat{\beta}_2}}}_{C_1} \frac{1}{K} + \underbrace{\frac{\sigma^2 \alpha}{2\sqrt{v_*}\beta_1 \hat{\beta}_1}}_{C_2} \frac{1}{b} + \underbrace{\frac{G^2 \alpha}{2\sqrt{v_*}\beta_1 \hat{\beta}_1} + \mathrm{Dist}(\boldsymbol{\theta})\left\{\frac{G\hat{\beta}_1}{\beta_1} + 2\sqrt{\sigma^2 + G^2}\left(\frac{1}{\beta_1} + 2\hat{\beta}_1\right)\right\}}_{C_3}.
\end{aligned}$$

(ii) A discussion similar to the one showing (ii) in Theorem 3.1 for SGD would show (ii) in Theorem 3.1 for Adam.

(iii) An argument similar to that which obtained (iii) in Theorem 3.1 for SGD would prove (iii) in Theorem 3.1 for Adam. $\qquad\square$

### A.3.4 AMSGRAD

We show that $C_i$ for AMSGrad are (6).

*Proof.* Let us consider AMSGrad (Algorithm 1 with $\hat{\boldsymbol{m}}_k = \boldsymbol{m}_k$, $\hat{\boldsymbol{v}}_k = \boldsymbol{v}_k$, $\tilde{v}_{k,i} := \max\{\tilde{v}_{k-1,i}, v_{k,i}\}$, and $\mathsf{H}_k := \mathrm{diag}(\sqrt{\tilde{v}_{k,i}})$). Induction, together with (26) and $\tilde{v}_{k,i} := \max\{\tilde{v}_{k-1,i}, v_{k,i}\}$, ensures that, for all $k \in \mathbb{N}$, $\tilde{v}_{k,i} \le M$, where $\tilde{\boldsymbol{v}}_{-1} = (\tilde{v}_{-1,i}) = \boldsymbol{0}$. An argument similar to that which showed (i) in Appendix A.3.3 ensures that

$$\sum_{k=1}^{K} a_k \le \frac{dD(\boldsymbol{\theta})\sqrt{M}}{2\alpha\beta_1}, \ b_k \le \frac{\alpha}{2\sqrt{\tilde{v}_*}\beta_1}\left(\frac{\sigma^2}{b} + G^2\right), \text{ and } c_k \le \mathrm{Dist}(\boldsymbol{\theta}) G \frac{\hat{\beta}_1}{\beta_1},$$

where $a_k$, $b_k$, and $c_k$ are defined as in (22) and $\tilde{v}_* := \inf\{\min_{i \in [d]} \tilde{v}_{k,i} : k \in \mathbb{N}\}$. Inequalities (21) and (33) thus imply that $C_i$ for AMSGrad are (6). $\qquad\square$

### A.4 RESULTS FOR VARYING LEARNING RATES

We can establish the upper bound of $\mathrm{VI}(K, \boldsymbol{\theta})$ for Adam with varying learning rates from a discussion similar to the one for proving Theorem 3.1(i) (Appendix A.3.3). Let $(\alpha_k)_{k \in \mathbb{N}}$ be monotone decreasing. Then, $\gamma_k := \frac{\hat{\beta}_{1k}}{2\beta_1 \alpha_k}$ satisfies $\gamma_{k+1} \ge \gamma_k$ ($k \in \mathbb{N}$). Hence, (25) holds. The discussion in Appendix A.3.3 thus ensures that

$$\mathrm{VI}(K, \boldsymbol{\theta}) \le \frac{dD(\boldsymbol{\theta})\sqrt{M}}{2\beta_1 \sqrt{1 - \beta_2}\alpha_K K} + \frac{\sigma^2 b^{-1} + G^2}{2\sqrt{v_*}\beta_1(1 - \beta_1)K} \sum_{k=1}^{K} \alpha_k \sqrt{1 - \beta_2^{k+1}} + h(\beta_1). \tag{34}$$

Let us consider the case where $\alpha_k = \frac{1}{\sqrt{k}}$ $(k \geq 1)$. From (34), we have that

$$
\begin{aligned}
\text{VI}(K, \boldsymbol{\theta}) &\leq \frac{dD(\boldsymbol{\theta})\sqrt{M}}{2\beta_1\sqrt{1-\beta_2}\sqrt{K}} + \frac{\sigma^2 b^{-1} + G^2}{2\sqrt{v_*}\beta_1(1-\beta_1)K} \sum_{k=1}^{K} \frac{1}{\sqrt{k}} + h(\beta_1) \\
&\leq \frac{dD(\boldsymbol{\theta})\sqrt{M}}{2\beta_1\sqrt{1-\beta_2}\sqrt{K}} + \frac{\sigma^2 b^{-1} + G^2}{\sqrt{v_*}\beta_1(1-\beta_1)} \frac{\sqrt{K}}{K} + h(\beta_1) \\
&= \frac{dD(\boldsymbol{\theta})\sqrt{M}}{2\beta_1\sqrt{1-\beta_2}\sqrt{K}} + \frac{\sigma^2 b^{-1} + G^2}{\sqrt{v_*}\beta_1(1-\beta_1)} \frac{1}{\sqrt{K}} + h(\beta_1),
\end{aligned}
$$

where $1 - \beta_2^{k+1} \leq 1$ and $\sum_{k=1}^{K} \frac{1}{\sqrt{k}} \leq 2\sqrt{K}$. Hence,

$$
\begin{aligned}
\text{VI}(K, \boldsymbol{\theta}) &\leq \left( \frac{dD(\boldsymbol{\theta})\sqrt{M}}{2\beta_1\sqrt{1-\beta_2}} + \frac{G^2}{\sqrt{v_*}\beta_1(1-\beta_1)} \right) \frac{1}{\sqrt{K}} + \frac{\sigma^2}{\sqrt{v_*}\beta_1(1-\beta_1)} \frac{1}{b\sqrt{K}} + h(\beta_1) \\
&=: \frac{\tilde{C}_1}{\sqrt{K}} + \frac{\tilde{C}_2}{b\sqrt{K}} + h(\beta_1).
\end{aligned}
\tag{35}
$$

Theorem 3.1(i) indicates that Adam with a constant learning rate $\alpha$ satisfies

$$
\text{VI}(K, \boldsymbol{\theta}) \leq \frac{C_1}{K} + \frac{C_2}{b} + C_3.
$$

Hence, using $\alpha_k = \frac{1}{\sqrt{k}}$ mitigates the variance (the term $\frac{\tilde{C}_2}{b\sqrt{K}}$) in contrast to the term $\frac{C_2}{b}$ using $\alpha_k = \alpha$. However, the bias term $\frac{C_1}{K}$ using $\alpha_k = \alpha$ would be better than the bias term $\frac{\tilde{C}_1}{\sqrt{K}}$ using $\alpha_k = \frac{1}{\sqrt{k}}$ in the sense of minimizing the upper bound of $\text{VI}(K, \boldsymbol{\theta})$.

Let $T \geq 1$, $\gamma \in (0, 1)$, $\alpha > 0$, and $P := \frac{K}{T}$. Next, let us consider the following varying learning rate

$$
(\alpha_k) = (\alpha_T, \alpha_{2T}, \ldots, \alpha_{PT}),
$$

where $\alpha_{jT} = (\gamma^{j-1}\alpha, \gamma^{j-1}\alpha, \ldots, \gamma^{j-1}\alpha)$ $(j = 1, 2, \ldots, P)$. That is, $(\alpha_k)$ is

$$
((\underbrace{\alpha, \alpha, \ldots, \alpha}_{T}), (\underbrace{\gamma\alpha, \gamma\alpha, \ldots, \gamma\alpha}_{T}), \ldots, (\underbrace{\gamma^{P-1}\alpha, \gamma^{P-1}\alpha, \ldots, \gamma^{P-1}\alpha}_{T})).
$$

Let $\underline{\alpha} > 0$ be the lower bound of $\alpha_k$. From

$$
\sum_{k=1}^{K} \alpha_k = \alpha T + \gamma\alpha T + \cdots + \gamma^{P-1}\alpha T \leq \frac{\alpha T}{1 - \gamma}
$$

and (34), we have that

$$
\begin{aligned}
\text{VI}(TP, \boldsymbol{\theta}) &\leq \frac{dD(\boldsymbol{\theta})\sqrt{M}}{2\beta_1\sqrt{1-\beta_2}\underline{\alpha}TP} + \frac{\sigma^2 b^{-1} + G^2}{2\sqrt{v_*}\beta_1(1-\beta_1)TP} \frac{\alpha T}{1-\gamma} + h(\beta_1) \\
&= \frac{dD(\boldsymbol{\theta})\sqrt{M}}{2\beta_1\sqrt{1-\beta_2}\underline{\alpha}TP} + \frac{(\sigma^2 b^{-1} + G^2)\alpha}{2\sqrt{v_*}\beta_1(1-\beta_1)(1-\gamma)P} + h(\beta_1) \\
&= \left( \frac{dD(\boldsymbol{\theta})\sqrt{M}}{2\beta_1\sqrt{1-\beta_2}\underline{\alpha}T} + \frac{G^2\alpha}{2\sqrt{v_*}\beta_1(1-\beta_1)(1-\gamma)} \right) \frac{1}{P} + \frac{\sigma^2\alpha}{2\sqrt{v_*}\beta_1(1-\beta_1)(1-\gamma)} \frac{1}{Pb} \\
&\quad + h(\beta_1) \\
&= \mathcal{O}\left( \frac{1}{P} + \frac{1}{Pb} \right) + h(\beta_1).
\end{aligned}
$$

Here, we recall that Adam with $\alpha_k = \alpha$ satisfies

$$
\text{VI}(TP, \boldsymbol{\theta}) \leq \frac{C_1}{TP} + \frac{C_2}{b} + C_3 = \mathcal{O}\left( \frac{1}{P} + \frac{1}{b} \right) + C_3,
$$

and Adam with $\alpha_k = \frac{1}{\sqrt{k}}$ satisfies

$$\text{VI}(TP, \boldsymbol{\theta}) \leq \frac{\tilde{C}_1}{\sqrt{TP}} + \frac{\tilde{C}_2}{b\sqrt{TP}} + h(\beta_1) = \mathcal{O}\left(\frac{1}{\sqrt{P}} + \frac{1}{b\sqrt{P}}\right) + h(\beta_1).$$

Therefore, using

$$(\alpha_k) = (\alpha_T, \alpha_{2T}, \ldots, \alpha_{PT})$$

is more desirable to minimize the upper bound of $\text{VI}(K, \boldsymbol{\theta})$ for Adam than using $\alpha_k = \alpha, \frac{1}{\sqrt{k}}$.

## A.5 Boundedness condition of $(\nabla f(\theta_k))_{k\in\mathbb{N}}$

Let $f: \mathbb{R}^d \to \mathbb{R}$ be convex. Then, $f$ is Lipschitz continuous (i.e., $|f(\boldsymbol{x}) - f(\boldsymbol{y})| \leq G\|\boldsymbol{x} - \boldsymbol{y}\|$) if and only if $\|\nabla f(\boldsymbol{x})\| \leq G$ ($\boldsymbol{x} \in \mathbb{R}^d$) (see, e.g., Theorem 6.2.2, Corollary 6.1.2, and Exercise 6.1.9(c) in (Borwein & Lewis, 2000)). Let $\boldsymbol{\theta}^*$ be a local minimizer of a Lipschitz continuous function $f$. The continuity of $f$ ensures that $f$ is convex around $\boldsymbol{\theta}^*$. Hence, for any $\boldsymbol{\theta}$ belonging to a neighborhood $N(\boldsymbol{\theta}^*)$ of $\boldsymbol{\theta}^*$, $\|\nabla f(\boldsymbol{\theta})\| \leq G$. If the sequence $(\boldsymbol{\theta}_k)_{k\in\mathbb{N}}$ generated by an optimizer approximates $\boldsymbol{\theta}^*$, then $\boldsymbol{\theta}_k \in N(\boldsymbol{\theta}^*)$ for sufficiently large $k$, i.e., $\|\nabla f(\boldsymbol{\theta}_k)\| \leq G$.

## A.6 Condition $\epsilon - C_3 > 0$

Under Lipschitz smoothness condition of $f$ (i.e., $\|\nabla f(\boldsymbol{x}) - \nabla f(\boldsymbol{y})\| \leq L\|\boldsymbol{x} - \boldsymbol{y}\|$), the gradient descent (GD) with a constant learning rate $\alpha = \mathcal{O}(L^{-1})$ satisfies that $\lim_{k\to+\infty}\|\nabla f(\boldsymbol{\theta}_k)\| = 0$ by the descent lemma ($f(\boldsymbol{y}) \leq f(\boldsymbol{x}) + \nabla f(\boldsymbol{x})^\top(\boldsymbol{y}-\boldsymbol{x}) + \frac{L}{2}\|\boldsymbol{y}-\boldsymbol{x}\|^2$). Under non-smoothness condition of $f$, GD with a constant learning rate $\alpha > 0$ satisfies that $\liminf_{k\to+\infty} \nabla f(\boldsymbol{\theta}_k)^\top(\boldsymbol{\theta}_k - \boldsymbol{\theta}) \leq \frac{G^2}{2}\alpha$. Hence, it is not guaranteed that, under non-smoothness condition of $f$, GD with a constant learning rate $\alpha > 0$ converges. However, we can expect that using a small constant learning rate $\alpha$ approximates a stationary point of $f$.

Meanwhile, under non-smoothness condition of $f$, GD with a diminishing learning rate $\alpha_k = \frac{1}{\sqrt{k}}$ satisfies that

$$\text{VI}(K, \boldsymbol{\theta}) \leq \frac{D(\boldsymbol{\theta})}{2K\alpha_K} + \frac{\sigma^2 b^{-1} + G^2}{2K}\sum_{k=1}^{K}\alpha_k = \frac{D(\boldsymbol{\theta})}{2K\alpha_K} + \frac{G^2}{2K}\sum_{k=1}^{K}\alpha_k = \mathcal{O}\left(\frac{1}{\sqrt{K}}\right)$$

(see also Appendix A.3.1 when $\alpha$ is $\frac{1}{\sqrt{k}}$). Hence, GD with a diminishing learning rate does not need $C_3$ used in Theorem 3.1. This result strongly depends on the condition $\beta_1 = 0$.

Under non-smoothness condition of $f$, Adam with a diminishing learning rate $\alpha_k = \frac{1}{\sqrt{k}}$ and $b = n$ (i.e., $\sigma = 0$) satisfies that

$$\text{VI}(K, \boldsymbol{\theta}) \leq \frac{\tilde{C}_1}{\sqrt{K}} + \frac{\tilde{C}_2}{b\sqrt{K}} + h(\beta_1) = \frac{\tilde{C}_1}{\sqrt{K}} + h(\beta_1)$$

(see (35)) and Adam with a constant learning rate $\alpha$ and $b = n$ (i.e., $\sigma = 0$) satisfies that

$$\text{VI}(K, \boldsymbol{\theta}) \leq \frac{C_1}{K} + \frac{C_2}{b} + C_3 = \frac{C_1}{K} + C_3.$$

The previous results (Zaheer et al., 2018) indicated that Adam with a constant learning rate $\alpha = \mathcal{O}(L^{-1})$, $\beta_1 = 0$, and $\sigma = 0$ satisfies that

$$\frac{1}{K}\sum_{k=1}^{K}\mathbb{E}\left[\|\nabla f(\boldsymbol{\theta}_k)\|^2\right] \leq 2\left(\sqrt{\beta_2}G + \epsilon\right)\frac{f(\boldsymbol{\theta}_1) - f(\boldsymbol{\theta}^\star)}{\alpha K} =: \frac{\hat{C}_1}{K}.$$

The condition $\beta_1 \neq 0$ is essential to analyze Adam since the practically useful $\beta_1$ is close to 1. The upper bound of $\text{VI}(K, \boldsymbol{\theta})$ for Adam with $\beta_1 \neq 0$ depends on the term with respect to $\beta_1$, i.e., $C_3 := \frac{G^2\alpha}{2\sqrt{v_*}\beta_1(1-\beta_1)} + h(\beta_1)$. Hence, $\epsilon > C_3$ and $\epsilon > h(\beta_1)$ are needed to consider $K$ to satisfy $\text{VI}(K, \boldsymbol{\theta}) \leq \frac{\tilde{C}_1}{\sqrt{K}} + h(\beta_1) = \epsilon$ (diminishing learning rate case) and $\text{VI}(K, \boldsymbol{\theta}) \leq \frac{C_1}{K} + C_3 = \epsilon$ (constant learning rate case) even if $\sigma = 0$.

## A.7 Lower and upper bounds on $K$ with $\mathrm{VI}(K, \boldsymbol{\theta}^*) = \epsilon$

Let us consider SGD under (C1)–(C3) and let $\boldsymbol{\theta}^*$ be a stationary point $f$. From Lemma A.2, we have that

$$2\alpha \mathbb{E}\left[(\boldsymbol{\theta}_k - \boldsymbol{\theta}^*)^\top \nabla f(\boldsymbol{\theta}_k)\right] = \mathbb{E}\left[\|\boldsymbol{\theta}_k - \boldsymbol{\theta}^*\|^2\right] - \mathbb{E}\left[\|\boldsymbol{\theta}_{k+1} - \boldsymbol{\theta}^*\|^2\right] + \alpha^2 \mathbb{E}\left[\|\nabla f_{B_k}(\boldsymbol{\theta}_k)\|^2\right].$$

Hence, for all $K \geq 1$,

$$2\alpha \sum_{k=1}^{K} \mathbb{E}\left[(\boldsymbol{\theta}_k - \boldsymbol{\theta}^*)^\top \nabla f(\boldsymbol{\theta}_k)\right] = \mathbb{E}\left[\|\boldsymbol{\theta}_1 - \boldsymbol{\theta}^*\|^2\right] - \mathbb{E}\left[\|\boldsymbol{\theta}_{K+1} - \boldsymbol{\theta}^*\|^2\right] + \alpha^2 \sum_{k=1}^{K} \mathbb{E}\left[\|\nabla f_{B_k}(\boldsymbol{\theta}_k)\|^2\right].$$

Here, we assume that there exists $c \in [0, 1]$ such that $c\sigma^2 \leq \mathbb{E}[\|\mathsf{G}_{\xi_k}(\boldsymbol{\theta}_k) - \nabla f(\boldsymbol{\theta}_k)\|^2] \leq \sigma^2$. From (14), we have that

$$\mathbb{E}\left[\|\nabla f_{B_k}(\boldsymbol{\theta}_k)\|^2 \Big| \boldsymbol{\theta}_k\right] = \mathbb{E}\left[\|\nabla f_{B_k}(\boldsymbol{\theta}_k) - \nabla f(\boldsymbol{\theta}_k)\|^2 \Big| \boldsymbol{\theta}_k\right] + \|\nabla f(\boldsymbol{\theta}_k)\|^2 \geq \frac{c\sigma^2}{b}.$$

Since $\boldsymbol{\theta}_{K+1}$ approximates $\boldsymbol{\theta}^*$, we assume that there exists $X(\boldsymbol{\theta}^*) \geq 0$ such that $\mathbb{E}[\|\boldsymbol{\theta}_1 - \boldsymbol{\theta}^*\|^2] - \mathbb{E}[\|\boldsymbol{\theta}_{K+1} - \boldsymbol{\theta}^*\|^2] \geq X(\boldsymbol{\theta}^*)$. Hence,

$$2\alpha \sum_{k=1}^{K} \mathbb{E}\left[(\boldsymbol{\theta}_k - \boldsymbol{\theta}^*)^\top \nabla f(\boldsymbol{\theta}_k)\right] \geq X(\boldsymbol{\theta}^*) + \frac{c\sigma^2 \alpha^2 K}{b},$$

which implies that

$$\mathrm{VI}(K, \boldsymbol{\theta}^*) \geq \frac{D_1}{K} + \frac{D_2}{b},$$

where

$$D_1 := \frac{X(\boldsymbol{\theta}^*)}{2\alpha} \quad \text{and} \quad D_2 := \frac{c\sigma^2 \alpha}{2}.$$

Meanwhile, Theorem 3.1(i) indicates that

$$\mathrm{VI}(K, \boldsymbol{\theta}^*) \leq \frac{C_1}{K} + \frac{C_2}{b} + C_3,$$

where

$$C_1 := \frac{\mathbb{E}[\|\boldsymbol{\theta}_1 - \boldsymbol{\theta}^*\|^2]}{2\alpha}, \quad C_2 := \frac{\sigma^2 \alpha}{2}, \quad \text{and} \quad C_3 := \frac{G^2 \alpha}{2}.$$

Suppose that $\mathrm{VI}(K, \boldsymbol{\theta}^*) \leq \epsilon$. Then, we have that

$$\frac{D_1}{K} + \frac{D_2}{b} \leq \epsilon,$$

which implies that

$$\underline{K}(b) := \frac{D_1 b}{\epsilon b - D_2} \leq K.$$

Suppose that $\mathrm{VI}(K, \boldsymbol{\theta}^*) \geq \epsilon$. Then, we have that

$$\frac{C_1}{K} + \frac{C_2}{b} + C_3 \geq \epsilon,$$

which implies that

$$\overline{K}(b) := \frac{C_1 b}{(\epsilon - C_3) b - C_2} \geq K.$$

From

$$D_1 = \frac{X(\boldsymbol{\theta}^*)}{2\alpha} \leq \frac{\mathbb{E}[\|\boldsymbol{\theta}_1 - \boldsymbol{\theta}^*\|^2] - \mathbb{E}[\|\boldsymbol{\theta}_{K+1} - \boldsymbol{\theta}^*\|^2]}{2\alpha} \leq \frac{\mathbb{E}[\|\boldsymbol{\theta}_1 - \boldsymbol{\theta}^*\|^2]}{2\alpha} = C_1,$$

$$D_2 = \frac{c\sigma^2\alpha}{2} \leq \frac{\sigma^2\alpha}{2} = C_2,$$

and

$$(\epsilon - C_3)b - C_2 \leq \epsilon b - C_2 \leq \epsilon b - D_2,$$

we have that $\underline{K}(b) \leq \overline{K}(b)$. Hence, $K$ with $\text{VI}(K, \boldsymbol{\theta}^*) = \epsilon$ satisfies

$$\underline{K}(b) \leq K \leq \overline{K}(b)$$

and the SFO complexity $N = Kb$ satisfies

$$\underline{K}(b)b =: \underline{N} \leq N \leq \overline{N} := \overline{K}(b)b.$$

Moreover, $b_\star := \frac{2D_2}{\epsilon}$ minimizing $\underline{N}$ and $b^\star := \frac{2C_2}{\epsilon - C_3}$ minimizing $\overline{N}$ (see Theorem 3.1(iii)) satisfy that $b_\star \leq b^\star$. Let $\alpha$ be small enough. Then, we have that

$$b_\star \approx b^\star \text{ and } \underline{K}(b_\star)b_\star \approx \overline{K}(b^\star)b^\star,$$

where

$$C_1 = \frac{\mathbb{E}[\|\boldsymbol{\theta}_1 - \boldsymbol{\theta}^*\|^2]}{2\alpha} \approx D_1 = \frac{X(\boldsymbol{\theta}^*)}{2\alpha}, \ C_2 = \frac{\sigma^2\alpha}{2} \approx D_2 = \frac{c\sigma^2\alpha}{2}, \text{ and } C_3 = \frac{G^2\alpha}{2} \approx 0$$

are used. Hence, the batch sizes $b_\star$ and $b^\star$ ($b_\star \approx b^\star$) approximate the batch size minimizing $N = Kb$.

## A.8 SFO COMPLEXITY REGARDING COMPUTATIONAL COST OF PARALLELIZABLE STEPS

Let us consider Tail-averaged SGD (Jain et al., 2018, Algorithm 1) independently in $P$ machines, each of which contains $\frac{n}{P}$ samples, for minimizing the expected square loss function $f(\boldsymbol{\theta}) := \frac{1}{2}\mathbb{E}_{(\boldsymbol{x},y)\sim\mathcal{D}}[y - \boldsymbol{x}^\top\boldsymbol{\theta}]$ (see Appendix A.1.1). Let $\bar{\boldsymbol{\theta}}_i$ be the point generated by Tail-averaged SGD with the batch size $b$ on machine $i$ ($i \in [P]$) and let $\bar{\boldsymbol{\theta}} := \frac{1}{P}\sum_{i=1}^P \bar{\boldsymbol{\theta}}_i$ be the point generated by Parallelizing Tail-averaged SGD. Then, Theorem 6 in (Jain et al., 2018) guarantees that

$$\mathbb{E}[f(\bar{\boldsymbol{\theta}})] - f(\boldsymbol{\theta}^\star) \leq \frac{(1-\alpha\mu)^s}{\alpha^2\mu^2(\frac{n}{Pb} - s)^2}\frac{2 + (P-1)(1-\alpha\mu)^s}{P}(f(\boldsymbol{\theta}_0) - f(\boldsymbol{\theta}^\star)) + \frac{4\sigma^2}{Pb(\frac{n}{Pb} - s)},$$

where $s$ is the initial iterations, and $n$ is the total samples, $\mu > 0$ is the smallest eigenvalue of the Hessian of $f$, and $(\boldsymbol{\theta}_i)_{i>s}$ and $\bar{\boldsymbol{\theta}}_i := \frac{1}{\lfloor\frac{n}{Pb}\rfloor - s}\sum_{i>s}\boldsymbol{\theta}_i$ are the sequences generated by Tail-averaged SGD (see also Appendix A.1.1). This result with $P = 1$ coincides with (12). Setting $s > \kappa_b \log P$ and $\alpha = \frac{\alpha_{b,\max}}{2}$ (Jain et al., 2018, P. 15, Remarks) ensures that

$$\mathbb{E}[f(\bar{\boldsymbol{\theta}})] - f(\boldsymbol{\theta}^\star) \leq \exp\left(-\frac{s}{\kappa_b}\right)\frac{3\kappa_b^2}{(\frac{n}{Pb} - s)^2 P}(f(\boldsymbol{\theta}_0) - f(\boldsymbol{\theta}^\star)) + \frac{4\sigma^2}{Pb(\frac{n}{Pb} - s)},$$

which, together with $K := \frac{n}{Pb} - s$, implies that

$$\mathbb{E}[f(\bar{\boldsymbol{\theta}})] - f(\boldsymbol{\theta}^\star) = \mathcal{O}\left(\frac{1}{K^2 P} + \frac{1}{KPb}\right). \tag{36}$$

Hence, (36) indicates that the larger the number of machines $P$ is, the smaller the upper bound of $\mathbb{E}[f(\bar{\boldsymbol{\theta}})] - f(\boldsymbol{\theta}^\star)$ becomes.

Next, let us consider SGD independently in $P$ machines, each of which contains $\frac{n}{P}$ samples, for minimizing the expected square loss function $f$. Given $\boldsymbol{\theta}_k \in \mathbb{R}^d$, machine $i$ ($i \in [P]$) generates the point

$$\boldsymbol{\theta}_{k+1}^{(i)} := \boldsymbol{\theta}_k - \alpha\nabla f_{B_k^{(i)}}(\boldsymbol{\theta}_k) = \boldsymbol{\theta}_k - \frac{\alpha}{b}\sum_{j\in[b]}\mathsf{G}_{\xi_{k,j}^{(i)}}(\boldsymbol{\theta}_k)$$

using SGD with the batch size $b$ and computes

$$\boldsymbol{\theta}_{k+1} := \frac{1}{P}\sum_{i=1}^P \boldsymbol{\theta}_{k+1}^{(i)}, \tag{37}$$

where $\xi_{k,j}^{(i)}$ is a random variable generated by the $j$-th sampling in the $k$-the iteration for machine $i$ and we assume that the stochastic gradient $\mathsf{G}_{\xi_{k,j}^{(i)}}(\boldsymbol{\theta}_k)$ satisfies (C2).

We have the following theorem.

**Theorem A.1.** *Consider minimizing the expected square loss function $f(\boldsymbol{\theta}) := \frac{1}{2}\mathbb{E}_{(\boldsymbol{x},y)\sim\mathcal{D}}[y - \boldsymbol{x}^\top\boldsymbol{\theta}]$ and let $\boldsymbol{\theta}^\star$ be a minimizer of $f$. Then, under (C2) and (A1), the sequence $(\boldsymbol{\theta}_k)$ generated by Parallelizing SGD (37) satisfies the following:*

(i) [Upper bound of $\mathrm{VI}(K, \boldsymbol{\theta}^\star)$] *For all $K \geq 1$,*

$$\mathbb{E}\left[f\left(\frac{1}{K}\sum_{k=1}^{K}\boldsymbol{\theta}_k\right)\right] - f(\boldsymbol{\theta}^\star) \leq \mathrm{VI}(K, \boldsymbol{\theta}^\star) \leq \frac{C_1}{K} + \frac{C_2}{Pb} + C_3,$$

*where $P$ is the number of machines,*

$$C_1 := \frac{\max_{i\in[P]}\mathbb{E}[\|\boldsymbol{\theta}_1^{(i)} - \boldsymbol{\theta}^\star\|^2]}{2\alpha}, \ C_2 := \frac{\sigma^2\alpha}{2}, \ C_3 := \frac{G^2\alpha}{2}.$$

(ii) [Steps to satisfy $\mathrm{VI}(K, \boldsymbol{\theta}^\star) \leq \epsilon$] *The number of steps $K_P$ defined by*

$$K_P(b) = \frac{C_1 Pb}{(\epsilon - C_3)Pb - C_2} \tag{38}$$

*satisfies $\mathrm{VI}(K_P, \boldsymbol{\theta}^\star) \leq \epsilon$ and the function $K_P(b)$ defined by (38) is convex and monotone decreasing with respect to $b$ $(> \frac{C_2}{(\epsilon - C_3)P} > 0)$.*

(iii) [Minimization of SFO complexity] *The SFO complexity defined by*

$$N_P = K_P(b)b = \frac{C_1 Pb^2}{(\epsilon - C_3)Pb - C_2} \tag{39}$$

*is convex with respect to $b$ $(> \frac{C_2}{(\epsilon - C_3)P} > 0)$. The batch size*

$$b_P^\star := \frac{2C_2}{(\epsilon - C_3)P}$$

*attains the following minimum value of $N_P$ defined by (39):*

$$N_P^\star = K_P(b_P^\star)b_P^\star = \frac{4C_1C_2}{(\epsilon - C_3)^2 P}. \tag{40}$$

Let $\boldsymbol{\theta}_1^{(i)} = \boldsymbol{\theta}_1$ $(i \in [P])$. Then, the SFO complexity $N^\star = K(b^\star)b^\star$ for unparallelizing SGD with the batch size $b^\star = \frac{2C_2}{\epsilon - C_3}$ is obtained by (40) with $P = 1$ (see also Theorem 3.1(iii)), i.e.,

$$N^\star = N_1^\star = K_1(b^\star)b^\star = \frac{4C_1C_2}{(\epsilon - C_3)^2}.$$

Meanwhile, Parallelizing SGD (37) with the batch size $b_P^\star = \frac{2C_2}{(\epsilon - C_3)P}$, where $P$ $(> 1)$ is the number of machines, has SFO complexity (40), i.e.,

$$N_P^\star = \frac{4C_1C_2}{(\epsilon - C_3)^2 P} < \frac{4C_1C_2}{(\epsilon - C_3)^2} = N^\star.$$

Therefore, we can conclude that the larger $P$ is, the smaller SFO complexity becomes.

*Proof of Theorem A.1.* (i) Let $k \in \mathbb{N}$. The definition of $\boldsymbol{\theta}_{k+1}^{(i)} := \boldsymbol{\theta}_k - \alpha\nabla f_{B_k^{(i)}}(\boldsymbol{\theta}_k)$ ensures that, for all $i \in [P]$,

$$\|\boldsymbol{\theta}_{k+1}^{(i)} - \boldsymbol{\theta}^\star\|^2 = \|\boldsymbol{\theta}_k - \boldsymbol{\theta}^\star\|^2 - 2\alpha(\boldsymbol{\theta}_k - \boldsymbol{\theta}^\star)^\top\nabla f_{B_k^{(i)}}(\boldsymbol{\theta}_k) + \alpha^2\|\nabla f_{B_k^{(i)}}(\boldsymbol{\theta}_k)\|^2,$$

which implies that

$$\mathbb{E}\left[\|\boldsymbol{\theta}_{k+1}^{(i)} - \boldsymbol{\theta}^\star\|^2\Big|\boldsymbol{\theta}_k\right]$$

$$= \|\boldsymbol{\theta}_k - \boldsymbol{\theta}^\star\|^2 - 2\alpha\mathbb{E}\left[(\boldsymbol{\theta}_k - \boldsymbol{\theta}^\star)^\top\nabla f_{B_k^{(i)}}(\boldsymbol{\theta}_k)\Big|\boldsymbol{\theta}_k\right] + \alpha^2\mathbb{E}\left[\|\nabla f_{B_k^{(i)}}(\boldsymbol{\theta}_k)\|^2\Big|\boldsymbol{\theta}_k\right]$$

$$= \|\boldsymbol{\theta}_k - \boldsymbol{\theta}^\star\|^2 - 2\alpha(\boldsymbol{\theta}_k - \boldsymbol{\theta}^\star)^\top\mathbb{E}\left[\nabla f_{B_k^{(i)}}(\boldsymbol{\theta}_k)\Big|\boldsymbol{\theta}_k\right] + \alpha^2\mathbb{E}\left[\|\nabla f_{B_k^{(i)}}(\boldsymbol{\theta}_k)\|^2\Big|\boldsymbol{\theta}_k\right].$$

Hence, (C2) implies that, for all $i \in [P]$,

$$
\begin{aligned}
&\mathbb{E}\left[\|\boldsymbol{\theta}_{k+1}^{(i)} - \boldsymbol{\theta}^\star\|^2\right] \\
&= \mathbb{E}\left[\|\boldsymbol{\theta}_k - \boldsymbol{\theta}^\star\|^2\right] - 2\alpha\mathbb{E}\left[(\boldsymbol{\theta}_k - \boldsymbol{\theta}^\star)^\top \nabla f(\boldsymbol{\theta}_k)\right] + \alpha^2 \mathbb{E}\left[\mathbb{E}\left[\|\nabla f_{B_k^{(i)}}(\boldsymbol{\theta}_k)\|^2 \Big| \boldsymbol{\theta}_k\right]\right].
\end{aligned}
\tag{41}
$$

A discussion similar to the one showing (14) implies that, for all $i \in [P]$,

$$
\begin{aligned}
\mathbb{E}\left[\|\nabla f_{B_k^{(i)}}(\boldsymbol{\theta}_k)\|^2 \Big| \boldsymbol{\theta}_k\right] &= \mathbb{E}\left[\|\nabla f_{B_k^{(i)}}(\boldsymbol{\theta}_k) - \nabla f(\boldsymbol{\theta}_k) + \nabla f(\boldsymbol{\theta}_k)\|^2 \Big| \boldsymbol{\theta}_k\right] \\
&= \mathbb{E}\left[\|\nabla f_{B_k^{(i)}}(\boldsymbol{\theta}_k) - \nabla f(\boldsymbol{\theta}_k)\|^2 \Big| \boldsymbol{\theta}_k\right] + \mathbb{E}\left[\|\nabla f(\boldsymbol{\theta}_k)\|^2 \Big| \boldsymbol{\theta}_k\right] \\
&\quad + 2\mathbb{E}\left[(\nabla f_{B_k^{(i)}}(\boldsymbol{\theta}_k) - \nabla f(\boldsymbol{\theta}_k))^\top \nabla f(\boldsymbol{\theta}_k) \Big| \boldsymbol{\theta}_k\right] \\
&= \mathbb{E}\left[\|\nabla f_{B_k^{(i)}}(\boldsymbol{\theta}_k) - \nabla f(\boldsymbol{\theta}_k)\|^2 \Big| \boldsymbol{\theta}_k\right] + \|\nabla f(\boldsymbol{\theta}_k)\|^2.
\end{aligned}
$$

The definition of $\boldsymbol{\theta}_k$ and the linearity of $\nabla f$ guarantee that

$$
\begin{aligned}
&\mathbb{E}\left[\|\nabla f_{B_k^{(i)}}(\boldsymbol{\theta}_k) - \nabla f(\boldsymbol{\theta}_k)\|^2 \Big| \boldsymbol{\theta}_k\right] \\
&= \mathbb{E}\left[\left\|\frac{1}{b}\sum_{j\in[b]}\left(\mathsf{G}_{\xi_{k,j}^{(i)}}(\boldsymbol{\theta}_k) - \nabla f(\boldsymbol{\theta}_k)\right)\right\|^2 \Bigg| \boldsymbol{\theta}_k\right] \\
&= \frac{1}{b^2}\mathbb{E}\left[\left\|\sum_{j\in[b]}\left(\mathsf{G}_{\xi_{k,j}^{(i)}}\left(\frac{1}{P}\sum_{i\in[P]}\boldsymbol{\theta}_k^{(i)}\right) - \nabla f\left(\frac{1}{P}\sum_{i\in[P]}\boldsymbol{\theta}_k^{(i)}\right)\right)\right\|^2 \Bigg| \boldsymbol{\theta}_k\right] \\
&= \frac{1}{P^2 b^2}\mathbb{E}\left[\left\|\sum_{j\in[b]}\sum_{i\in[P]}\left(\mathsf{G}_{\xi_{k,j}^{(i)}}(\boldsymbol{\theta}_k^{(i)}) - \nabla f(\boldsymbol{\theta}_k^{(i)})\right)\right\|^2 \Bigg| \boldsymbol{\theta}_k\right],
\end{aligned}
$$

which, together with (C2), in turn implies that

$$
\mathbb{E}\left[\|\nabla f_{B_k^{(i)}}(\boldsymbol{\theta}_k) - \nabla f(\boldsymbol{\theta}_k)\|^2 \Big| \boldsymbol{\theta}_k\right] \leq \frac{Pb\sigma^2}{P^2 b^2} = \frac{\sigma^2}{Pb}.
$$

Hence, from (A1),

$$
\mathbb{E}\left[\|\nabla f_{B_k^{(i)}}(\boldsymbol{\theta}_k)\|^2\right] \leq \frac{\sigma^2}{Pb} + G^2.
\tag{42}
$$

Accordingly, from (41) and (42), for all $i \in [P]$,

$$
\mathbb{E}\left[\|\boldsymbol{\theta}_{k+1}^{(i)} - \boldsymbol{\theta}^\star\|^2\right] \leq \mathbb{E}\left[\|\boldsymbol{\theta}_k - \boldsymbol{\theta}^\star\|^2\right] - 2\alpha\mathbb{E}\left[(\boldsymbol{\theta}_k - \boldsymbol{\theta}^\star)^\top \nabla f(\boldsymbol{\theta}_k)\right] + \alpha^2\left(\frac{\sigma^2}{Pb} + G^2\right).
$$

Since the convexity of $\|\cdot\|^2$ and the definition of $\boldsymbol{\theta}_k$ imply that

$$
\mathbb{E}\left[\|\boldsymbol{\theta}_{k+1} - \boldsymbol{\theta}^\star\|^2\right] \leq \frac{1}{P}\sum_{i\in[P]}\mathbb{E}\left[\|\boldsymbol{\theta}_{k+1}^{(i)} - \boldsymbol{\theta}^\star\|^2\right],
$$

we have that

$$
\mathbb{E}\left[\|\boldsymbol{\theta}_{k+1} - \boldsymbol{\theta}^\star\|^2\right] \leq \mathbb{E}\left[\|\boldsymbol{\theta}_k - \boldsymbol{\theta}^\star\|^2\right] - 2\alpha\mathbb{E}\left[(\boldsymbol{\theta}_k - \boldsymbol{\theta}^\star)^\top \nabla f(\boldsymbol{\theta}_k)\right] + \alpha^2\left(\frac{\sigma^2}{Pb} + G^2\right).
$$

Hence, for all $K \geq 1$,

$$
\begin{aligned}
\mathrm{VI}(K, \boldsymbol{\theta}^\star) &:= \frac{1}{K}\sum_{k=1}^K \mathbb{E}\left[(\boldsymbol{\theta}_k - \boldsymbol{\theta}^\star)^\top \nabla f(\boldsymbol{\theta}_k)\right] \leq \frac{\mathbb{E}\left[\|\boldsymbol{\theta}_1 - \boldsymbol{\theta}^\star\|^2\right]}{2\alpha K} + \frac{\alpha}{2}\left(\frac{\sigma^2}{Pb} + G^2\right) \\
&\leq \frac{\sum_{i\in[P]}\mathbb{E}[\|\boldsymbol{\theta}_1^{(i)} - \boldsymbol{\theta}^\star\|^2]}{2\alpha P}\frac{1}{K} + \frac{\sigma^2\alpha}{2}\frac{1}{Pb} + \frac{G^2\alpha}{2} \\
&\leq \underbrace{\frac{\max_{i\in[P]}\mathbb{E}[\|\boldsymbol{\theta}_1^{(i)} - \boldsymbol{\theta}^\star\|^2]}{2\alpha}}_{C_1}\frac{1}{K} + \underbrace{\frac{\sigma^2\alpha}{2}\frac{1}{Pb}}_{C_2} + \underbrace{\frac{G^2\alpha}{2}}_{C_3}.
\end{aligned}
$$

Since $f$ is convex, we have that

$$\mathbb{E}\left[f\left(\frac{1}{K}\sum_{k=1}^{K}\boldsymbol{\theta}_k\right)\right] - f(\boldsymbol{\theta}^\star) \leq \frac{1}{K}\sum_{k=1}^{K}\mathbb{E}\left[f(\boldsymbol{\theta}_k) - f(\boldsymbol{\theta}^\star)\right] \leq \mathrm{VI}(K, \boldsymbol{\theta}^\star).$$

(ii) A discussion similar to the one showing (ii) in Theorem 3.1 for SGD would show (ii) in Theorem A.1.

(iii) An argument similar to that which obtained (iii) in Theorem 3.1 for SGD would prove (iii) in Theorem A.1. □

### A.9 ADDITIONAL NUMERICAL RESULTS

Table 3: Elapsed time and training accuracy of SGD when $f(\boldsymbol{\theta}_K) \leq 10^{-1}$ for training ResNet-20 on CIFAR-10

| | SGD | | | | | | | |
|---|---|---|---|---|---|---|---|---|
| Batch Size | $2^2$ | $2^3$ | $2^4$ | $2^5$ | $2^6$ | $2^7$ | $2^8$ | $2^9$ |
| Time (s) | 16983.64 | 9103.76 | 6176.19 | 3759.25 | — | — | — | — |
| Accuracy (%) | 96.75 | 96.69 | 96.66 | 96.88 | — | — | — | — |

Table 4: Elapsed time and training accuracy of Momentum when $f(\boldsymbol{\theta}_K) \leq 10^{-1}$ for training ResNet-20 on CIFAR-10

| | Momentum | | | | | | | |
|---|---|---|---|---|---|---|---|---|
| Batch Size | $2^2$ | $2^3$ | $2^4$ | $2^5$ | $2^6$ | $2^7$ | $2^8$ | $2^9$ |
| Time (s) | 7978.90 | 3837.72 | 2520.82 | 1458.70 | 887.01 | 678.66 | 625.10 | 866.65 |
| Accuracy (%) | 96.49 | 96.79 | 96.51 | 96.72 | 96.70 | 96.94 | 96.94 | 98.34 |

Table 5: Elapsed time and training accuracy of Adam when $f(\boldsymbol{\theta}_K) \leq 10^{-1}$ for training ResNet-20 on CIFAR-10

| | Adam | | | | | | | |
|---|---|---|---|---|---|---|---|---|
| Batch Size | $2^2$ | $2^3$ | $2^4$ | $2^5$ | $2^6$ | $2^7$ | $2^8$ | $2^9$ |
| Time (s) | 10601.78 | 4405.73 | 2410.28 | 1314.01 | 617.14 | 487.75 | 281.74 | 225.03 |
| Accuracy (%) | 96.46 | 96.38 | 96.65 | 96.53 | 96.43 | 96.68 | 96.58 | 96.72 |

Table 6: Elapsed time and training accuracy of Adam when $f(\boldsymbol{\theta}_K) \leq 10^{-1}$ for training ResNet-20 on CIFAR-10

| | Adam | | | | | | |
|---|---|---|---|---|---|---|---|
| Batch Size | $2^{10}$ | $\mathbf{2^{11}}$ | $2^{12}$ | $2^{13}$ | $2^{14}$ | $2^{15}$ | $2^{16}$ |
| Time (s) | 197.78 | 195.40 | 233.70 | 349.81 | 691.04 | 644.19 | 1148.68 |
| Accuracy (%) | 96.74 | 97.21 | 97.54 | 97.75 | 97.51 | 99.05 | 99.03 |

Table 7: Elapsed time and training accuracy of SGD when $f(\boldsymbol{\theta}_K) \leq 10^{-1}$ for training ResNet-18 on MNIST

| | SGD | | | | | | | |
|---|---|---|---|---|---|---|---|---|
| Batch Size | $2^2$ | $2^3$ | $2^4$ | $2^5$ | $2^6$ | $2^7$ | $2^8$ | $2^9$ |
| Time (s) | 1620.98 | 1002.89 | 1140.47 | 1298.52 | — | — | — | — |
| Accuracy (%) | 99.70 | 99.69 | 99.70 | 99.70 | — | — | — | — |

Table 8: Elapsed time and training accuracy of Momentum when $f(\boldsymbol{\theta}_K) \leq 10^{-1}$ for training ResNet-18 on MNIST

| | Momentum | | | | | | | |
|---|---|---|---|---|---|---|---|---|
| Batch Size | $2^2$ | $2^3$ | $2^4$ | $2^5$ | $2^6$ | $2^7$ | $2^8$ | $2^9$ |
| Time (s) | 1949.66 | 550.28 | 298.72 | 238.66 | 268.98 | 362.89 | 567.54 | — |
| Accuracy (%) | 99.68 | 99.65 | 99.68 | 99.69 | 99.71 | 99.70 | 99.72 | — |

Table 9: Elapsed time and training accuracy of Adam when $f(\boldsymbol{\theta}_K) \leq 10^{-1}$ for training ResNet-18 on MNIST

| | Adam | | | | | | | |
|---|---|---|---|---|---|---|---|---|
| Batch Size | $2^2$ | $2^3$ | $2^4$ | $2^5$ | $2^6$ | $2^7$ | $2^8$ | $2^9$ |
| Time (s) | — | 7514.43 | 614.27 | 297.97 | 216.10 | 160.93 | 127.97 | 102.98 |
| Accuracy (%) | — | 99.74 | 99.66 | 99.64 | 99.66 | 99.66 | 99.69 | 99.71 |

Table 10: Elapsed time and training accuracy of Adam when $f(\boldsymbol{\theta}_K) \leq 10^{-1}$ for training ResNet-18 on MNIST

| | Adam | | | | |
|---|---|---|---|---|---|
| Batch Size | $\mathbf{2^{10}}$ | $2^{11}$ | $2^{12}$ | $2^{13}$ | $2^{14}$ |
| Time (s) | 93.53 | 100.22 | 129.29 | 217.23 | 375.87 |
| Accuracy (%) | 99.68 | 99.67 | 99.67 | 99.70 | 99.69 |

