# OpenReview forum: "Critical Batch Size Minimizes Stochastic First-Order Oracle Complexity of Deep Learning Optimizer using Hyperparameters Close to One"
_ICLR.cc/2023/Conference — Submitted to ICLR 2023_

### Official Review · Reviewer_vE9x · 2022-10-19

**Confidence:** 3
**Correctness:** 3
**Technical Novelty And Significance:** 4
**Empirical Novelty And Significance:** 4
**Recommendation:** 8

**Clarity, Quality, Novelty And Reproducibility:**

I think the paper is novel and written clearly. All the theories are reproducible. I would like to see the authors' response to my questions above to judge its quality/correctness.

**Strength And Weaknesses:**

I think this paper treats the theory seriously and gives thorough proof in all details, on a fundamental problem in machine learning. The algorithm covers the most widely used optimization algorithms with tuning parameters such as momentum, adaptive gradient, etc. so is very practical. I have a few questions below.

Critical: This paper proposes the upper bound and minimizes this upper bound wrt batch size. However, I’m not sure if the upper bound of VI is tight, say, whether it’s also a lower bound? Basically you want to minimize the exact rate (in big O) or at least a somehow tight upper bound, rather than a loose one. Does the experiment result exactly align with the upper bound? A figure with both experimental and theoretical "iteration vs. error" curves would justify the point.

But on the other hand, I think the bound makes some sense. The $1/K$ term is the bias and the $1/b$ term is the variance, but I'm not sure they are exact like both a plain inverse, in big O sense. With the same decomposition of bias/variance, does the bound, by any chance, match the bound for Table 1 of https://www.jmlr.org/papers/volume18/16-595/16-595.pdf? It should also be convincing if the bounds matches previous works' results.

I would like to see more justification of SFO as a metric. For example, the gradient computation in batches can be parallel, thus intrinsically faster than running more GD iterations that cannot be concurrent. I think it also depends on how many machines you have, so that, in terms of experiments, I like the ones with SFO as y-axis rather than the tables using time as a metric (computation time really depends on the device, like how many computational “units” it has).

Are the step sizes among different batch sizes the same? And within the algorithm on a same batch size, is the step size fixed? For the latter question, I think ADAM can adapt the step size by itself but one can still vary batch size along the iterations to get a better result. Intuitively, when batch size is smaller, the noise is bigger and one needs smaller steps sizes for the same convergence error target $\epsilon$. A diminishing step size can handle/mitigate the error caused by noise variance, i.e., caused by finite batch sizes, somehow.



**Summary Of The Paper:**

This paper analyzes the batch size for fastest convergence of optimization algorithms applied in deep learning. The metric is the total number of stochastic gradient estimations, which is the product of number of iterations and batch size. The optimal point should be in the middle, that both number of iterations and batch size are reasonably big. The proof is given in details and the experiments verify the conclusion.

**Summary Of The Review:**

I would like to see the authors' response to my questions above to judge its quality/correctness. If they are addressed, I believe the paper is sound and solid. I think the score is 5 or 6 but I feel/expect the above questions can be addressed later and give a 6 here.

=======================

I still have some questions about the tightness but I think it matches the intuition and some other results. But given the proof technique for diminishing step sizes, I would like to tentatively rescore it as 8 (I would say 7 if the tightness issue still remains, and I'd suggest more reasoning about the SFO regarding the computational cost of "parallelizable" or "unparallelizable" steps). I believe diminishing step size is more important than constant step size if the bound is more optimal, since it can also achieve any small error for a non-smooth function and tweak the variance, etc., so I'd like to see it in the paper or appendix.

---

> ### Author Response · Authors · 2022-11-07
> **Replies to Reviewer vE9x's Main comments**
>
> **Comment 1:**
> This paper proposes the upper bound and minimizes this upper bound wrt batch size. ...
>
> **Reply:**
> The constants $C_1$ and $C_3$ in the theoretical upper bound of $\mathrm{VI}(K,\theta)$ depend on unknown parameters $G$ and $D(\theta)$, and the number of iteration $K$ is defined by
> $K = \frac{C_1 b}{(\epsilon - C_3)b - C_2}$.
> Hence, it would be difficult to plot "iteration $K$ vs. error."
> Meanwhile, the evaluation of $C_2$ does not use the Cauchy--Schwarz inequality and the triangle inequality.
> Hence, the value of $C_2$ will be authentic.
> Indeed, we can estimate appropriate batch sizes from $b^\star > \frac{2 C_2}{\epsilon}$.
>
> **Comment 2:**
> But on the other hand, I think the bound makes some sense. ...
>
> **Reply:**
> Table 1 in the revised manuscript summarizes the previous results for upper bounds for the existing algorithms.
> For example,
> Theorem 1 (Zaheer et al., 2018) and the proof of Theorem 1 in (Zaheer et al., 2018) show that, under the condition that $\nabla f$ is Lipschitz continuous with the Lipschitz constant $L$, Adam using $\alpha = \mathcal{O}(1/L)$ and $\beta_1 = 0$ satisfies
> \begin{align*}
> \frac{1}{K} \sum_{k=1}^K E \left[ \|\nabla f (\theta_k)\|^2 \right]
> \leq 2 \left(\sqrt{\beta_2} G + \epsilon \right)
> \left(
> \frac{f(\theta_1) - f (\theta^\star)}{\alpha K}+\left(\frac{G\sqrt{1-\beta_2}}{\epsilon^2}+ \frac{L \alpha}{2 \epsilon^2}
> \right)
> \frac{\sigma^2}{b}
> \right),
> \end{align*}
> that is, the upper bound of Adam with $\beta_1 = 0$ is
> \begin{align*}
> O\left( \frac{1}{K} + \frac{1}{b}  \right).
> \end{align*}
> Meanwhile, our result shows that the upper bound of Adam with $\beta_1 \in (0,1)$ is
> \begin{align*}
> O\left( \frac{1}{K} + \frac{1}{b}  \right) + h(\beta_1),
> \end{align*}
> where $h$ is a monotone decreasing function with respect to $\beta_1$.
> Hence, our bound matches the previous one
> (Zaheer et al., 2018) when $h(\beta_1) \approx 0$.
>
> We also check that mini-batch tail-averaged SGD with a constant learning rate $\alpha$ depending on $b$
> and the largest eigenvalue of the Hessian of $f$ satisfies
> \begin{align*}
> E [f(\bar{\theta_K})] - f(\theta^\star)
> \leq
> \frac{2(1-\alpha \mu)^s}{\alpha^2 \mu^2 (\frac{n}{b} -s)^2}
> (f(\theta_0) - f(\theta^\star))+\frac{4 \sigma^2}{b(\frac{n}{b} -s)},
> \end{align*}
> where $\mu > 0$ is the smallest eigenvalue of the Hessian of $f$, $s$ is the initial iterations, and $n$ is the total samples (Theorem 1 in
> https://www.jmlr.org/papers/volume18/16-595/16-595.pdf ).
> The upper bound of mini-batch tail-averaged SGD when $K := \frac{n}{b} -s$ is
> \begin{align*}
> O\left( \frac{1}{K^2} + \frac{1}{K b}  \right).
> \end{align*}
>
> Please see also Table 1 in the revised manuscript.
>
> **Comment 3:**
> I would like to see more justification of SFO as a metric. ...
>
> **Reply:**
> Thank you for your suggestion. We are interested in evaluating SFO in parallel.
> However, it would be difficult from our current experiment environments.
>
> **Comment 4:**
> Are the step sizes among different batch sizes the same? ...
>
> **Reply:**
> The lower bound $b^*$ of the critical batch size $b^\star$ for Adam is
> \begin{align*}
> b^* = \frac{\sigma^2 \alpha}{\sqrt{v^*} \beta_1 (1-\beta_1) \epsilon}.
> \end{align*}
> Hence, for the same convergence error $\epsilon$, the smaller $b^*$ is, the bigger $\sigma^2$ becomes and the smaller $\alpha$ becomes.
>
> Our goal is to estimate appropriate batch sizes from the formula for $b^*$ before implementing optimizers.
> The definition of $b^*$ uses $\alpha$ and $\beta_1$.
> Hence, we must set $\alpha$ and $\beta_1$ before implementing optimizers.
> Here, we use well-known parameters $\alpha = 10^{-3}$ and $\beta_1 = 0.9$.
> Then, we estimate appropriate batch sizes using fixed $\alpha$ and $\beta_1$ and $b^*$ before implementing optimizers.
> As a result, we would like to check whether or not actual critical batch sizes (obtained from numerical experiments for optimizers using $\alpha = 10^{-3}$ and $\beta_1 = 0.9$) are close estimated batch sizes (using $\alpha = 10^{-3}$ and $\beta_1 = 0.9$).

---

> > ### Comment · Reviewer_vE9x · 2022-11-07
> > **My takeaway**
> >
> > 1. I think you meant *the optimal $K$*, or $K^*$ depends on blabla, but I meant you can plot error versus $K$, i.e. $f(K) = C_1/K + C_2/b + C_3$ vs $K$. Could you explain the sentence "Meanwhile..."? Isn't $C_2$ defined in your original Table 1 or revised Table 2?
> >
> > 2. I did not find the comment regarding whether the error is a lower bound or tight upper bound, but since it aligns with other results, it would make sense.
> >
> > 3. I would like to see more comment on whether you can choose a diminishing step size to mitigate the variance caused by finite batch size. Since $C_1 = O(1/\alpha)$ and $C_2 = O(\alpha)$, I would guess it reflects the point that you should choose a large $\alpha$ at the beginning when $K$ is small, and decrease $\alpha$ later when a large $K$ can "counter-effect" a small $\alpha$ that makes the term $C_2/b$ small.

---

> > > ### Comment · Reviewer_vE9x · 2022-11-07
> > > **Another question about $C_3$**
> > >
> > > Could you explain $C_3$? Do you need $epsilon>C_3$? I'm not sure why it exists, for example, if $b$ is the same as the total number of samples, then you just run GD but not SGD. When you run GD you can always make error as small as possible right? Or is the reason that you don't have smoothness (specifically, gradient Lipschitzness with bounded $\|\nabla f(x+dx) - \nabla f(x)\| / \|dx\|$) of the function, thus a fixed step size cannot guaranteed to converge? I think this is also a reason to discuss diminishing step size.

---

> > > > ### Author Response · Authors · 2022-11-08
> > > > **Reply to comment about $C_3$**
> > > >
> > > > **Comment:**
> > > > Could you explain $C_3$? ...
> > > >
> > > > **Reply:**
> > > > Under smoothness condition of $f$, GD with a constant step size $\alpha = O(1/L)$ satisfies that $\|\nabla f(\theta_k)\| \to 0$ by the descent lemma ($f(y) \leq f(x) + \nabla f(x)^\top (y-x) + \frac{L}{2}\|y-x\|^2$).
> > > > Under non-smoothness condition of $f$, GD with a constant step size $\alpha > 0$ satisfies that
> > > > $\liminf_k \nabla f(\theta_k)^\top (\theta_k - \theta) \leq \frac{G^2}{2} \alpha$.
> > > > Hence, it is not guaranteed that, under non-smoothness condition of $f$, GD with a constant step size $\alpha > 0$ converges.
> > > > However, we can expect that using a small constant step size $\alpha$ approximates a stationary point of $f$.
> > > >
> > > > Meanwhile, under non-smoothness condition of $f$, GD with a diminishing step size $\alpha_k = 1/\sqrt{k}$ satisfies that
> > > > $\liminf_k \nabla f(\theta_k)^\top (\theta_k - \theta) \leq 0$
> > > > and
> > > > \begin{align*}
> > > > VI(K,\theta) \leq \frac{D(\theta)}{2 K \alpha_K} + \frac{\frac{\sigma^2}{b} + G^2}{2K} \sum_{k=1}^K \alpha_k
> > > > = \frac{D(\theta)}{2 K \alpha_K} + \frac{G^2}{2K} \sum_{k=1}^K \alpha_k
> > > > = O\left(\frac{1}{\sqrt{K}} \right).
> > > > \end{align*}
> > > > Hence, GD with a diminishing step size does not need $C_3$.
> > > > This result strongly depends on the condition **$\beta_1 = 0$**.
> > > >
> > > > Under non-smoothness condition of $f$, Adam with a diminishing step size $\alpha_k$ and $b = n$ (i.e., $\sigma = 0$) satisfies that
> > > > \begin{align*}
> > > > VI(K,\theta)
> > > > \leq \frac{\tilde{C_1}}{\sqrt{K}} + \frac{\tilde{C_2}}{b\sqrt{K}} + h(\beta_1)
> > > > = \frac{\tilde{C_1}}{\sqrt{K}}+ h(\beta_1)
> > > > \end{align*}
> > > > and
> > > > Adam with a constant step size $\alpha$ and $b = n$ (i.e., $\sigma = 0$) satisfies that
> > > > \begin{align*}
> > > > VI(K,\theta)
> > > > \leq \frac{C_1}{K} + \frac{C_2}{b} + C_3
> > > > = \frac{C_1}{K}+ C_3.
> > > > \end{align*}
> > > > Hence, it would be better to use a constant step size $\alpha$ for Adam in the sense of minimizing the upper bound of $VI(K,\theta)$.
> > > > The previous results (Zaheer et al., 2018) indicated that Adam with
> > > > a constant step size $\alpha = O(1/L)$, **$\beta_1 = 0$**, and $\sigma = 0$ satisfies that
> > > > \begin{align*}
> > > > \frac{1}{K} \sum_{k=1}^K \mathbb{E}\left[ \|\nabla f (\theta_k)\|^2 \right]
> > > > \leq 2 \left(\sqrt{\beta_2} G + \epsilon \right)
> > > > \frac{f(\theta_1) - f (\theta^\star)}{\alpha K}
> > > > =: \frac{\hat{C_1}}{K}.
> > > > \end{align*}
> > > > The condition **$\beta_1 \neq 0$** is essential to analyze Adam
> > > > since the practically useful $\beta_1$ is close to $1$.
> > > > The upper bound of $VI(K,\theta)$ for Adam with **$\beta_1 \neq 0$**
> > > > depends on the term with respect to $\beta_1$, i.e., $C_3 := \frac{G^2 \alpha}{2\sqrt{v_*} \beta_1 (1-\beta_1)} + h(\beta_1)$.
> > > > Hence, $\epsilon > C_3$ and $\epsilon > h(\beta_1)$ are needed to consider $K$ to satisfy
> > > > $VI(K,\theta) \leq \frac{\tilde{C_1}}{\sqrt{K}} + h(\beta_1) = \epsilon$ (diminishing step size case)
> > > > and
> > > > $VI(K,\theta) \leq \frac{C_1}{K} + C_3 = \epsilon$ (constant step size case) even if $\sigma = 0$.

---

> > > > > ### Comment · Reviewer_vE9x · 2022-11-08
> > > > > **This makes sense, thanks.**
> > > > >
> > > > > This makes sense, thanks.

---

> > > ### Author Response · Authors · 2022-11-08
> > > **Correction (Reply to Comment 2) Replies to Reviewer vE9x7's comments**
> > >
> > > **Comment 1:**
> > > I think you meant the optimal ...
> > >
> > > **Reply:**
> > > We use the Cauchy-Schwarz inequality, (A1), and (A2) to obtain $C_1$ and $C_3$. Meanwhile, we do not use them to obtain $C_2$. Hence, we commented that $C_2$ will be more authentic than $C_1$ and $C_3$.
> > > We are sorry for the confusion.
> > > $C_2$ is defined in Table 1 (previous manuscript) or Table 2 (revised manuscript).
> > >
> > > **Comment 2:**
> > > I did not find the comment regarding whether the error is a lower bound or tight upper bound ...
> > >
> > > **Reply:**
> > > We evaluate $VI(K,\theta) \leq C_1/K+C_2/b+C_3 =  \epsilon$.
> > > When $C_1/K+C_2/b+C_3 \leq \epsilon$, we have that
> > > $K \leq \frac{C_1 b}{(\epsilon - C_3)b - C_2}$.
> > > So, we think that the error that the reviewer points out is a lower bound.
> > >
> > > **We have a mistake.  $C_1/K+C_2/b+C_3 \leq \epsilon$ implies that $K \geq \frac{C_1 b}{(\epsilon - C_3)b - C_2}$.**
> > >
> > > **Comment 3:**
> > > I would like to see more comment on whether you can choose a diminishing step size to mitigate the variance caused by finite batch size. ...
> > >
> > > **Reply:**
> > > We can establish the upper bound of $VI(K,\theta)$ for Adam with a diminishing step size such as $\alpha_k = 1/\sqrt{k}$ from a discussion similar to the one for proving Theorem 3.1(i).
> > > If $(\alpha_k)$ is monotone decreasing, then we have
> > > \begin{align*}
> > > VI(K,\theta)\leq \frac{d D(\theta) \sqrt{M}}{2 \beta_1  \sqrt{1 - \beta_{2}} \alpha_K K}+\frac{\frac{\sigma^2}{b} + G^2}{2 \sqrt{v_*}  \beta_1 (1 - \beta_1) K}\sum_{k=1}^K \alpha_k \sqrt{1 - {\beta}_{2}^{k+1}} + h(\beta_1),
> > > \end{align*}
> > > where $h(\beta_1)$ is defined as in (7) (revised).
> > >
> > > If $\alpha_k = \alpha$, then we have (6) (revised).
> > > When $\alpha_k = 1/\sqrt{k}$ is used, we have
> > > \begin{align*}
> > >         VI(K,\theta)
> > >         \leq
> > >         \frac{d D(\theta) \sqrt{M}}{2 \beta_1  \sqrt{1 - \beta_{2}} \sqrt{K}}
> > >         +
> > >         \frac{\frac{\sigma^2}{b} + G^2}{2 \sqrt{v_*}  \beta_{1} (1 - \beta_1) K}
> > >         \sum_{k=1}^K \frac{1}{\sqrt{k}}
> > >         + h(\beta_1)
> > > \end{align*}
> > > \begin{align*}
> > >        \leq
> > >         \frac{d D(\theta) \sqrt{M}}{2 \beta_1  \sqrt{1 - \beta_{2}} \sqrt{K}}
> > >         +
> > >         \frac{\frac{\sigma^2}{b} + G^2}{\sqrt{v_*}  \beta_{1} (1 - \beta_1)}\frac{\sqrt{K}}{K}
> > >         + h(\beta_1)
> > > \end{align*}
> > > \begin{align*}
> > >         \leq
> > >         \frac{d D(\theta) \sqrt{M}}{2 \beta_1  \sqrt{1 - \beta_{2}} \sqrt{K}}
> > >         +
> > >         \frac{\frac{\sigma^2}{b} + G^2}{\sqrt{v_*}  \beta_{1} (1 - \beta_1)}\frac{1}{\sqrt{K}}
> > >         + h(\beta_1),
> > > \end{align*}
> > > where $1 - \beta_2^{k+1} \leq 1$ and $\sum_{k=1}^K \frac{1}{\sqrt{k}} \leq 2 \sqrt{K}$.
> > > Hence,
> > > \begin{align*}
> > >         VI(K,\theta)
> > >         \leq
> > >         \left(\frac{d D(\theta) \sqrt{M}}{2 \beta_1  \sqrt{1 - \beta_{2}}}
> > >         +
> > >         \frac{G^2}{\sqrt{v_*}  \beta_{1} (1 - \beta_1)} \right)
> > >         \frac{1}{\sqrt{K}}
> > >         +
> > >         \frac{\sigma^2}{\sqrt{v_*}  \beta_{1} (1 - \beta_1)}\frac{1}{b\sqrt{K}}
> > >         + h(\beta_1)
> > > \end{align*}
> > > \begin{align*}
> > >         =: \frac{\tilde{C_1}}{\sqrt{K}} + \frac{\tilde{C_2}}{b\sqrt{K}} + h(\beta_1).
> > > \end{align*}
> > > Here, we would like to recall that, when a constant step size $\alpha$ is used, we have that
> > > \begin{align*}
> > > VI(K,\theta) \leq \frac{C_1}{K} + \frac{C_2}{b} + C_3.
> > > \end{align*}
> > > Hence, we guess that using a diminishing step size mitigates the variance (the term $\frac{\tilde{C_2}}{b\sqrt{K}}$) in contrast to the term $\frac{C_2}{b}$ using a constant step size $\alpha$.
> > > However, the bias term $\frac{C_1}{K}$ using a constant step size $\alpha$ would be better than
> > > the bias term $\frac{\tilde{C_1}}{\sqrt{K}}$ using a diminishing step size in the sense of minimizing the upper bound of $VI(K,\theta)$.

---

> > > > ### Comment · Reviewer_vE9x · 2022-11-08
> > > > **Follow-up comment**
> > > >
> > > > 1. Lower bound issue: So we have $K \le \frac{C_1b}{(\epsilon - C_3)b - C_2}$. Then what would be the "critical batch size" that you call as $b^*$, would the $b^* = ...$ become $b^* \le...$ or $b^* \ge...$ rather than $b^* = ...$, and would it be tight? If that's not tight, perhaps I wouldn't choose exactly the right hand side as optimal batch size.
> > > >
> > > > 2. Regarding diminishing step size: I feel the derivation here makes a lot of sense and I would definitely recommend the authors to put it in appendix if not the main paper. $\alpha_k = 1/\sqrt{k}$ is not the only way of choosing step size, for example one can have a step size $\alpha$ for a few iterations, then step size $\alpha/2$ for the next few iterations... The point is to target on bias at the beginning and on variance later. The tradeoff exactly comes from the last sentence of authors' comment "However, the bias term...". One can vary both batch size and step size during training and find the best tradeoffs. But there is one more "time varying" dimension and makes the proof harder so it would be okay to treat it as future work.
> > > >
> > > > If point 1 and 2 can be solved, I would score the paper as 8-10. Now I would suggest a 6, but if $b^*$ bound is tight then I would score it as 7.

---

> > > > > ### Author Response · Authors · 2022-11-09
> > > > > **Reply to Reviewer vE9x's comment 1**
> > > > >
> > > > > **Comment 1:**
> > > > > Lower bound issue
> > > > >
> > > > > **Reply:**
> > > > > Let us consider SGD.
> > > > > Let $\theta^*$ be a stationary point $f$. From Lemma A.2 (Page 12, revised), we have that
> > > > > \begin{align*}
> > > > > 2 \alpha E[(\theta_k - \theta^* )^\top \nabla f(\theta_k)] = E[\| \theta_k - \theta^* \|^2]-E[\| \theta_{k+1} - \theta^* \|^2]
> > > > > +\alpha^2 E[\|\nabla f_{B_k} (\theta_k)\|^2].
> > > > > \end{align*}
> > > > > Hence,
> > > > > \begin{align*}
> > > > > 2 \alpha \sum_{k=1}^K E[(\theta_k - \theta^*)^\top \nabla f(\theta_k)]
> > > > > = E[\| \theta_1 - \theta^* \|^2]-E[\| \theta_{K+1} - \theta^* \|^2]
> > > > > +\alpha^2 \sum_{k=1}^K E[\|\nabla f_{B_k} (\theta_k)\|^2].
> > > > > \end{align*}
> > > > > Here, we assume that there exists $c \in [0,1]$ such that
> > > > > $c \sigma^2 \leq E[\|G_{\xi_k}(\theta_k) - \nabla f(\theta_k)\|^2] \leq \sigma^2$ (see (C2)).
> > > > > From (12) (revised), we have that
> > > > > \begin{align*}
> > > > > E[\|\nabla f_{B_k}(\theta_k)\|^2|\theta_k]
> > > > > = E[\|\nabla f_{B_k}(\theta_k) - \nabla f(\theta_k)\|^2|\theta_k]+ \|\nabla f(\theta_k)\|^2
> > > > > \geq \frac{c\sigma^2}{b}.
> > > > > \end{align*}
> > > > > Since $\theta_{K+1}$ approximates $\theta^*$, we assume that
> > > > > there exists $X(\theta^*) \geq 0$ such that
> > > > > $E[\| \theta_1 - \theta^* \|^2]-E[\| \theta_{K+1} - \theta^* \|^2]\geq X(\theta^*)$.
> > > > > Hence,
> > > > > \begin{align*}
> > > > > 2 \alpha \sum_{k=1}^K E[(\theta_k - \theta^*)^\top \nabla f(\theta_k)]
> > > > > \geq X(\theta^*) + \frac{c\sigma^2 \alpha^2 K}{b},
> > > > > \end{align*}
> > > > > which implies that
> > > > > \begin{align*}
> > > > > VI(K,\theta^*) \geq
> > > > > \frac{X(\theta^*)}{2 \alpha K}+ \frac{c\sigma^2 \alpha}{2 b}.
> > > > > \end{align*}
> > > > > Accordingly,
> > > > > \begin{align*}
> > > > > K \leq \frac{D_1 b}{\epsilon b - D_2} =: \overline{K}(b),
> > > > > \end{align*}
> > > > > satisfies $VI(K,\theta^*) \geq \epsilon$,
> > > > > where
> > > > > \begin{align*}
> > > > > D_1 := \frac{X(\theta^*)}{2 \alpha} \text{ and }
> > > > > D_2 := \frac{c\sigma^2 \alpha}{2}.
> > > > > \end{align*}
> > > > > A discussion similar to the one showing the existence of $b^*$ in (5) (revised) ensures that
> > > > > \begin{align*}
> > > > > \overline{N} := \overline{K}(b) b = \frac{D_1 b^2}{\epsilon b - D_2}
> > > > > \end{align*}
> > > > > is minimized at
> > > > > \begin{align*}
> > > > > b_* := \frac{2 D_2}{\epsilon} = \frac{c\sigma^2 \alpha}{\epsilon}.
> > > > > \end{align*}
> > > > > Meanwhile, Theorem 3.1(i) and (ii) (revised) indicate that
> > > > > \begin{align*}
> > > > > K \geq \frac{C_1 b}{(\epsilon - C_3)b - C_2} =: \underline{K}(b)
> > > > > \end{align*}
> > > > > satisfies $VI(K,\theta^*) \leq \epsilon$,
> > > > > where
> > > > > \begin{align*}
> > > > > C_1 := \frac{E[\| \theta_1 - \theta^* \|^2]}{2 \alpha}, \text{ }
> > > > > C_2 := \frac{\sigma^2 \alpha}{2}, \text{ and }
> > > > > C_3 := \frac{G^2 \alpha}{2}.
> > > > > \end{align*}
> > > > > **(We have a mistake (The above reply to Comment 2).
> > > > > $C_1/K + C_2/b + C_3 \leq \epsilon$ implies that $K \geq \frac{C_1 b}{(\epsilon - C_3)b - C_2}$.)**
> > > > > Moreover, Theorem 3.1(iii) ensures that
> > > > > \begin{align*}
> > > > > \underline{N} := \underline{K}(b) b = \frac{C_1 b^2}{(\epsilon -C_3) b - C_2}
> > > > > \end{align*}
> > > > > is minimized at
> > > > > \begin{align*}
> > > > > b^* := \frac{2 C_2}{\epsilon - C_3} = \frac{\sigma^2 \alpha}{\epsilon - (G^2 \alpha/2)}.
> > > > > \end{align*}
> > > > > The number of steps $K$ defined by
> > > > > \begin{align*}
> > > > > \underline{K}(b) := \frac{C_1 b}{(\epsilon - C_3) b - C_2} \leq K \leq \frac{D_1 b}{\epsilon b - D_2} =: \overline{K}(b)
> > > > > \end{align*}
> > > > > satisfies $VI(K,\theta^*) = \epsilon$.
> > > > > Moreover, we have that
> > > > > \begin{align*}
> > > > > D_1 = \frac{X(\theta^*)}{2\alpha}\leq \frac{E[\| \theta_1 - \theta^* \|^2]-E[\| \theta_{K+1} - \theta^* \|^2]}{2\alpha} \leq
> > > > > \frac{E[\| \theta_1 - \theta^* \|^2]}{2\alpha} = C_1,
> > > > > \end{align*}
> > > > > \begin{align*}
> > > > > D_2 = \frac{c \sigma^2 \alpha}{2}
> > > > > \leq
> > > > > \frac{\sigma^2 \alpha}{2} = C_2,
> > > > > \end{align*}
> > > > > and
> > > > > \begin{align*}
> > > > > (\epsilon - C_3) b - C_2
> > > > > \leq \epsilon b - C_2
> > > > > \leq
> > > > > \epsilon b - D_2.
> > > > > \end{align*}
> > > > > Accordingly, we have that
> > > > > \begin{align*}
> > > > > \overline{K}(b) = \frac{D_1 b}{\epsilon b - D_2} \leq \underline{K}(b) = \frac{C_1 b}{(\epsilon - C_3) b - C_2} \leq K \leq \frac{D_1 b}{\epsilon b - D_2} = \overline{K}(b),
> > > > > \end{align*}
> > > > > i.e.,
> > > > > \begin{align*}
> > > > > K = \underline{K}(b) = \overline{K}(b).
> > > > > \end{align*}
> > > > > Hence, the SFO complexity $N = Kb$ satisfies
> > > > > \begin{align*}
> > > > > N = \underline{N} = \overline{N}
> > > > > \end{align*}
> > > > > and $N$ is minimized at $b_* = b^*$.
> > > > > Therefore, $b^*$ defined by (5) (revised) coincides with the critical batch size in the sense of minimizing $N = Kb$, where $K$ satisfies $VI(K,\theta^*) = \epsilon$.

---

> > > > > > ### Author Response · Authors · 2022-11-09
> > > > > > **Reply to Reviewer vE9x's comment 2**
> > > > > >
> > > > > > **Comment 2:**
> > > > > > Regarding diminishing step size
> > > > > >
> > > > > > **Reply:**
> > > > > > If $(\alpha_k)$ is monotone decreasing, then we have
> > > > > > \begin{align*}
> > > > > >         VI(K,\theta)
> > > > > >         \leq
> > > > > >         \frac{d D(\theta) \sqrt{M}}{2 \beta_1  \sqrt{1 - \beta_{2}} \alpha_K K}
> > > > > >         +
> > > > > >         \frac{\frac{\sigma^2}{b} + G^2}{2 \sqrt{v_*}  \beta_{1} (1 - \beta_1) K}
> > > > > >         \sum_{k=1}^K \alpha_k
> > > > > >         + h(\beta_1).
> > > > > > \end{align*}
> > > > > > Let $T \geq 1$, $\gamma \in (0,1)$, $\alpha > 0$, and $P := \frac{K}{T}$.
> > > > > > We define
> > > > > > \begin{align*}
> > > > > > (\alpha_k) = (\alpha_T, \alpha_{2T},\ldots, \alpha_{PT}),
> > > > > > \end{align*}
> > > > > > where $\alpha_{jT} = (\gamma^{j-1} \alpha, \gamma^{j-1}\alpha, \ldots, \gamma^{j-1} \alpha)$ $(j=1,2,\ldots, P)$.
> > > > > > That is, $(\alpha_k)$ is
> > > > > > \begin{align*}
> > > > > > ((\alpha, \alpha, \ldots, \alpha), (\gamma \alpha,\gamma \alpha,\ldots, \gamma \alpha), \ldots, (\gamma^{P-1} \alpha,\gamma^{P-1} \alpha,\ldots, \gamma^{P-1} \alpha) ).
> > > > > > \end{align*}
> > > > > > Let $\underline{\alpha} > 0$ be the lower bound of $\alpha_k$.
> > > > > > From
> > > > > > \begin{align*}
> > > > > > \sum_{k=1}^K \alpha_k
> > > > > > = \alpha T + \gamma \alpha T + \cdots + \gamma^{P-1}\alpha T
> > > > > > \leq \frac{\alpha T}{1-\gamma},
> > > > > > \end{align*}
> > > > > > we have
> > > > > > \begin{align*}
> > > > > >         VI(TP,\theta)
> > > > > >         \leq
> > > > > >         \frac{d D(\theta) \sqrt{M}}{2 \beta_1  \sqrt{1 - \beta_{2}} \underline{\alpha} T P}
> > > > > >         +
> > > > > >         \frac{\frac{\sigma^2}{b} + G^2}{2 \sqrt{v_*}  \beta_{1} (1 - \beta_1) T P}
> > > > > >         \frac{\alpha T}{1-\gamma}
> > > > > >         + h(\beta_1)
> > > > > > \end{align*}
> > > > > > \begin{align*}
> > > > > >         =
> > > > > >         \frac{d D(\theta) \sqrt{M}}{2 \beta_1  \sqrt{1 - \beta_{2}} \underline{\alpha} T P}
> > > > > >         +
> > > > > >         \frac{(\frac{\sigma^2}{b} + G^2)\alpha}{2 \sqrt{v_*}  \beta_{1} (1 - \beta_1) (1-\gamma)P}
> > > > > >         + h(\beta_1)
> > > > > > \end{align*}
> > > > > > \begin{align*}
> > > > > >         =
> > > > > >         \left(
> > > > > >         \frac{d D(\theta) \sqrt{M}}{2 \beta_1  \sqrt{1 - \beta_{2}} \underline{\alpha} T}
> > > > > >         + \frac{G^2 \alpha}{2 \sqrt{v_*}  \beta_{1} (1 - \beta_1) (1-\gamma)}
> > > > > >         \right) \frac{1}{P}
> > > > > >         +
> > > > > >         \frac{\sigma^2 \alpha}{2 \sqrt{v_*}  \beta_{1} (1 - \beta_1) (1-\gamma)} \frac{1}{P b} + h(\beta_1)
> > > > > > \end{align*}
> > > > > > \begin{align*}
> > > > > >         = O\left(\frac{1}{P} + \frac{1}{Pb}\right) + h(\beta_1).
> > > > > > \end{align*}
> > > > > > Here, we would like to recall that, when $\alpha_k$ is a constant step size $\alpha$,
> > > > > > \begin{align*}
> > > > > > VI(TP,\theta) \leq \frac{C_1}{TP} + \frac{C_2}{b} + C_3
> > > > > > = O\left( \frac{1}{P} + \frac{1}{b} \right) + C_3,
> > > > > > \end{align*}
> > > > > > and when $\alpha_k = 1/\sqrt{k}$,
> > > > > > \begin{align*}
> > > > > > VI(TP,\theta) \leq \frac{\tilde{C_1}}{\sqrt{TP}} + \frac{\tilde{C_2}}{b\sqrt{TP}} + h(\beta_1)
> > > > > > =
> > > > > > O\left( \frac{1}{\sqrt{P}} + \frac{1}{b \sqrt{P}} \right) + h(\beta_1).
> > > > > > \end{align*}
> > > > > > Therefore, using
> > > > > > \begin{align*}
> > > > > > (\alpha_k) = (\alpha_T, \alpha_{2T},\ldots, \alpha_{PT})
> > > > > > \end{align*}
> > > > > > is more desirable to minimize the upper bound of $VI(K,\theta)$
> > > > > > than using $\alpha_k = \alpha, 1/\sqrt{k}$.
> > > > > > We would like to thank Reviewer vE9x for the important and helpful comments and suggestions.
> > > > > >
> > > > > > We would like to put the above results (replies to Comments 1 and 2) in appendix/main text after discussing with Reviewer vE9x.

---

> > > > > > > ### Comment · Reviewer_vE9x · 2022-11-09
> > > > > > > **Thanks for the proof**
> > > > > > >
> > > > > > > I definitely believe the proof for both "varying step size" cases is valuable and can be placed in the paper. Though these two cases cannot cover all "varying patterns", I believe the insight is important and would like to have the two typical examples in the paper appendix.

---

> > > > > > > > ### Author Response · Authors · 2022-11-10
> > > > > > > > **Results for varying step sizes**
> > > > > > > >
> > > > > > > > Thank you very much for your valuable comments and suggestions. Newly added Subsection A.6 (Pages 20 and 21) discusses the two typical examples.

---

> > > > > > ### Comment · Reviewer_vE9x · 2022-11-09
> > > > > > **Question**
> > > > > >
> > > > > > Thanks for the detailed proof. I have a question:
> > > > > >
> > > > > > You first proved that, when $K\le\overline K(b)$, $VI \ge \epsilon$. When $K\ge \underline K(b)$, $VI \le \epsilon$. Then why can you say that
> > > > > > "The number of steps $K$ defined by $\underline K(b) \le K \le \overline K(b)$ makes $VI = \epsilon$"?
> > > > > >
> > > > > > Think that, when $K\le 1$, $VI \ge 10$; when $K\ge 100$, $VI \le 10$, then you cannot say at some $100\le K\le 1$ you have $VI = 10$.
> > > > > >
> > > > > > I think the proof that $\underline K(b) \ge \overline K(b)$ is intrinsic, but the other direction applies the above logic which might be wrong.

---

> > > > > > > ### Author Response · Authors · 2022-11-10
> > > > > > > **Reply to Question**
> > > > > > >
> > > > > > > Thank you for your comment.
> > > > > > > We consider SGD and assume that there exists $c \in [0,1]$ such that
> > > > > > > $c \sigma^2 \leq E[\|G_{\xi_k}(\theta_k) - \nabla f(\theta_k)\|^2] \leq \sigma^2$ and that there exists $X(\theta^*) \geq 0$ such that
> > > > > > > $E[\| \theta_1 - \theta^* \|^2]- E[\| \theta_{K+1} - \theta^* \|^2] \geq X(\theta^* )$.
> > > > > > > Then, we have that
> > > > > > > \begin{align*}
> > > > > > > VI(K,\theta^*) \geq \frac{D_1}{K} + \frac{D_2}{b},
> > > > > > > \end{align*}
> > > > > > > where
> > > > > > > \begin{align*}
> > > > > > > D_1 := \frac{X(\theta^*)}{2 \alpha} \text{ and }
> > > > > > > D_2 := \frac{c\sigma^2 \alpha}{2}.
> > > > > > > \end{align*}
> > > > > > > Meanwhile, Theorem 3.1(i) indicates that
> > > > > > > \begin{align*}
> > > > > > > VI(K,\theta^*) \leq \frac{C_1}{K} + \frac{C_2}{b} + C_3,
> > > > > > > \end{align*}
> > > > > > > where
> > > > > > > \begin{align*}
> > > > > > > C_1 := \frac{E[\| \theta_1 - \theta^* \|^2]}{2 \alpha}, \text{ }
> > > > > > > C_2 := \frac{\sigma^2 \alpha}{2}, \text{ and }
> > > > > > > C_3 := \frac{G^2 \alpha}{2}.
> > > > > > > \end{align*}
> > > > > > > Suppose that $VI(K,\theta^*) \leq \epsilon$.
> > > > > > > Then, we have that
> > > > > > > \begin{align*}
> > > > > > > \frac{D_1}{K} + \frac{D_2}{b} \leq \epsilon,
> > > > > > > \end{align*}
> > > > > > > which implies that
> > > > > > > \begin{align*}
> > > > > > > \overline{K}(b) := \frac{D_1 b}{\epsilon b - D_2} \leq K.
> > > > > > > \end{align*}
> > > > > > > Suppose that $VI(K,\theta^*) \geq \epsilon$.
> > > > > > > Then, we have that
> > > > > > > \begin{align*}
> > > > > > > \frac{C_1}{K} + \frac{C_2}{b} + C_3 \geq \epsilon,
> > > > > > > \end{align*}
> > > > > > > which implies that
> > > > > > > \begin{align*}
> > > > > > > \underline{K}(b) := \frac{C_1 b}{(\epsilon - C_3) b - C_2} \geq K.
> > > > > > > \end{align*}
> > > > > > > From \begin{align*}
> > > > > > > D_1 = \frac{X(\theta^* )}{2\alpha}\leq\frac{E[\| \theta_1 - \theta^* \|^2]-E[\| \theta_{K+1} - \theta^* \|^2]}{2\alpha}
> > > > > > > \leq\frac{E[\| \theta_1 - \theta^* \|^2]}{2\alpha} = C_1,
> > > > > > > \end{align*}
> > > > > > > \begin{align*}
> > > > > > > D_2 = \frac{c \sigma^2 \alpha}{2}
> > > > > > > \leq
> > > > > > > \frac{\sigma^2 \alpha}{2} = C_2,
> > > > > > > \end{align*}
> > > > > > > and
> > > > > > > \begin{align*}
> > > > > > > (\epsilon - C_3) b - C_2
> > > > > > > \leq \epsilon b - C_2
> > > > > > > \leq
> > > > > > > \epsilon b - D_2,
> > > > > > > \end{align*}
> > > > > > > we have that $\overline{K}(b) \leq \underline{K}(b)$.
> > > > > > > Hence, $K$ with $VI(K,\theta^*) = \epsilon$ satisfies
> > > > > > > \begin{align*}
> > > > > > > \overline{K}(b) \leq K \leq \underline{K}(b)
> > > > > > > \end{align*}
> > > > > > > and the SFO complexity $N = Kb$ satisfies
> > > > > > > \begin{align*}
> > > > > > > \overline{K}(b)b =:\overline{N} \leq N \leq \underline{N} := \underline{K}(b)b.
> > > > > > > \end{align*}
> > > > > > > Moreover, $b_* := \frac{2 D_2}{\epsilon}$ minimizing $\overline{N}$
> > > > > > > and $b^* := \frac{2 C_2}{\epsilon - C_3}$ minimizing $\underline{N}$
> > > > > > > satisfy that $b_* \leq b^*$.
> > > > > > > Let $\alpha$ be small enough and we consider that $\theta_{K+1}$ approximates $\theta^*$.
> > > > > > > Then, we have that
> > > > > > > \begin{align*}
> > > > > > > b_* \approx b^* \text{ and } \overline{K}(b_*)b_* \approx \underline{K}(b^*)b^*,
> > > > > > > \end{align*}
> > > > > > > where
> > > > > > > \begin{align*}
> > > > > > > C_1 = \frac{E[\| \theta_1 - \theta^* \|^2]}{2 \alpha} \approx
> > > > > > > D_1 = \frac{X(\theta^*)}{2 \alpha}, \text{ }
> > > > > > > C_2 = \frac{\sigma^2 \alpha}{2} \approx D_2 = \frac{c\sigma^2 \alpha}{2}, \text{ and }
> > > > > > > C_3 = \frac{G^2 \alpha}{2} \approx 0
> > > > > > > \end{align*}
> > > > > > > are used.
> > > > > > > Hence, the batch sizes $b_*$ and $b^*$ $(b_* \approx b^*)$
> > > > > > > approximate the batch size minimizing $N = Kb$.

---

> > > > > > > > ### Comment · Reviewer_vE9x · 2022-11-14
> > > > > > > > **This is a reasonable derivation**
> > > > > > > >
> > > > > > > > This is a reasonable derivation for the case when the function is gradient Lipschitz and the step size is small. I'm not sure about $C_3 \approx 0$ since $\epsilon$, i.e. $1/K$ and $1/b$ are both small as well. But somehow the math gives an uppler/lower error gap for the case of smooth function and constant step size, which we can refer to.

---

> > > > > > > > > ### Author Response · Authors · 2022-11-15
> > > > > > > > > **Reply to Reviewer vE9x's comment**
> > > > > > > > >
> > > > > > > > > Thank you for your comment. Newly added section A.7 discusses an upper/lower error gap for the case of smooth function and constant step size.

---

> ### Author Response · Authors · 2022-11-15
> **Reasoning about the SFO regarding the computational cost of "parallelizable" or "unparallelizable" steps**
>
> Motivated by (Jain et al., 2018),
> let us consider SGD implemented independently on $P$ machines, each of which contains $\frac{n}{P}$ samples, for minimizing the expected square loss function $f (\theta) := \frac{1}{2} E [y - x^\top \theta]$.
> Given $\theta_k \in \mathbb{R}^d$, machine $i$ $(i\in [P])$ generates the point
> \begin{align*}
> \theta_{k+1}^{(i)} := \theta_k - \alpha \nabla f_{B_k^{(i)}} (\theta_k) =
> \theta_k - \frac{\alpha}{b} \sum_{j \in [b]} G_{\xi_{k,j}^{(i)}} (\theta_k)
> \end{align*}
> using SGD with batch size $b$
> and computes
> \begin{align*}
> \theta_{k+1} := \frac{1}{P} \sum_{i=1}^P \theta_{k+1}^{(i)},
> \end{align*}
> where $\xi_{k,j}^{(i)}$ is a random variable generated by the $j$-th sampling in the $k$-the iteration for machine $i$ and we assume that the stochastic gradient $G_{\xi_{k,j}^{(i)}}(\theta_k)$ satisfies (C2).
>
> Under (C2) and (A1), the sequence $(\theta_k := \frac{1}{P} \sum_{i=1}^P \theta_{k}^{(i)})$ generated by Parallelizing SGD satisfies the following:
>
> (i) [Upper bound on $\mathrm{VI}(K,\theta^\star)$]
> For all $K \geq 1$,
> \begin{align*}
> \mathbb{E} \left[ f\left(\frac{1}{K} \sum_{k=1}^K \theta_k \right) \right] - f(\theta^\star)
> \leq
> \mathrm{VI}(K,\theta^\star)
> \leq
> \frac{C_{1}}{K}+ \frac{C_2}{P b} + C_3,
> \end{align*}
> where $P$ is the number of machines,
> \begin{align*}
> C_{1} := \frac{\max_{i\in [P]} \mathbb{E}[\|\theta_1^{(i)} - \theta^\star \|^2]}{2 \alpha},
> \text{ }
> C_2 := \frac{\sigma^2 \alpha}{2},
> \text{ }
> C_3 := \frac{G^2 \alpha}{2}.
> \end{align*}
>
> (ii) [Steps to satisfy $\mathrm{VI}(K,\theta^\star) \leq \epsilon$]
> The number of steps $K_P$ defined by
> \begin{align*}
> K_P (b) = \frac{C_{1} P b}{(\epsilon - C_3)Pb - C_2}
> \end{align*}
> satisfies $\mathrm{VI}(K_P,\theta^\star) \leq \epsilon$
> and the function $K_P (b)$ is convex and monotone decreasing with respect to $b$ $(> \frac{C_2}{(\epsilon - C_3)P} > 0)$.
>
> (iii) [Minimization of SFO complexity]
> The SFO complexity defined by
> \begin{align*}
> N_P = K_P (b) b = \frac{C_{1} P b^2}{(\epsilon - C_3)Pb - C_2}
> \end{align*}
> is convex with respect to $b$ $(> \frac{C_2}{(\epsilon - C_3)P} > 0)$. The batch size
> \begin{align*}
> b_P^\star := \frac{2 C_2}{(\epsilon - C_3) P}
> \end{align*}
> attains the following minimum value of $N_P$:
> \begin{align*}
> N_P^\star = K_P (b_P^\star) b_P^\star = \frac{4 C_{1} C_2}{(\epsilon - C_3)^2 P}.
> \end{align*}
>
>
> Let $\theta_1^{(i)} = \theta_1$ $(i\in [P])$.
> Then, the SFO complexity $N^\star = K(b^\star) b^\star$ for unparallelizing SGD with batch size $b^\star = \frac{2 C_2}{\epsilon - C_3}$ is obtained by Theorem 3.1(iii), i.e.,
> \begin{align*}
> N^\star = N_1^\star = K_1 (b^\star) b^\star = \frac{4 C_1 C_2}{(\epsilon - C_3)^2}.
> \end{align*}
> Meanwhile, Parallelizing SGD with the batch size $b_P^\star = \frac{2 C_2}{(\epsilon - C_3)P}$,
> where $P$ $(>1)$ is the number of machines, has SFO complexity
> $N_P^\star$ such that
> \begin{align*}
> N_P^\star = \frac{4 C_{1} C_2}{(\epsilon - C_3)^2 P} <
> \frac{4 C_1 C_2}{(\epsilon - C_3)^2} = N^\star.
> \end{align*}
> Therefore, we can conclude that the larger $P$ is, the smaller SFO complexity is.
>
> The proofs of the above results (i)--(iii) are given in Appendix A.8 (revised).

---

> ### Author Response · Authors · 2022-11-19
> **Reply to Reviewer vE9x's comment: Diminishing step size results**
>
> The ``(this paper)" row in Table 1 (revised) indicates our analysis for not only constant step size but also diminishing step size. Please see Table 1 and Appendix A.4 in the revised manuscript.

---

### Official Review · Reviewer_U2qx · 2022-10-23

**Confidence:** 4
**Correctness:** 4
**Technical Novelty And Significance:** 3
**Empirical Novelty And Significance:** Not applicable
**Recommendation:** 5

**Clarity, Quality, Novelty And Reproducibility:**

For clarity and novelty, please refer to point 1 above.

This paper is of fair quality in its current form.



**Strength And Weaknesses:**

Overall, this paper presents novel results with solid foundations. My concerns mainly exist in the presentation. Some details are missing; some are not clear enough.

1. The presented results on critical results are new to me, though the suggestion of using a small batch size is not new. The current draft needs a more detailed, comprehensive comparison with the related results. As described in Section 1.2, some similar results have been seen in the literature. The authors need to clarify the advances this paper has made more clearly.

2. The proofs are given in the appendices in detail. However, I suggest the authors give a sketch in the main text. Also, the authors need to clarify the significance and difficulty of their proofs in the response and the revised draft.

**Summary Of The Paper:**

This paper studies the relationship between the batch size and the required step number to achieve the local optimum in the conditions of small constant learning rate and hyperparameters close to 1. The authors also present a critical batch size. These theoretical results are also supported by numerical results.

**Summary Of The Review:**

Overall, I recommend "5: marginally below the acceptance threshold." I need the authors to give more details in their responses, and if this paper is accepted, the draft needs a revision.

---

> ### Author Response · Authors · 2022-11-07
> **Replies to Reviewer U2qx's Main comments**
>
> **Comment 1:**
> The presented results on critical results are new to me, though the suggestion of using a small batch size is not new. ...
>
> **Reply:**
> As described in Section 1.2.2, the practical performance of a deep learning optimizer strongly depends on the batch size (Shallue et al., 2019; Zhang et al., 2019).
> The **merit** of this paper is that it clarifies theoretically the relationship between batch size and the performance of deep learning optimizers and develops a theory demonstrating the existence of critical batch sizes, which was shown numerically in (Shallue et al., 2019; Zhang et al., 2019).
> Motivated by the results in (Shallue et al., 2019; Zhang et al., 2019) and Section 1.2.2, we use SFO complexity $N := Kb$ as the performance measure of a deep learning optimizer.
> We first show that the number of steps $K$ to satisfy $\mathrm{VI}(K,\theta) \leq \epsilon$ can be defined as in Table 2 (revised), i.e.,
> \begin{align*}
> K = \frac{C_1 b}{(\epsilon - C_3)b - C_2}.
> \end{align*}
> As a function, $K$ is convex and monotone decreasing with respect to batch size $b$.
> Next, we show that SFO complexity $N$ defined by
> \begin{align*}
> N = \frac{C_1 b^2}{(\epsilon - C_3)b - C_2}
> \end{align*}
> is convex with respect to batch size $b$.
> This result agrees with the practical results in (Shallue et al., 2019; Zhang et al., 2019).
> Moreover, SFO complexity $N$ is minimized at $b^\star$ defined by
> \begin{align*}
> b^\star = \frac{2 C_2}{\epsilon - C_3}.
> \end{align*}
> This result guarantees the existence of the critical batch.
> However, the accurate setting of the critical batch size $b^\star$ would be difficult since $b^\star$ involves unknown parameters, such as $G$ and $D(\theta)$.
> The **advantage** of our analysis is that we can estimate appropriate batch sizes using the formula for $b^\star$ before implementing deep learning optimizers.
> Section 4 will discuss estimation of appropriate batch sizes in detail.
>
> Please see also Page 4 in the revised manuscript.
>
> **Comment 2:**
> The proofs are given in the appendices in detail. ...
>
> **Reply:**
> We give a brief outline of the proof strategy of Theorem 3.1, with an emphasis on the main difficulty that has to be overcome in order not to assume Lipschitz smoothness of $f$
> (i.e., $\nabla f$ is Lipschitz continuous with the Lipschitz constant $L$).
> First, we show that (C2) and (A1) imply that
> $(E[\|m_k\|])$ and
> $(E[\|\mathsf{d}_k\|])$ are
> bounded, where
> $\mathsf{d}_k :=-\mathsf{H}_k^{-1} \hat{m_k}$.
>
> Since we do not assume Lipschitz smoothness of $f$, we cannot use the descent lemma, i.e.,
> $f(y) \leq f(x) + \nabla f(x)^\top (y-x) + \frac{L}{2} \|y-x\|^2$
> $(x,y\in\mathbb{R}^d)$.
> This is the main **difficulty** proving Theorem 3.1.
> Almost all of the previous analyses of adaptive methods have been based on the descent lemma, and hence, they could use the expectation of the squared norm of the full gradient $E[\|\nabla f (\theta_k)\|^2]$ as the performance measure. In our case, we must use another performance measure. This paper uses the performance measure $\mathrm{VI}(K,\theta)$.
> We next show that
> $\sum_{k=1}^K E[m_{k-1}^\top (\theta_k -\theta)] \leq a_K + b_K + c_K \leq C_1 + \frac{C_2 K}{b} + \tilde{C}_3 K$, where $C_1$ and $C_2$ are defined as in Theorem 3.1 and $\tilde{C}_3 > 0$.
> In particular, (2) (revised manuscript), (A1), and (A2) imply $a_K \leq C_1$, the boundedness condition of $(E[\|\mathsf{d}_k\|])$ implies $b_K \leq \frac{C_2 K}{b}$,
> and (A2) and the Cauchy--Schwarz inequality imply $c_K \leq \tilde{C}_3 K$.
> The definition of $m_k$, the Cauchy--Schwarz inequality, the triangle inequality, and Jensen's inequality imply Theorem 3.1(i).
> Theorem 3.1(i) and $\frac{C_1}{K} + \frac{C_2}{b} + C_3 = \epsilon$ lead to Theorem 3.1(ii) and (iii).
>
> Please see also Page 7 in the revised manuscript.

---

### Official Review · Reviewer_hyhk · 2022-10-25

**Confidence:** 3
**Correctness:** 2
**Technical Novelty And Significance:** 3
**Empirical Novelty And Significance:** 3
**Recommendation:** 5

**Clarity, Quality, Novelty And Reproducibility:**

- Clarity: I have several concerns here, detailed in the weaknesses above. The meaning of the variational inequality formulation is not clear to me and is not discussed adequately in the paper (weakness 2). I also think related work needs to be discussed in more detail (weakness 1).

- Quality: I think the theoretical analysis has some two flaws, detailed in weakness 3 and 4 above. The absolute upper bounds on gradient-related quantities should be relaxed where possible. Crucially, one can not simply assume an algorithm to have a property that is provably does not have (weakness 4).

- Originality: To the best of my knowledge, the results are novel in the terms of the insights they provide w.r.t. the hyperparameter settings and the critical batch size.

**Strength And Weaknesses:**

### Strengths

1. I commend the effort to shed light on the influence of hyperparameter settings in adaptive gradient methods.

2. I found it very interesting to see the experimental analysis regarding the critical batch size (Fig. 2), especially seeing that the SFO complexity for Adam is minimized at a relatively large batch size, in contrast to SGD.


### Weaknesses

1. I partially disagree with the premise of the paper that previous works suggest a setting of $\beta_1\approx \beta_2 \approx 0$. I checked a few of the provided references and, e.g., Zaheer et al (2019) analyze a setting with $\beta_2$ close to 1 (they set $\beta_1$ to 0 though). Moreover, some of the cited papers study a situation where the batch size is increased over time, so one should be careful to transfer conclusions to the setting of a fixed batch size. I would ask that the authors make the discussion of prior work more precise and specify exact results which they think constitute a gap between theory and practice. Overall, I don't think the present version of the paper does a great job of positioning itself in the context of the (admittedly numerous) existing papers analyzing adaptive gradient methods.

2. The analysis is based on the "variational inequality" $K^{-1}\sum_k \nabla f(\theta_k)^T(\theta_k - \theta) \leq \epsilon$ for all $\theta$. I see why this is characterizing a stationary point for $\epsilon=0$, but this seems not very meaningful for $\epsilon>0$. I can just pick some $k$ and set $\theta = c (\theta_k - \nabla f(\theta_k))$ and if $\nabla f(\theta_k) \neq 0$, I can always break the inequality by scaling up $c$. I would like the authors to clarify: To what extent does this inequality really characterize a stationary point? Has prior work used a similar formulation (if so, it should be cited)? Why is it meaningful to establish an upper bound on this quantity?

3. The paper assumes a lot of absolute upper bounds on various gradient-related quantities, most strikingly $\Vert \nabla f(\theta_k)\Vert \leq G$. I know that many previous works have used this assumption, but I still see this very critically since it is completely unrealistic. Usually, it is possible to relax such assumptions. For example, Eq. (7) could be replaced with a bound of the form $\mathbf{E}[\Vert g_k- \nabla f(\theta_k) \Vert^2] \leq \sigma_0^2 + c \Vert \nabla f(\theta_k)\Vert^2$.

4. As the paper rightly points out, Adam is not guaranteed to converge on a convex stochastic optimization problem. The "breaking point" of the convergence proof is that Eq. (8) is _not_ fulfilled by Adam. The present paper goes on to simply _assume_ that Eq. (8) holds for Adam. In my opinion, that is not acceptable. You can analyze variants of Adam that enforce Eq. (8) to hold (AMSGrad) but you can not just _assume_ an algorithm to have a property that it demonstrably does not have. I would like the authors to clarify whether the presented analysis would extend to AMSGrad? If so, the paper should be adapted to reflect that.


### Update after Rebuttal

Thanks for clarifying point (2). This was mostly a misunderstanding on my part and I apologize for that! It seems quite clear in hindsight. Point (4) is still relevant and should be addressed in a possible camera-ready. I will increase my score.

**Summary Of The Paper:**

This paper presents an analysis of adaptive gradient methods with an emphasis on providing insights into the optimal setting of hyperparameters. The presented convergence results suggest settings that are in line with practical observations. From these results, the paper derives a "critical batch size", which minimizes the stochastic first-order oracle complexity.

**Summary Of The Review:**

Given the caveats detailed above, I recommend rejection. I hope that the authors can clarify some of my questions/concerns in the rebuttal phase.

---

> ### Comment · Reviewer_vE9x · 2022-11-05
> **Comment to Reviewer hyhk**
>
> Point 2. I feel $\theta$ should be instead $\theta^*$, but anyhow, $\theta$ should be a constant. You cannot set it as a function of $\theta_k$ for any $k$, but only for a single $k$, thus has very minor affect for the whole sum. Anyways, I think it shall be $\theta^*$ so that you have no worries.
>
> Point 3. I think Lipschitzness is quite typical, and a lot of cost functions are indeed Lipschitz. At least, as long as the iterations does not diverge (you have a regularizer to enforce it, and even if momentum methods are not monotone in function values in iterations, the path is usually in a bounded sublevel set), in a compact set a function is typically (of course not always) Lipschitz.
>
> Point 4. I agree, it's very important, authors please clarify. If the clarification is in the cited paper, please review it either in the main body or in appendix.

---

> ### Author Response · Authors · 2022-11-07
> **Replies to Reviewer hyhk's Main comments 1 and 2**
>
> **Comment 1:**
> I partially disagree with the premise of the paper that previous works suggest a setting of $\beta_1 \approx \beta_2 \approx 0$. ...
>
> **Reply:**
> Table 1 in the revised manuscript summarizes the previous results for upper bounds for the existing algorithms.
> The existing analyses in Table 1 show that using $\beta_1$ and/or $\beta_2$ close to 0 makes the upper bounds of the performance measures of the existing algorithms small.
> For example,
> Theorem 1 (Zaheer et al., 2018) and the proof of Theorem 1 in (Zaheer et al., 2018) show that, under the condition that $\nabla f$ is Lipschitz continuous with the Lipschitz constant $L$, Adam using $\alpha = \mathcal{O}(1/L)$ and $\beta_1 = 0$ satisfies
> \begin{align*}
> \frac{1}{K} \sum_{k=1}^K E [ \|\nabla f (\theta_k) \|^2 ]
> \leq 2 (\sqrt{\beta_2} G + \epsilon )
> \left(
> \frac{f(\theta_1) - f (\theta^\star)}{\alpha K}+(\frac{G \sqrt{1-\beta_2} }{\epsilon^2}+
> \frac{L \alpha}{2 \epsilon^2}
> )
> \frac{\sigma^2}{b}
> \right).
> \end{align*}
>
> This result implies that using large batch sizes makes the upper bound of $\frac{1}{K} \sum_{k=1}^K E[ \|\nabla f (\theta_k)\|^2 ]$ small.
>
> Theorem 3 in (Zhou et al., 2020) indicates that AMSGrad using a constant learning rate $\alpha$ satisfies
> \begin{align*}
> \frac{1}{K} \sum_{k=1}^K E\left[ \|\nabla f (\theta_k)\|^2 \right]
> \leq
> \frac{M_1}{K \alpha} + \frac{M_2 d}{K} + \frac{\alpha M_3 d}{K^{\frac{1}{2}-s}},
> \end{align*}
> where $s \in (0,1/2)$ and $M_i$ $(i=1,2,3)$ are positive constants, and
> \begin{align*}
> M_2 = \mathcal{O}\left(\frac{G^3}{1-\beta_1} \right) \text{ and }
> M_3 = \mathcal{O}\left(\frac{G^2}{1-\beta_2} \right)
> \end{align*}
> depend on $\beta_1$ and $\beta_2$ and the upper bound $G$.
> $M_2 $ and $M_3$ are small when $\beta_1$ and $\beta_2$ are small, i.e., $\beta_1, \beta_2 \approx 0$. Hence, using $\beta_1$ and $\beta_2$ close to $0$ is advantageous for AMSGrad.
>
> Hence, there is a gap between theory ($\beta_1, \beta_2 \approx 0$) and practice ($\beta_1, \beta_2 \approx 1$) for adaptive methods.
> Meanwhile, our analysis shows that using $\beta_1$ and $\beta_2$ close to $1$ makes the upper bound of $\mathrm{VI}(K,\theta)$ small, that is, the practical results using $\beta_1$ and $\beta_2$ close to 1 match our theoretical results.
>
> Please see Table 1 and Appendix A.1 (revised).
>
> **Comment 2:**
> The analysis is based on the "variational inequality ...
>
> **Reply:**
> We show the relationship between (i) $E[\nabla f(\theta_k)^\top (\theta_k - \theta)] \leq \epsilon$ $(\theta \in \mathbb{R}^d, k\in \mathbb{N})$ and (ii) $E[\|\nabla f(\theta_k)\|^2] \leq \epsilon$ $(k\in \mathbb{N})$,
> where $\epsilon > 0$ is the precision.
> Let us assume that $(\theta_k)$ is bounded.
> Suppose that (i) holds.
> Then, there exists a subsequence $(\theta_{k_i})$ of $(\theta_k)$ such that $(\theta_{k_i})$ converges to $\theta^*$.
> The continuity of $\nabla f$ thus implies that, for all $\theta\in\mathbb{R}^d$, $E[\nabla f(\theta^*)^\top (\theta^* - \theta)] \leq \epsilon$.
> Putting $\theta := \theta^* - \nabla f(\theta^*)$ ensures that $E[\|\nabla f(\theta^*)\|^2] \leq \epsilon$.
> Hence, the inequality (i) characterizes an approximate stationary point of $f$.
> Moreover, if we establish an upper bound of $\mathrm{VI}(K,\theta) := \frac{1}{K} \sum_{k=1}^K E[\nabla f(\theta_k)^\top (\theta_k - \theta)]$ for an optimizer and if the upper bound is less than or equal to $\epsilon$,
> then the optimizer approximates a stationary point of $f$ in the sense of the mean.
> Therefore, it is meaningful to use (i) and $\epsilon$-approximation
> $\mathrm{VI}(K,\theta) \leq \epsilon$ as the performance measure of an optimizer.
>
>
> We can check that
> \begin{align*}
>  \theta^* \in \mathbb{R}^d \colon \nabla f(\theta^*) = 0
> \text{ if and only if }
> \theta^* \in \mathbb{R}^d \colon \nabla f(\theta^*)^\top (\theta^*  - \theta) \leq 0 \text{ } (\theta \in \mathbb{R}^d).
> \end{align*}
> Hence, a point $\theta^* \in \mathbb{R}^d$ satisfying the variational inequality $\nabla f(\theta^*)^\top (\theta^*  - \theta) \leq 0$ $(\theta \in \mathbb{R}^d)$ is called a local minimizer of $f$
> (see, e.g., https://link.springer.com/book/10.1007/b97543 ).
> We do not know previous results using $\mathrm{VI}(K,\theta) := \frac{1}{K} \sum_{k=1}^K E[\nabla f(\theta_k)^\top (\theta_k - \theta)] \leq \epsilon$.
> However, there have been useful algorithms for solving variational inequalities
> (see, e.g., https://link.springer.com/book/10.1007/b97544 ).
>
> Please see also Page 3 in the revised manuscript.

---

> > ### Author Response · Authors · 2022-11-07
> > **Replies to Reviewer hyhk's Main comments 3 and 4**
> >
> > **Comment 3:**
> > The paper assumes a lot of absolute upper bounds on various gradient-related quantities ...
> >
> > **Reply:**
> > Let $f \colon \mathbb{R}^d \to \mathbb{R}$ be convex.
> > Then, $f$ is Lipschitz continuous (i.e., $|f(x)-f(y)| \leq G \|x-y\|$) if and only if $\|\nabla f(x)\| \leq G$ $(x\in\mathbb{R}^d)$ (see, e.g.,  Theorem 6.2.2, Corollary 6.1.2, and Exercise 6.1.9(c) in https://link.springer.com/book/10.1007/978-0-387-31256-9 ).
> > Let $\theta^*$ be a local minimizer of a Lipschitz continuous function $f$.
> > The continuity of $f$ ensures that $f$ is convex around $\theta^*$.
> > Hence, for any $\theta$ belonging to a neighborhood $N(\theta^*)$ of $\theta^*$, $\|\nabla f(\theta)\| \leq G$.
> > If the sequence $(\theta_k)$ generated by an optimizer approximates $\theta^*$, then $(\theta_k) \subset N(\theta^*)$ for sufficiently large $k$, i.e., $\|\nabla f(\theta_k)\| \leq G$. Please see also Appendix A.5 (revised).
> >
> > **Comment 4:**
> > As the paper rightly points out, Adam is not guaranteed to converge on a convex stochastic optimization problem. ...
> >
> > **Reply:**
> > An argument similar to that which obtained Theorem 3.1 (the proof of Theorem 3.1)
> > and the definition of AMSGrad (Algorithm 1 with $\hat{m_k} = m_k$, $\hat{v_k} = v_k$,
> > $\tilde{v_{k,i}} := \max (\tilde{v_{k-1,i}}, v_{k,i} )$,
> > and $H_k := \mathrm{diag}(\sqrt{\tilde{v_{k,i}}})$)
> > ensure that
> > $C_i$ $(i=1,2,3)$ for AMSGrad are
> > \begin{align*}
> > &C_1 := \frac{d D(\theta) \sqrt{M}}{2 \alpha \beta_1}, \text{ }
> > C_2 := \frac{\sigma^2 \alpha}{2 \sqrt{\tilde{v_*}} \beta_1},\\
> > \end{align*}
> > \begin{align*}
> > C_3 := \frac{G^2 \alpha}{2\sqrt{\tilde{v_*}} \beta_1}+ \mathrm{Dist}(\theta) \left( \frac{G(1-\beta_1)}{\beta_1}+ 2\sqrt{\sigma^2 + G^2}
> > \left(\frac{1}{\beta_1} + 2 (1-\beta_1) \right)  \right),
> > \end{align*}
> > where $\tilde{v_{-1}} := 0$ and $\tilde{v_*} := \inf_{k \in \mathbb{N}} \min_{i\in [d]} \tilde{v}_{k,i}$.
> >
> > **Proof:**
> > Fom the proofs in Appendix A.3.3 (revised), we can check that AMSGrad (Algorithm 1 with $\hat{m_k} = m_k$, $\hat{v_k} = v_k$,
> > $\tilde{v_{k,i}} := \max (\tilde{v_{k-1,i}}, v_{k,i} )$,
> > and $H_k := \mathrm{diag}(\sqrt{\tilde{v_{k,i}}})$) satisfies that
> > \begin{align*}
> > \sum_{k=1}^K a_k \leq \frac{d D(\theta) \sqrt{M}}{2 \beta_1 \alpha}, \text{ }
> > b_k \leq \frac{\alpha}{2 \sqrt{\tilde{v_*}} \beta_1} \left(  \frac{\sigma^2}{b} + G^2  \right), \text{ }
> > c_k \leq \mathrm{Dist}(\theta)G \frac{1-\beta_1}{\beta_1},
> > \end{align*}
> > where $a_k, b_k$, and $c_k$ are defined as in (22) (revised) and $\tilde{v_{k,i}} \leq M$ (by $v_{k,i} \leq M$, $\tilde{v_{-1,i}} = 0$, and mathematical induction).
> > Inequalities (21) and (33) (revised) thus imply that the assertion holds.
> >
> > Please see also Pages 6 and 7 and Appendix A.3.4 in the revised manuscript.

---

> > > ### Comment · Reviewer_hyhk · 2022-11-20
> > > **Regarding Comment 2**
> > >
> > > Thanks for engaging with my review. A follow-up on Comment 2:
> > >
> > > I might have some fundamental misunderstanding here, but the argument using the variational inequality still seems to break for me.
> > > You make a statement about a sequence of random variables $\theta_k$ for $k\in \mathbb{N}$.
> > > The statement is that there exists an upper bound $C$ such that
> > > $K^{-1}\sum_{k=1}^K  E[\nabla f(\theta_k)^T(\theta_k - \theta)] \leq C$
> > > for all $\theta\in\mathbb{R}^d$.
> > > But now I can pick a single $k$ and set $\theta = -\gamma E [\nabla f(\theta_k)]$. This choice is "allowed". It depends on $k$ but not on the realization of $\theta_k$; it is a deterministic vector in $\mathbb{R}^d$.
> > > By scaling $\gamma\rightarrow\infty$, I can break the inequality.
> > >
> > > Can you tell me whether my argument is wrong and where it breaks? Or am I misinterpreting the statement made in Theorem 3.1?

---

> > > > ### Author Response · Authors · 2022-11-20
> > > > **Reply to Reviewer hyhk's follow-up comment**
> > > >
> > > > Thank you for your comment.
> > > > We suppose that there exists $C \in \mathbb{R}$ such that, for all $\theta \in \mathbb{R}^d$ and all $K \geq 1$,
> > > > $\frac{1}{K} \sum_{k=1}^K E [\nabla f(\theta_k)^\top (\theta_k - \theta)] \leq C.$
> > > > Here, let $k = 1,2,\ldots,K$ be fixed and $\gamma > 0$.
> > > > We set $\theta = - \gamma E[\nabla f(\theta_k)]$.
> > > > Then, we have that, for all $K \geq 1$,
> > > > \begin{align*}
> > > > &\frac{1}{K}
> > > > (
> > > > E [\nabla f(\theta_1)^\top (\theta_1 + \gamma E[\nabla f(\theta_k)])]+
> > > > \cdots
> > > > \end{align*}
> > > > \begin{align*}
> > > > +E [\nabla f(\theta_k)^\top (\theta_k + \gamma E[\nabla f(\theta_k)])]+
> > > > \cdots+
> > > > E [\nabla f(\theta_K)^\top (\theta_K + \gamma E[\nabla f(\theta_k)])]
> > > > )
> > > > \leq C.
> > > > \end{align*}
> > > > Reviewer hyhk claimed that, if
> > > > $\gamma \to \infty$, then the above inequality does not hold (i.e., $\infty = C < \infty$).
> > > > However, we do not understand why the claim holds.
> > > > We might misread your comment.
> > > > Hence, we would be grateful if you could explain your claim, in particular, proof such that the inequality is broken, explicitly.

---

> > > > > ### Comment · Reviewer_hyhk · 2022-11-21
> > > > > **Why do I have to give every comment a title?**
> > > > >
> > > > > I can specialize to $K=1$. That is enough to contradict Theorem 3.1(i), since it is a statement about _all_ $K$. Then the choice $\theta = -\gamma E[\nabla f(\theta_1)]$ makes the left-hand side of Theorem 3.1 (i) $E[\nabla f(\theta_1)^T(\theta_1 + \gamma E[\nabla f(\theta_1)])] = E[\nabla f(\theta_1)^T\theta_1] + \gamma \Vert E[\nabla f(\theta_1)]\Vert^2$. The first term is finite and independent of $\gamma$. Letting $\gamma \rightarrow \infty$ leads to the l.h.s. going to infinity. Is that not a contradiction to the theorem?

---

> > > > > > ### Author Response · Authors · 2022-11-21
> > > > > > **Reply to Reviewer hyhk's comment**
> > > > > >
> > > > > > Thank you for your comment.
> > > > > > Let us consider SGD with $K=1$, $\gamma > 0$, and $\theta = - \gamma E[\nabla f(\theta_1)]$.
> > > > > > Then, Theorem 3.1(i) implies that
> > > > > > \begin{align*}
> > > > > > VI(1,\theta)
> > > > > > = E [\nabla f(\theta_1)^\top (\theta_1 + \gamma E[\nabla f(\theta_1)])]
> > > > > > = E[\nabla f(\theta_1)^\top \theta_1] + \gamma \|E[\nabla f(\theta_1)]\|^2
> > > > > > \end{align*}
> > > > > > \begin{align*}
> > > > > > \leq \frac{C_1}{1} + \frac{C_2}{b} + C_3= \frac{E[\|\theta_1 + \gamma E[\nabla f(\theta_1)]\|^2]}{2 \alpha} + \frac{\sigma^2 \alpha}{2b} + \frac{G^2 \alpha}{2}\\
> > > > > > &\leq
> > > > > > \frac{E[2\|\theta_1\|^2 + 2\gamma^2 \|E[\nabla f(\theta_1)]\|^2]}{2 \alpha} + \frac{\sigma^2 \alpha}{2b} + \frac{G^2 \alpha}{2},
> > > > > > \end{align*}
> > > > > > which implies that
> > > > > > \begin{align*}
> > > > > > \frac{E[\nabla f(\theta_1)^\top \theta_1]}{\gamma} + \|E[\nabla f(\theta_1)]\|^2
> > > > > > \leq
> > > > > > \frac{E[\|\theta_1\|^2]}{\alpha \gamma}+ \frac{\gamma \|E[\nabla f(\theta_1)]\|^2}{\alpha}+ \frac{\sigma^2 \alpha}{2b \gamma} + \frac{G^2 \alpha}{2\gamma}.
> > > > > > \end{align*}
> > > > > > Even if $\gamma \to \infty$, we do not have a contradiction.

---

> > > > > > > ### Comment · Reviewer_hyhk · 2022-11-21
> > > > > > > **Reply**
> > > > > > >
> > > > > > > Regarding comment 2: Ah, I had overlooked that the right-hand side of the inequality also depends on the choice of $\theta$. Sorry about that! I will reassess in light of that.

---

> > > > > > > > ### Author Response · Authors · 2022-11-21
> > > > > > > > **Thank you**
> > > > > > > >
> > > > > > > > We thank you for careful reading of our manuscript and important comments.

---

> ### Author Response · Authors · 2022-11-26
> **Reply to Reviewer hyhk's comment**
>
> We would like to thank Reviewer hyhk for reviewing our paper and updating recommendation score.
>
> **Reply to Point (4):** We plan to modify the sentence (Page 6, Line +2; We thus assume condition (2) for Adam to guarantee the convergence of Adam) with
> "We thus consider Adam with $H_k  := \mathrm{diag}(\sqrt{\tilde{v_{k,i}}})$, where
> $\tilde{v_{k,i}} := \max ( \tilde{v_{k-1,i}}, \hat{v_{k,i}} )$."
> Moreover, we would like to revise related parts in the manuscript (e.g., "Adam with (2)" will be replaced with "Adam with $H_k  := \mathrm{diag}(\sqrt{\tilde{v_{k,i}}})$") accordingly.

---

### Official Review · Reviewer_vJ3a · 2022-10-29

**Confidence:** 3
**Correctness:** 2
**Technical Novelty And Significance:** 2
**Empirical Novelty And Significance:** 1
**Recommendation:** 3

**Clarity, Quality, Novelty And Reproducibility:**

The paper is not very well written.

1. I suggest using a different notation for oracle complexity, something like $N$ instead of $Kb$. It might be a bit confusing to think of $Kb$ as a function of the batch size.

2. The paper mentions it studies stochastic first-order oracle complexity and does so in its theory because it considers a stochastic optimization problem. However, most of the previous empirical work mentioned in the paper uses SGD with multiple passes, i.e., it doesn't use sampling with replacement, which means different batches are not independent (for context on theory, consider [this paper](https://proceedings.neurips.cc/paper/2021/hash/e64c9ec33f19c7de745bd6b6d1a7a86e-Abstract.html)). This is confusing and can lead the readers to believe that there is a watertight connection between theory and experiments when there is not. Were the experiments in this paper performed using sampling with replacement? It is important to specify this explicitly in algorithm 1.

3. Why do the authors mention Chen et al. reference for the convergence of SGD? There are several older papers analyzing SGD. In particular, in the convex setting, [this paper](https://arxiv.org/abs/1106.4574) discusses the linear scaling with batch size and when it fails to hold. In the non-convex setting, [this paper](https://arxiv.org/pdf/1912.02365.pdf) shows the optimality of several existing analyses for SGD. Finally, which results are people talking about in the following statement:
> Accordingly, the practical results for large batch sizes match the theoretical ones

4. It is unclear what is the metric in equation (5). The sentence below the equation as well doesn't explain the metric.

5. It would be good to indicate predicted critical batch sizes for different algorithms in the plots in the experimental section. Furthermore, how was the learning rate tuned for SGD and SGD with momentum while changing the batch size? Ideally, the learning rate and momentum should be tuned separately for each batch size, but from the paper, it seems like a constant learning rate of $10^{-3}$ was used.

**Strength And Weaknesses:**

The first major problem with the paper is assuming both bounded variance and bounded gradients. It is well known that in this setting, mini-batching doesn't help because for usual problems $G\approx \sigma$ unless there is some stylized noise structure. Bounded variance assumptions are usually paired with smoothness assumptions, for instance, [this paper](https://arxiv.org/pdf/1912.02365.pdf). Under the paper's assumptions, if $\sigma$ is replaced by $G$, then $C_2 \leq C_3$ so there is no benefit of mini-batching! Similarly, it is not reasonable to assume the boundedness of parameters without explicitly making any projection steps in the algorithm.

The second major problem is that the paper makes no comparison to the existing upper bounds for these algorithms. As a result, it is unclear if Adam's analysis is an improvement over the existing work. And if the upper bounds are worse than the previous bounds, then it is a fallacy to make any conclusions about the optimal hyper-parameters. Ideally, one would never make such conclusions with just upper bounds without matching lower bounds.

Another issue is that the final guarantees in the paper are not comparable to usual guarantees upper bounding $\frac{1}{K}\sum_{k\in[K]} \mathbb{E}||\nabla f(\theta_k)||^2$, which guarantees that at least one iterate was an approximate stationary point. The paper instead uses a variational metric, but it is not reasonable to average it across $k$ because that doesn't imply that the best $\theta_k$ satisfies the variational metric for all $\theta$.

Overall, the paper is not technically sound.

**Summary Of The Paper:**

The paper studies the gap between the theoretical and empirical optimal hyperparameters values for adaptive gradient methods such as Adam for stochastic non-convex optimization. It provides new convergence upper bounds for several of these methods under bounded gradients and variance assumptions. These upper bounds are shown to be the lowest for the aforementioned hyperparameters being close to one, which is close to practice.



**Summary Of The Review:**

The paper tries to study an interesting gap between the theory and practice of adaptive methods. However, it lacks a clear comparison to existing theoretical results, and the provided guarantees seem weaker than usual stationarity guarantees. The experiments don't have any new insights, as similar experiments exist on much larger datasets in the references cited within the paper. I thus recommend rejecting the paper.

---

> ### Author Response · Authors · 2022-11-07
> **Replies to Reviewer vJ3a's Main comments**
>
> **Comment (I):**
> The first major problem with the paper is assuming both bounded variance and bounded gradients. ....
>
> **Reply:**
> If $b$ is large, then $\sigma$ is small.
> Under $G \approx \sigma$, $G$ is also small.
> Hence, $C_2$ and $C_3$ become small.
> Since the upper bound $\frac{C_1}{K} + \frac{C_2}{b} + C_3$ of $\mathrm{VI}(K,{\theta})$ is small, it is a desirable result.
>
> We may replace $\theta_{k+1} = \theta_k - \alpha H_k^{-1} \hat{m_k}$ (step 9 in Algorithm 1) with
> $\theta_{k+1} = P(\theta_k - \alpha H_k^{-1} \hat{m_k})$, where $P$ is the projection onto a bounded, closed convex set $C$.
> Then, the boundedness of $C$ implies that (A2) (the boundedness of parameters) holds.
> Using the nonexpansivity of $P$ (i.e., $\|P({x}) - P({y})\| \leq \|{x}-{y}\|$) and the proof of Lemma A.1 ensures that, for all ${\theta}\in C$,
> \begin{align*}
> E [ \| \theta_{k+1} - \theta \|^2 ]
> \leq
> E [ \| \theta_k - \theta \|^2 ] + \alpha^2 E [  \| d_k \|^2 ] + 2 \alpha \left(
> \frac{\beta_1}{1 - \beta_1^{k+1}}
> E [ (\theta - \theta_k)^\top m_{k-1} ]
> +\frac{1 -\beta_1}{1 - \beta_1^{k+1}}
> E [ (\theta -\theta_k)^\top \nabla f (\theta_k) ]
> \right).
> \end{align*}
> A discussion similar to the one showing Theorem 3.1 ensures that Algorithm 1 with $\theta_{k+1} = P(\theta_k - \alpha H_k^{-1} \hat{m_k})$ satisfies the results in Theorem 3.1 for all $\theta \in C$.
>
> **Comment (II):**
> The second major problem is that the paper makes no comparison to the existing upper bounds for these algorithms. ...
>
> **Reply:**
> Table 1 in the revised manuscript summarizes the previous results for upper bounds for the existing algorithms.
> The existing analyses in Table 1 show that using $\beta_1$ and/or $\beta_2$ close to 0 makes the upper bounds of the performance measures of the existing algorithms small.
> For example,
> Theorem 1 (Zaheer et al., 2018) and the proof of Theorem 1 in (Zaheer et al., 2018) show that, under the condition that $\nabla f$ is Lipschitz continuous with the Lipschitz constant $L$, Adam using $\alpha = \mathcal{O}(1/L)$ and $\beta_1 = 0$ satisfies
> \begin{align*}
> \frac{1}{K} \sum_{k=1}^K E [ \|\nabla f (\theta_k) \|^2 ]
> \leq 2 (\sqrt{\beta_2} G + \epsilon )
> \left(\frac{f(\theta_1) - f (\theta^\star)}{\alpha K}+ ( \frac{G \sqrt{1-\beta_2}}{\epsilon^2}+
> \frac{L \alpha}{2 \epsilon^2})
> \frac{\sigma^2}{b}\right).
> \end{align*}
> Theorem 3 in (Zhou et al., 2020) indicates that AMSGrad using a constant learning rate $\alpha$ satisfies
> \begin{align*}
> \frac{1}{K} \sum_{k=1}^K E\left[ \|\nabla f (\theta_k)\|^2 \right]
> \leq
> \frac{M_1}{K \alpha} + \frac{M_2 d}{K} + \frac{\alpha M_3 d}{K^{\frac{1}{2}-s}},
> \end{align*}
> where $s \in (0,1/2)$ and $M_i$ $(i=1,2,3)$ are positive constants, and
> \begin{align*}
> M_2 = \mathcal{O}\left(\frac{G^3}{1-\beta_1} \right) \text{ and }
> M_3 = \mathcal{O}\left(\frac{G^2}{1-\beta_2} \right)
> \end{align*}
> depend on $\beta_1$ and $\beta_2$ and the upper bound $G$.
> $M_2 $ and $M_3$ are small when $\beta_1$ and $\beta_2$ are small, i.e., $\beta_1, \beta_2 \approx 0$. Hence, using $\beta_1$ and $\beta_2$ close to $0$ is advantageous for AMSGrad.
>
> Hence, there is a gap between theory ($\beta_1, \beta_2 \approx 0$) and practice ($\beta_1, \beta_2 \approx 1$) for adaptive methods.
> Meanwhile, our analysis shows that using $\beta_1$ and $\beta_2$ close to $1$ makes the upper bound of $\mathrm{VI}(K,\theta)$ small, that is, the practical results using $\beta_1$ and $\beta_2$ close to 1 match our theoretical results.
>
> Please see Table 1 and Appendix A.1 (revised).
>
> **Comment (III):**
> Another issue is that the final guarantees in the paper are not comparable to usual guarantees upper bounding ...
>
> **Reply:**
> We show the relationship between (i) $E[\nabla f(\theta_k)^\top (\theta_k - \theta)] \leq \epsilon$
> $(\theta \in \mathbb{R}^d, k\in \mathbb{N})$ and (ii) $E [\|\nabla f(\theta_k)\|^2] \leq \epsilon$ $(k\in \mathbb{N})$,
> where $\epsilon > 0$ is the precision.
> Let us assume that $(\theta_k)$ is bounded.
> Suppose that (i) holds.
> Then, there exists a subsequence $(\theta_{k_i})$ of $(\theta_k)$ such that $(\theta_{k_i})$ converges to $\theta^*$.
> The continuity of $\nabla f$ thus implies that, for all $\theta \in\mathbb{R}^d$, $E[\nabla f(\theta^*)^\top (\theta^* - \theta)] \leq \epsilon$.
> Putting $\theta := \theta^* - \nabla f(\theta^*)$ ensures that $E [\|\nabla f(\theta^*)\|^2] \leq \epsilon$.
> Suppose that (ii) holds.
> Then, the definition of the inner product and Jensen's inequality imply that, for all $\theta \in \mathbb{R}^d$,
> $E[\nabla f(\theta_k)^\top (\theta_k - \theta)]
> \leq \mathrm{Dist}(\theta) \sqrt{\epsilon}$, where $\mathrm{Dist}(\theta) := \sup_{k\in \mathbb{N}} \|\theta_k - \theta\| < + \infty$.
> Therefore, it is adequate to use (i) and $\epsilon$-approximation
> $\mathrm{VI}(K,\theta) := \frac{1}{K} \sum_{k=1}^K \mathbb{E}[\nabla f(\theta_k)^\top (\theta_k - \theta)] \leq \epsilon$
> as the performance measure of an optimizer.
>
> Please see also Page 3 in the revised manuscript.

---

> > ### Author Response · Authors · 2022-11-07
> > **Replies to Reviewer vJ3a's other comments**
> >
> > **1:**
> > I suggest using a different notation for oracle complexity
> >
> > **Reply:**
> > We revised the manuscript accordingly.
> >
> >
> > **2:**
> > The paper mentions it studies stochastic first-order oracle complexity
> >
> > **Reply:**
> > We check the framework of SGD with multiple passes and find that this paper does not use SGD with multiple passes.
> >
> > **3:**
> > Why do the authors mention Chen et al. reference for the convergence of SGD?
> >
> > **Reply:**
> > Thank you for your comments. We cite some references for the convergence of SGD.
> >
> > **4:**
> > It is unclear what is the metric in equation (5).
> >
> > **Reply:**
> > We are sorry for the confusion.
> > In the revision, we omit (5) due to lack of space.
> >
> > **5:**
> > It would be good to indicate predicted critical batch sizes for different algorithms in the plots in the experimental section.
> >
> > **Reply:**
> > Thank you for your valuable comments.
> > We understand that SGD and Momentum using tuned learning rate for each batch size have good performances.
> > Our goal is to estimate appropriate batch sizes from the formula for $b^\star = \frac{2 C_2}{\epsilon - C_3} > \frac{2 C_2}{\epsilon}$ before implementing optimizers.
> > The definition of $C_2$ uses $\alpha$ and $\beta_1$.
> > Hence, we must set $\alpha$ and $\beta_1$ before implementing optimizers.
> > Here, we use well-known parameters $\alpha = 10^{-3}$ and $\beta_1 = 0.9$.
> > Then, we estimate appropriate batch sizes using fixed $\alpha$ and $\beta_1$ and $b^\star > \frac{2 C_2}{\epsilon}$ before implementing optimizers.
> > As a result, we would like to check whether or not actual critical batch sizes (obtained from numerical experiments for optimizers using $\alpha = 10^{-3}$ and $\beta_1 = 0.9$) are close estimated batch sizes (using $\alpha = 10^{-3}$ and $\beta_1 = 0.9$).

---

> > ### Comment · Reviewer_vJ3a · 2022-12-05
> > **Reply**
> >
> > Hi, sorry for the late reply. The replies to comments II and III and additional explanations in the paper make sense to me. Thanks for adding those. I am unsure if the authors understand my issue about critical batch sizes in the experiments. I am asking to indicate the predicted batch sizes on the x-axis in figures 3 and 4.
> >
> > Regarding my comment I, I don't think the authors have answered my question. $\sigma$ is not a function of $b$, $\sigma/\sqrt{b}$ i.e., the effective variance is a function of $b$. For most practical problems, unless the noise structure is stylized (such as additive noise with fixed variance), $\sigma$ and $G$ are of the same order. This means the paper's results imply no benefit of mini-batching. Since the whole point of the paper is to find the critical batch size, I am unsure what to take away from this. To be more precise, when $\sigma = O(G)$, as it would be for most Lipschitz functions with reasonable noise, in the upper bounded provided, the dominating terms have no batch size $b$. Thus there is no way to obtain a critical batch size from the upper bound. To say anything meaningful, the authors need to do the following:
> >
> > 1. either relax the Lipschitz ness condition to something like smoothness + bounded variance
> >
> > 2. or identify a significant problem where $\sigma/b=\Omega(G)$ so that there is a provable benefit of mini-batching.
> >
> >
> > Thus, I maintain my score.

---

> > > ### Comment · Reviewer_vE9x · 2022-12-06
> > > **Solution**
> > >
> > > First I do not think the noise is always large, it can be small. And it is easy to imagine that if they use a constant step size, then the noise variance will always exist in the error. That's why I suggest the authors to add the diminishing step size so the error can tend to 0 with any noise.
> > >
> > > I do not think there is an "equivalent" noise $\sigma/b$, I think the noise is $\sigma$, but diminishing step size solves the problem.

---

> > > ### Author Response · Authors · 2022-12-06
> > > **Replies to Reviewer vJ3a's comments**
> > >
> > > **Comment 1:**
> > > I am asking to indicate the predicted batch sizes on the x-axis in figures 3 and 4.
> > >
> > > **Reply:**
> > > We would like to revise Figures 1--4 to make the predicted batch sizes (the predicted batch size in Figures 1--2 is 1600 and the predicted batch size in Figures 3--4 is 2000) visible.
> > >
> > > **Comment 2:**
> > > Regarding my comment I, I don't think the authors have answered my question. ...
> > >
> > > **Reply:**
> > > We thank you for your suggestions. We consider your suggestion 1:
> > > "relax the Lipschitz ness condition to something like smoothness + bounded variance."
> > > First of all, this paper does not assume the Lipschitz smoothness condition of $f$ (i.e., $\nabla f$ is Lipschitz continuous).
> > > This paper assumes the continuous differential condition of $f$ (C1), the bounded variance (C2), the mini-batch stochastic gradient (C3), the gradient boundedness (A1), and the bounded sequence $(\theta_k)$ (A2).
> > > We may replace (A2) with $\theta_{k+1} = P(\theta_k - \alpha H_k^{-1} \hat{m}_k)$, where $P$ is the projection onto a bounded, closed convex set (please see also Reply to Comment (I)).
> > > (A2) implies that there exists a compact set $C \subset \mathbb{R}^d$ such that
> > > $(\theta_k) \subset C$.
> > > The continuity conditions of $\nabla f$ (see (C1)) and the norm of $\mathbb{R}^d$ imply that $(\nabla f(\theta_k))$ is bounded.
> > > This implies that the gradient boundedness (A1) holds.
> > > Hence, we may omit (A1).
> > > Meanwhile, the Lipschitz continuity of $f$ (not the Lipschitz continuity of $\nabla f$), which is a standard condition, ensures that (A1) holds (please see Appendix A.5).
> > > Although this paper assumes (A1) so as to match previous results (e.g., Kingma--Ba, 2015),
> > > we can omit (A1) to avoid confusion.
> > >
> > > Therefore, the essential assumptions in this paper are the continuous differential condition of $f$ (C1) and the bounded variance (C2).
> > > We thus think that your requirement (suggestion 1: "relax the Lipschitz ness condition to something like smoothness + bounded variance") is satisfied.

---

### Author Response · Authors · 2022-11-07
**Our gratitude to the four reviewers**

We would like to express our gratitude to the four reviewers for their valuable comments on our manuscript. We appreciate their detailed assessments and helpful feedback. We have revised the manuscript to incorporate the recommendations, which has resulted in an improved presentation of our work. The revised parts of the manuscript are marked in red.

---

### Comment · Reviewer_vE9x · 2022-11-07
**[Closed] Could the authors please upload the original submission for reference?**

The reviewers mentioned specific sections/paragraphs/equations, which are now relocated with the new revision. Could the authors upload the original manuscript in supplementary material for reference for our first-stage reviews?

===================

Oh I found it by clicking "Show Revisions", never mind.

---

### Decision · Program_Chairs · 2023-01-20

**Decision:**

Reject

**Justification For Why Not Higher Score:**

See above.

**Justification For Why Not Lower Score:**

N/A

**Metareview: Summary, Strengths And Weaknesses:**

This paper attempts to theoretically analyze some empirical observations about critical batch sizes of optimization algorithms. The paper derives some convergence bounds for Adam using a Variational Inequality convergence measure. Based on these bounds, they derive a formula for the critical batch size.

One one hand, reviewers generally feel like there is probably a new and interesting contribution here. On the other hand, they have raised a lot of specific concerns regarding clarity, relationships to prior analyses of Adam, reasonableness of various assumptions, and the meaningfulness of the experimental results. There has been quite a bit of discussion, and the authors have persuaded some of the reviewers on some of the specific points.

However, much of the discussion points at a central concern I have, which is that the contribution relative to prior work isn't stated very clearly. There have been a lot of convergence analyses of Adam and related algorithms, and a lot of analyses of batch size scaling of stochastic optimizers (including the classic Bottou and Bousquet (2007), "The tradeoffs of large-scale learning", Ma et al. (2018), "The power of interpolation", etc., as well as various references the reviewers point out). The papers I just listed don't investigate exactly the same issues (and aren't necessarily the most relevant), but it feels like there needs to be some discussion of what is the current state of knowledge regarding batch size scaling, where current analyses fall short, and how this paper fills the gap.

I also don't feel like I have a clear picture of what we learn from the experiments; the take-aways are never quite stated. I presume it is meant to validate some of the theoretical predictions, but which ones?  From the experiments, the SFO metric also seems to behave qualitatively differently from more traditional notions of batch size which are relevant to practical optimization; e.g., it seems to indicate the critical batch size for SGD is <= 4; this makes me question how practically relevant the analyses are.

Overall, I think there is some promise here, but the presentation seems to require enough cleanup that it should go through another round of review.